# Assessment of Contemporary Satellite Sea Ice Thickness Products for Arctic Sea Ice

Heidi Sallila[1], Sinéad Louise Farrell [2], Joshua McCurry[3] and Eero Rinne[1]

[1]Finnish Meteorological Institute, Helsinki, Finland
5 [2]Cooperative Institute for Climate and Satellites, University of Maryland, College Park, MD, USA
[3]Department of Atmospheric and Oceanic Science, University of Maryland, College Park, MD, USA

*Correspondence to*: Heidi Sallila (heidi.sallila@fmi.fi)

**Abstract.** Advances in remote sensing of sea ice over the past two decades have resulted in a wide variety of satellite-derived sea ice thickness data products becoming publicly available. Selecting the most appropriate product is challenging given end user objectives range from incorporating satellite-derived thickness information in operational activities, including 15 sea ice forecasting, routing of maritime traffic, and search and rescue, to climate change analysis, longer-term modeling, prediction, and future planning. Depending on the use case, selecting the most suitable satellite data product can depend on the region of interest, data latency, and whether the data are provided routinely, for example via a climate or maritime service provider. Here we examine a suite of current sea ice thickness data products, collating key details of primary interest to end users. We assess eight years of sea ice thickness observations derived from sensors onboard the CryoSat-2 (CS2), 20 Advanced Very-High-Resolution Radiometer (AVHRR) and Soil Moisture and Ocean Salinity (SMOS) satellites. We evaluate the satellite-only observations with independent ice draft and thickness measurements obtained from the Beaufort Gyre Exploration Project (BGEP) upward looking sonars (ULS) and Operation IceBridge (OIB), respectively. We find a number of key differences among data products, but find that products utilizing CS2-only measurements are reliable for sea ice thickness, particularly between ~0.5 m and 4 m. Among data compared, a blended CS2-SMOS product was the most 25 reliable for thin ice. Ice thickness distributions at the end of winter appeared realistic when compared with independent ice draft measurements, with the exception of those derived from AVHRR. There is disagreement among the products in terms of the magnitude of the mean thickness trends, especially in spring 2017. Regional comparisons reveal noticeable differences in ice thickness between products, particularly in the marginal seas in areas of considerable ship traffic.

## 1 Introduction

With the observed decline in Arctic sea ice extent (Parkinson and Cavalieri, 2008; Markus et al., 2009; Perovich et al., 2018) and interests in the exploitation of regional natural resources, human activities in the Arctic have increased, alongside concerns for the state of the ice cover. Numerous objectives, ranging from sea ice forecasting and climate monitoring to navigation, require observations of Arctic sea ice, and it is recognised as an Essential Climate Variable (ECV) in the Global Climate Observing System (GCOS, Belward and Dowell, 2016). Given its remote location, seasonally-available sunlight, and inhospitable climate, remote sensing provides the only means to obtain routine, basin-scale, and sustained observations of the Arctic Ocean.

Although sea ice extent is traditionally the most widely discussed variable, sea ice thickness measurements are just as important, and needed together with ice concentration to calculate sea ice volume, the best indicator of change in the Arctic ice cover (e.g., Laxon et al., 2013; Song, 2016). Even if ice extent remains stable in consecutive years, if the thickness decreases, the ice cover will be less resilient and more unlikely to regain thickness, eventually leading to decreased extent and volume. For example, Laxon et al. (2013) speculated that lower ice thickness, and volume, may have been a contributing factor to the September 2012 record minimum sea ice extent. As a result of advances over two decades in remote sensing of Arctic sea ice, a wide variety of satellite sea ice thickness data products have become available to the scientific community. Radar altimeter measurements from CryoSat-2 (Wingham et al., 2006) and laser altimeter measurements from ICESat and ICESat-2 (Markus et al., 2017) are major sources for estimating sea ice thickness, and due to high-inclination orbits, provide nearly full coverage of the Arctic Ocean (Laxon et al., 2013; Markus et al., 2017). Thickness is derived from processed altimeter measurements of ice surface elevation (Laxon et al., 2013; Kurtz et al. 2014), which requires knowledge of snow loading, and the assumption of hydrostatic equilibrium (Giles et al. 2007). Because of differing approaches for retracking radar altimeter waveforms, a variety of processing algorithms exist, resulting in an array of altimeter-derived thickness data products (e.g., Laxon et al., 2013; Kurtz et al., 2014; Ricker et al. 2014; Price et al. 2015; Tilling et al., 2015, 2016, 2018). These products broadly agree in the spatial distribution and basin-scale gradients of ice thickness across the Arctic Ocean, but differ in their absolute magnitude. Thickness may also be derived from passive microwave radiometer measurements, as well as from visible and synthetic aperture radar imagery, and these observations offer additional information such as coverage in the marginal ice zone or detection of sea ice conditions during the Arctic summer (e.g., Kaleschke et al., 2012; Key and Wang, 2015; Ricker et al., 2017a). Recent studies have demonstrated the utility of initialising numerical models with satellite-derived estimates of sea ice thickness to improve model predictions (Yang et al., 2014; Allard et al., 2018; Blockley and Peterson, 2018; Stroeve et al., 2018; Xie et al., 2018).

Given the variety of sea ice thickness data products that are now available, and broad observational objectives, ranging from advancing long-term climate predictions and future planning, to supporting operational activities, including navigation, commerce, hazard monitoring, search and rescue, and disaster response, identifying the most suitable product depends on the end user requirements. Determining which satellite data product is the most appropriate depends upon a variety of factors, including the end user's region of interest, and data product characteristics including spatial coverage, temporal and spatial resolution, accuracy and quality, as well as data availability and latency. In this paper we review a set of publicly-available satellite-derived sea ice thickness products, and compare them side by side for the first time, outlining their key attributes which are of interest to potential end users and for a range of applications. Although previous studies have evaluated individual satellite products (e.g. Laxon et al., 2013; Ricker et al., 2017b; Wang et al., 2016) and assessed the impact of retracker differences (e.g. Xia and Xie, 2018; Yi et al., 2019), none have focused on product intercomparison and have therefore lacked the details sought by many end users. To address this need, we compare data product attributes and assess differences across products from both a regional and seasonal perspective, across the central Arctic Ocean and peripheral seas (Figure 1). We also evaluate the satellite data products, through comparisons with independent thickness measurements obtained in situ.

The paper is structured as follows: Section 2 introduces the sea ice thickness products and the independent data sets used for validation. Section 3 describes the methods used for product comparison. In Section 4 we present the results of the comparisons across satellite products, and a product evaluation against independent estimates of ice thickness obtained from in situ and airborne sensors. We provide a summary discussion in Section 5. In Section 6 we conclude with a look ahead and provide recommendations for future satellite-derived thickness products aimed at addressing operational needs.

**2 Data**

**2.1 Satellite Data Products for Arctic Sea Ice Thickness**

Here we assess six contemporaneous sea ice thickness data products, derived from satellite measurements collected by CryoSat-2 (CS2), the Advanced Very-High-Resolution Radiometer (AVHRR), and the Soil Moisture and Ocean Salinity satellite (SMOS). Since our focus is an assessment of product utility for a range of operational activities we required that data products were open access, had basin-wide coverage of the Arctic Ocean (Figure 1), and were available for the majority of the CS2 observation period (April 2010 to present). Four products comprise CS2-only thickness estimates, and include the Centre for Polar Observation and Modeling (CPOM) seasonally-averaged thickness product (Laxon et al., 2013, Tilling et al., 2018), the Alfred Wegener Institute (AWI) monthly thickness product (Ricker et al., 2014), the NASA Jet Propulsion

Laboratory (JPL) monthly thickness product (Kwok and Cunningham, 2015) and the NASA Goddard Space Flight Center (GSFC) 30-day thickness product (Kurtz and Harbeck, 2017). For comparison with CS2-only thickness estimates, we also consider blended CS2 - Soil Moisture Ocean Salinity (CS2SMOS) weekly ice thickness data (Ricker et al., 2017a), and the NOAA Climate Data Record (CDR) Extended Advanced Very High Resolution Radiometer (AVHRR) Polar Pathfinder

(APP-x) daily ice thickness product. Further details about each data product are provided below. Common characteristics including measurement technique, temporal and spatial coverage, latency, frequency, resolution, and algorithm-specific details, are outlined in Table 1. The products selected for assessment provide a representative sample of available sources. We acknowledge that the list of satellite-derived ice thickness products is not exhaustive and other sources of similar observations exist. For example two additional sources of CS2-only sea ice data are now publicly available, one from the

ESA Climate Change Initiative (CCI, Hendricks et al., 2018) and a second from the Laboratoire d'Études en Géophysique et Océanographie Spatiales (LEGOS) Center for Topographic studies of the Ocean and Hydrosphere (CTOH, Guerreiro et al., 2017). Details of these two products are included in Table 1 for completion but not included in the analysis. Sea ice age characterisation, and ice thickness, from the Visible Infrared Imager Radiometer Suite (VIIRS) instrument on the Suomi National Polar-orbiting Partnership (S-NPP) satellite, may also be derived in a similar manner to the APP-x data product

(Key et al., 2013), however these data were not publicly available at the time of writing, and are hence not included in the analysis. Since our focus is the availability of current sea ice thickness measurements for end users, our study period spans the last eight years from fall 2010 to spring 2018, facilitating an assessment of recent sea ice conditions in the Arctic Ocean. Thus we do not consider satellite thickness records prior to the launch of CS2, such as from Envisat (e.g. Giles et al., 2008) or ICESat (e.g. Kwok et al., 2009).

**2.1.1 CPOM**

CPOM was first to produce publicly-available estimates of sea ice thickness from CS2 and they provide both near real time (NRT) thickness products for 28, 14, and 2-day observation periods (Tilling et al., 2016), which are updated on a daily basis with a typical latency of 3 days, as well as a monthly and seasonally-averaged thickness data products (Tilling et al., 2018, Table 1). Archived data coverage begins in November 2010, and monthly averages are available on an ad-hoc basis.

Thickness data are available for the months of October through April on a 5 km grid for the full northern hemisphere (regions 1–14, Fig. 1), as well as on a 1 km grid for specific subpolar regions.

CryoSat-2 radar waveforms delivered through the ESA Level 1b product are used to identify lead and ice floe elevations (Tilling et al., 2018). Radar waveforms associated with leads and ice floes are distinguished using fixed criteria for stack

standard deviation and pulse peakiness (Laxon et al., 2013; Tilling et al., 2018). Ice elevation is defined as 70 % of the peak amplitude on the leading edge of the first peak in the radar waveform (Tilling et al., 2018). Lead elevations are determined separately (Tilling et al., 2018) through application of the retracking method developed in Giles et al. (2007) where a Gaussian plus exponential model is fit to each waveform. The CPOM algorithm utilizes the UCL13 mean sea surface (MSS)

for the calculation of sea surface height anomalies, and to reduce the impact of geoid slope on freeboard estimates. This step is especially important in areas of low lead fraction (Skourup et al., 2017). Auxiliary information including the location of the ice edge and a product distinguishing first-year ice (FYI) from multi-year ice (MYI) is needed in the CPOM thickness algorithm. The sea ice edge is defined as the 75 % ice concentration contour in the SSMI/S Daily Polar Gridded Sea Ice

Concentration data set (available from the National Snow and Ice Data Center, NSIDC), while sea ice type is derived from the EUMETSAT's Ocean and Sea Ice Satellite Application Facility (OSI-SAF) product (Tilling et al., 2018). The depth and density of snow on sea ice are based on the snow climatology of Warren et al. (1999) and applied by ice type. On MYI, the monthly mean climatological value is applied, and this value is halved for FYI (Kurtz and Farrell, 2011), meaning constant snow depth and density values for MYI and FYI are applied each month (Tilling et al., 2018). At the time of writing, only

seasonal averages for two seasonal periods (October-November and March-April) were available, and this data version was used as the baseline data set in our analysis. Undefined thickness estimates are indicated by a value of 0.0000 in the CPOM product and these were removed before further processing.

### 2.1.2 AWI

AWI also provides monthly CS2 data products for October through April. Archived data begins in November 2010 and new

monthly data are made available on a variable basis, but typically with one month latency (Table 1). In addition to thickness, AWI offers a number of additional geophysical and instrument parameters in their data product, including sea ice freeboard and concentration. Of the products analysed here, it is the only product that provides uncertainty estimates for thickness and freeboard (Hendricks et al., 2016). All parameters are provided on a 25 km grid for all regions of the Arctic (i.e., regions 1–14).

The AWI product uses the ESA L1b baseline C CryoSat-2 radar waveforms, applying a 50 % Threshold-First-Maximum Retracker (TFMRA, Ricker et al., 2014) to derive elevation for all surface types. In contrast to the CPOM product, AWI currently uses the DTU15 MSS for the calculation of sea surface height anomalies (Hendricks et al., 2016). Skourup et al. (2017) demonstrated that processing with either the DTU15 or UCL13 MSS results in consistent freeboard estimates, with

only small deviations in areas of low lead concentration. For both sea ice concentration and type, AWI uses OSI-SAF products (Hendricks et al., 2016), and the ice edge is defined at the 70 % ice concentration contour. The AWI product applies snow depth based on the climatology of Warren et al. (1999), following the method described in Laxon et al. (2013), hereafter referred to as the "modified Warren climatology" (MWC). In the MWC, snow depth on FYI is 50 % of the climatology, and snow density is a monthly constant per ice type. The AWI product allows mixed ice types at the boundaries

between MYI and FYI, based on confidence levels given in the ice-type product (Ricker et al., 2014). For this analysis AWI product version 2.0 was used and ice concentration, freeboard and thickness were assessed.

### 2.1.3 JPL

The JPL product provides monthly CS2 thickness data from January 2011 to December 2015, on a 25 km grid (Table 1). Coverage is limited geographically to the central Arctic (regions 1–6). The JPL retracker for deriving surface elevation is based on the waveform centroid, rather than a leading-edge approach used in the AWI TFMRA algorithm. The JPL
freeboard algorithm also uses the EGM2008 geoid, rather than a MSS model, which can result in anomalous sea ice thickness estimates in areas of steep ocean topography, such as near the Lomonosov and Gakkel Ridges (Skourup et al., 2017). Information about which sea ice concentration product is applied in the JPL algorithm is not provided in the available literature, nor the ice concentration threshold used to define the ice edge. Kwok and Cunningham (2015) state that the sea ice type is derived from analysed fields of Advanced Scatterometer (ASCAT) data. Snow depth and density in the JPL product
is based on the MWC and applied similarly to Laxon et al. (2013), except that where Laxon et al. (2013) used a 50 % reduction in the climatology over FYI, Kwok and Cunningham (2015) examined both a 50 % and 70 % reduction, it was left unclear which version was selected for the final online product. The data are currently only accessible to the public after registration on a product website. Undefined thickness values corresponding to 9999.0 and -1.0 were removed before processing.

**2.1.4 GSFC**

GSFC provides 30-day ice thickness averages derived from CS2. Coverage begins in October 2010 and new data continue to be made available on a time-varying basis, but typically with a six week latency (Table 1). In addition to ice thickness, the GSFC product includes estimates of freeboard and surface roughness (Kurtz and Harbeck, 2017). All parameters are available on a 25 km Polar Stereographic SSM/I Grid, for regions 1–8.

Unlike other CS2 freeboard algorithms, the GSFC product is derived using a waveform-fitting method. The surface type, elevation, and other properties of the received radar waveform are derived through statistical comparison to analytically pre-computed waveforms, using a least squares error minimization. Kurtz et al. (2014) note that this approach should nominally result in freeboard values lower than those derived from TFMRA-based methods. The GSFC algorithm utilises the DTU10
MSS for freeboard calculation. Skourup et al. (2017) have demonstrated that more recent MSS models (e.g., UCL13, DTU15), which incorporate sea surface height data from CS2, enhance the definition of gravity features, resulting in a more accurate freeboard derivation. Indeed Skourup et al. (2017) found that the DTU10 MSS in particular was not sufficient for freeboard processing due to decimeter-level discontinuities at 81.5°N and 86°N as well as at the ice edge, which resulted in erroneous freeboard measurements at these locations. The GSFC product uses a 70 % ice concentration threshold derived
from the NSIDC Near-Real-Time DMSP SSMI/S Daily Polar Gridded Sea Ice Concentrations available at NSIDC to define the sea ice edge, and the OSI-SAF product is used for sea ice type (Kurtz and Harbeck, 2017). An additional unique attribute of the GSFC algorithm is the use of a single ice density value (915 kg/m3) for all ice types in the thickness derivation step

(Kurtz et al., 2014). In the case of the other CS2-only thickness products a dual ice density approximation is made, with the assumption of a lower density for MYI (882 kg/m3) than for FYI (917 kg/m3). Despite using a single density for both ice types, the GSFC algorithm employs the MWC for snow depth. Here we used the GSFC product version 1.0 ice freeboard and thickness data. Undefined thickness estimates are indicated by a value of -9999 in the GSFC product and these were

removed before further processing.

### 2.1.5 CS2SMOS

The CS2SMOS sea ice thickness product, developed by AWI and the University of Hamburg, is a blended product of thickness estimates from CS2 and SMOS. It provides weekly data for the Arctic northward of 50 $^{°}$N on an EASE2 grid, with 25 km grid resolution, across regions 1–14. It is available for a period starting in November 2010, ending April 2017.

The SMOS mission provides L-band observations of brightness temperature, which may be used to derive ice thickness in areas where thin sea ice exists (Kaleschke et al., 2012). CS2 exploits radar altimetry to measure the difference in height between the snow/ice surface and sea surface, which is then used to derive sea ice thickness through the assumption of hydrostatic equilibrium. Since CS2 was designed to measure ice thicker than 0.5 m, it may be advantageous to blend CS2

estimates with complementary estimates from SMOS. Due to the satellites having different spatial and temporal coverage, optimal interpolation is used to merge the two data sets (Ricker et al., 2017a). The algorithm includes weighting the data based on the known uncertainties of the products and modelled spatial covariances (Ricker et al., 2017a; 2017b). For sea ice concentration and type the OSI-SAF Arctic daily products are used. Snow depth and density follow the MWC. The CS2 product used is the AWI CS2 product with processor version 1.2 (Ricker et al., 2014; Hendricks et al. 2016) and the SMOS

thicknesses are from the University of Hamburg processor version 3.1 (Tian-Kunze et al., 2014; Kaleschke et al., 2016). For our analysis thickness data from CS2SMOS product version 1.3 were used.

### 2.1.6 APP-x

The NOAA extended AVHRR Polar Pathfinder (APP-x) Thematic Climate Data Record (CDR) provides sea ice thickness estimates, along with 18 other geophysical variables, in a climate data record (Key and Wang, 2015). Thickness estimates

are available for both the Arctic Ocean (regions 1-14) and Southern Ocean, spanning 1982 to present. Data are provided twice daily, with a typical latency of approximately 4 days (Table 1). In contrast to the other satellite-derived thickness products, year-round thickness estimates are available, including throughout the summer, and are provided on a 25 km grid. We note a gap in the thickness record at the time of writing for the period 8 March to 1 May 2017 and hence assessment of APP-x in spring 2017 was not possible.

Sea ice thickness estimates are derived from AVHRR satellite radiometer measurements using a one-dimensional thermodynamic model (OTIM). The OTIM model derives sea ice thickness as a function of surface heat fluxes, surface

albedo and radiation, which all contribute to surface energy budget (Wang et al., 2010). Furthermore, most of the flux and radiation parameters in the equations are functions of surface skin and air temperatures, surface air pressure, surface air relative humidity, ice temperature, wind speed, cloud amount and snow depth, which are input parameters in the model (Wang et al., 2010). The sea ice concentration source for the APP-x product is Nimbus-7 SMMR and DMSP SSM/I data

processed with the NASA Team Algorithm (Key and Wang, 2015) and the ice edge is defined at the 15 % ice concentration contour. The sea ice type is converted from the reflectances measured directly by AVHRR. Additionally the following input is needed for ice thickness: percentage cloud cover, surface skin temperature, surface broadband albedo and surface shortwave radiation fluxes, of which the latter two are obtained for daytime retrievals only. For the APP-x product the snow depth estimates are based on the snow depth climatology of Warren et al. (1999) but combined with field observations

through experimentation and applied using monthly look-up tables (Wang, pers. comm., 2018). For our analysis thickness data from APP-x CDR version 2.0 were used, and undefined ice thickness values identified as "9.96920996839e+36" were removed before processing.

Since sea ice dynamics are not included in the OTIM model, the thickness errors in the APP-x product are larger where the

ice surface is not smooth, i.e. in regions with pressure ridges, hummocks and melt ponds (Wang et al., 2010). Generally OTIM tends to overestimate ice thickness, in particular for thin ice while underestimating thick ice, as the energy budget approach is less sensitive for thick ice (Wang et al., 2010). Moreover since the satellite sensor retrieves 2 m air temperature, ice surface temperature is derived from the 2 m measurement. Wang et al. (2010) state that the thickness estimates are more accurate for nighttime retrievals, when 2 m air temperature and ice surface temperature are closer, resulting in a smaller

model error. For this reason, we utilise the nighttime estimates in our study. Wang et al. (2010) note that errors due to uncertainties in snow depth and cloud fraction are the primary sources of error in the OTIM thickness estimates. We also note that OTIM is applicable to other optical satellite data including observations from NASA's Moderate Resolution Imaging Spectroradiometer (MODIS) instrument and EUMETSAT's Spinning Enhanced Visible and Infrared Imager (SEVIRI) instrument (Wang et al., 2010).

**2.2 Evaluation Data Sets**

We evaluate the satellite-derived ice thickness estimates using independent upward looking sonar (ULS) observations of ice draft from Beaufort Gyre Exploration Project (BGEP) moorings, and airborne observations of ice thickness from Operation IceBridge (OIB). These data sets represent the most extensive and sustained record over the evaluation period, compared to many of the other publicly-available in situ thickness data sets.

**2.2.1 Beaufort Gyre Exploration Project**

Since August 2003, BGEP has operated a series of moorings in the Beaufort Sea, which have included an ULS instrument. From 2003 to 2014 the ULS instrument produced a range estimate every two seconds, increasing in frequency to once per

second starting with the 2014–2015 deployment. By subtracting the ULS range estimate from instrument depth, draft is measured to an accuracy of +/- 0.05 m per individual measurement (Krishfield et al., 2006).

Here we utilize ULS draft measurements from three mooring locations (A, B, and D, Fig. 1) in the Beaufort Sea over the six-month period spanning November–April, for all years from 2010-2017. Since the ULS measures ice draft as floes drift across the mooring location, the data represent a high-resolution, time-varying measurement of many individual leads and ice floes, thus providing a more complete picture of the regional ice thickness distribution. The draft to thickness ratio is approximately 0.9 (e.g., Rothrock et al., 2008), but to accurately compute thickness from draft, knowledge of ice type, ice density and snow loading are required. Here we do not convert draft to thickness, since that would introduce additional uncertainties. Rather we use the characteristics of the ice draft distribution to evaluate the satellite-derived thickness distribution.

### 2.2.2 Operation IceBridge

OIB was launched in 2009, and is a sustained airborne mission designed to continue the collection of sea ice and land ice elevation measurements in the temporal gap between the end of the ICESat mission in 2009 and the launch of ICESat-2 in 2018 (Koenig et al., 2010). The mission includes an altimeter, the Airborne Topographic Mapper (ATM), which provides high-resolution measurements of sea ice plus snow freeboard, and a snow radar instrument for derivation of snow depth (Newman et al., 2014). Together these allow for estimation of sea ice thickness (e.g., Farrell et al., 2012; Kurtz et al., 2013). Thickness uncertainty, calculated by propagation of estimated errors in the contributing variables, associated with IceBridge estimates is approximately 0.66 m (Richter-Menge and Farrell, 2013). Here we make use of the IDCSI4 (spring 2011) and NSIDC-0708 (spring 2012–2017) IceBridge thickness products available at the NSIDC (Kurtz et al., 2015; Kurtz, 2016). Due to the geographical layout of airborne flight surveys, the majority of IceBridge measurements sample multi-year sea ice in the Canada Basin (e.g., Richter-Menge and Farrell, 2013).

### 3 Methodology

### 3.1 Satellite Product Intercomparison

Seasonally-averaged ice thickness is computed for each product over two periods: fall (October and November), and spring (March and April). Seasonal averages are calculated by taking the arithmetic mean value of all available thickness estimates across the Arctic region (Fig. 1) within the two month period. Results for fall and spring are shown for each product over the period 2011 - 2017 in Figures 2a and 2b, respectively.

To evaluate spatial variations in thickness within products, mean regional ice thickness is computed for the central Arctic Ocean and each peripheral sea (Table 2) using the polygons shown in Figure 1, for both the fall and spring seasons. Thickness values are reported for any region containing valid data points in the relevant months for at least one year of the product record. Although all regions with data are reported, regions outside the main product coverage areas defined in Table 1 may contain only a few data points and hence the reported regional ice thickness for some of the peripheral seas may represent only a small portion of the region as a whole (see Fig. 2).

To assess temporal variations across products, we calculate a baseline mean ice thickness for the central Arctic for the period common to all products (2011-2015), for both the fall and spring seasons. Subject to the time span over which each product is available (Table 1) we compute the anomaly with respect to the baseline mean for each season and report these results in Table 3. We also assess trends in winter ice growth for each product by calculating monthly mean ice thickness in the central Arctic (regions 1-6) during the period of study spanning 2010 to 2018 (Figure 5).

For point-to-point comparisons of the satellite data products, and to compute correlations across products, all ice thickness datasets are placed on a common 0.4° latitude by 4° longitude grid. We define the CPOM seasonally-averaged thickness product as the reference dataset against which the other products are compared. This approach is justified since the CPOM product was the first publicly-available CryoSat-2 thickness dataset (Laxon et al., 2013), and has been widely used in studies by end users (e.g. Allard et al., 2018; Blockley and Peterson, 2018; Stroeve et al., 2018). We therefore compute product differences as follows:

$$\Delta SIT = SIT_P - SIT_C ,\tag{1}$$

where $SIT_c$ is the CPOM sea ice thickness, $SIT_p$ is the thickness of the product in question, and $\boldsymbol{\Delta}SIT$ is the difference between the two. Correlation statistics are calculated utilising grid cells in which both data sets contain thickness estimates. These grid cell pairs are used to compute the Pearson correlation coefficient (r), the product difference across common cells according to Eq. 1, and the standard deviation of this difference (Figure 4).

## 3.2 Satellite Product Evaluation

Since the BGEP moorings are tethered to the seafloor, ULS measurements are representative of discrete ice floes drifting over the mooring location. To facilitate comparison with the satellite data, we select all product data points within 200 km of the mooring, following the approach described in Laxon et al. (2013), creating a comparison region centred on the mooring location. In order to avoid influence from areas outside the comparison region, we use the original satellite thickness data as provided, rather than the gridded data described in Section 3.1 above.

Both the ULS draft data and satellite thickness data within the comparison region are averaged over one month intervals from November to April for the period of overlap between the product record and ULS data spanning fall 2010 to spring 2017. The correlation coefficient for ULS draft measurements and product thickness observations is calculated at each mooring based on these paired monthly data points (Figure 8). Any individual ULS draft measurement thinner than 0.1 m is not included in the averaging, as these measurements may represent leads rather than ice floes (Krishfield et al., 2006; Krishfield et al., 2014). In the case of the CPOM product, correlations with the ULS ice draft are based on seasonal averages, processed in a similar manner to the monthly averages as described above.

The sea ice thickness data from Operation IceBridge consists of flightlines across the Canada Basin and the central Arctic Ocean (e.g. Richter-Menge and Farrell, 2013). Following Tilling et al. (2018), data acquired during IceBridge spring campaigns in 2011-2017 were placed on a 0.4° latitude by 4° longitude grid and compared against the satellite thickness data for March–April, for all six products at common grid cell locations. This allows the calculation of thickness differences between each product and IceBridge, as follows:

$$\Delta SIT_{eval} = SIT_p - SIT_{oib} \ , \tag{2}$$

where $SIT_{oib}$ is the Operation IceBridge sea ice thickness, $SIT_p$ is the thickness of the product in question, and $\boldsymbol{\Delta} SIT_{eval}$ is the difference between the two. Using the gridded data, we also calculate correlation coefficients, and standard deviation between each product and the IceBridge thickness observations (Figure 9).

## 4 Results

Here we present, for the first time, a side-by-side comparison of a suite of available CS2-only products, alongside a blended CS2-SMOS (CS2SMOS) product and one altimetry-independent sea ice thickness product, APP-x. The results are presented in three parts. First we provide a review of Arctic sea ice thickness variability during the last eight years. Next we compare regional and temporal differences between the satellite products across the Arctic regions. Finally we evaluate the satellite-derived thicknesses through comparisons with independent measurements.

### 4.1 State of the Arctic Sea Ice Thickness

Seasonal ice thickness for fall and spring is shown in Figure 2 for the period 2011–2017. Following the observed low summer sea ice minimum extents in 2011 and 2012 (Parkinson and Comiso, 2013), we find that the lowest ice thickness was recorded in fall 2011 and 2012 (Fig. 2a). Mean ice thickness was 1.12 m (1.19 for the CS2-only products) in fall 2011, with variations of +0.3 to -0.23 m (+0.24 to -0.11 for CS2-only) among the products. For fall 2012 the mean thickness was 1.18 m (1.19 for CS2-only), varying from +0.34 to -0.36 m (+0.24 and -0.11 for CS2-only) among products. The loss of multi-year

ice in the summer of 2012 due to the record sea ice minimum resulted in an overall thinner ice cover during the following winter/spring (i.e. spring 2013) when the mean ice thickness was 2.05 m (2.0 m for CS2-only) among products (Fig. 2b). Consistent with the results first noted in Tilling et al. (2015), we see that following a cool summer in 2013, survival of ice through the melt season resulted in a rebound in thickness in fall 2013 with mean thickness of 1.46 m (1.58 for CS2-only), and a thicker winter mean sea ice thickness of 2.34 m in spring 2014 (all products), which has persisted in the central Arctic for subsequent seasons (Fig. 2). Tilling et al. (2015) also noted a slight recovery in 2013–2014 following low winter-time ice thickness in 2011 and 2012. The thickest ice at the end of winter was observed in 2014 (Fig. 2b) in a region stretching from northern Greenland near Kap Morris Jesup, to Banks Island in the Canadian Arctic Archipelago. This region of thick ice in the central Arctic has persisted throughout the following seasons. As of spring 2018 (not shown) the area of ice more than 3 meters thick adjacent to the northern coasts of Greenland and the Canadian Arctic Archipelago is still greater than that of spring 2012 and 2013, although the spatial extent of this thick ice area has diminished since 2014. In multiple springs (2012, 2014, 2015, 2016) an outflow of thick ice extends from the southern Canada Basin into the southern Beaufort Sea, due to the dynamic action of the Beaufort Gyre circulation (Fig. 2b). While this band of thick sea ice is captured in all of the CS2 products, it is less distinct in the CS2SMOS product, and does not appear in the APP-x product, apart from spring 2012 (Fig. 2b). This overall picture of the state of Arctic sea ice thickness over the last several years is consistent across the CS2-only products and the CS2SMOS blended product. Although the APP-x product captures the spatial gradient in thick to thin ice, from the northern coasts of Greenland and Canada to the Siberian coastline, respectively, the product does not resolve many of the recent major changes in sea ice thickness conditions in either the spring or fall.

**4.2 Regional Differences**

The satellite-derived thickness products differ in their regional coverage and the availability of thickness estimates across the northern hemisphere. APP-x has the most widespread coverage, although CPOM, AWI and CS2SMOS all provide thickness estimates in the sub-polar seas. The JPL product only provides estimates for an area approximately contiguous with regions 1–6, while the GSFC product provides estimates for an area approximated by regions 1–8 (Table 1, Fig. 2). Three of the six products (AWI, CS2SMOS, and APP-x) consistently resolve thin ice (<= 0.5 m) at the periphery of the ice pack (regions 3–7) during the fall, but only CS2SMOS and APP-x resolve thin ice in these regions in spring (Fig. 2b). In addition APP-x does not resolve the thicker ice of the central Arctic Ocean, as evident in the other products, especially in the fall.

Maps of differences in ice thickness across products, as defined in Eq. 1, are shown in Fig. 3, and distributions of the differences between products are provided in Fig. S1. Average seasonal ice thickness for each region of the Arctic (as defined in Fig. 1) over the available product record is provided in Table 2. In general, differences in ice thickness are larger in the fall than spring, although for spring even across the CS2-only products, differences range from 0.25 m to 2.12 m in regions 6-14 (Table 2). The closest agreement across products is found between the CPOM, AWI and JPL products (Fig. 3, 4, Fig. S1). The thickness estimates of these products have a correlation of 0.91-0.92 in the spring and 0.88 in the fall, with a

mean difference of 0.03 - 0.08 m (Fig. 4). However, there are noticeable regional differences in mean thickness during the observation period (Table 2), for example in spring in the Greenland Sea the AWI product is 0.32 m thicker than the CPOM product, while in the Canadian Archipelago the CPOM product is 0.37 m thicker. There are also differences in the spatial gradients of ice thickness (Fig. 3), particularly in the central Arctic Ocean. For example the JPL product estimates thicker ice

close to Greenland, and thinner ice near the North Pole, and along the Siberian shelf zone than the CPOM product (see also Table 2). Of the CS2-only data products, Figs. 3 and S1 demonstrate that the GSFC product is the most dissimilar to the CPOM data, with thickness in both the fall and spring periods being higher on average, though with year-to-year spatial variation. Mean ice thickness differences range from 0.02 to 0.25 m (Fig. 4), though despite these differences, the GSFC product is still highly correlated with the CPOM product (R = 0.85 in both spring and fall).

From a regional perspective, the GSFC product agrees closely with the CPOM product in spring (especially in regions 1-7), but larger thickness differences are recorded in fall across all regions (Table 2). The CS2SMOS product suggests thinner ice thicknesses than the CS2-only products, with mean differences of 0.08 to 0.36 m (Fig. 4) and differences increasing towards the ice edge. For example in the Barents Sea, average spring thickness for the period 2011–2017 was only 0.41 m, exactly 1

m lower than the CPOM product estimate (Table 2). Only in the MYI zone in spring does the CS2SMOS product provide estimates of ice thicker than the CPOM product in some years (Fig. 3b). Despite its lower thickness estimates, the CS2SMOS product correlates well with CPOM, 0.86 and 0.88 in the fall and spring, respectively (Fig. 4). We also note that in fall 2017 the APP-x product shows thickness data covering an area south of the typical ice edge in regions 8–10 (Fig. 2a). Anomalous data in this region may be due to errors in the sea ice concentration field (not shown). Ice concentration is passed

to the APP-x product from the lower-level APP product. In spring ice in the central Arctic in the APP-x product is consistently thicker over FYI and thinner across both the MYI zone and the thick outflow along the northern coast of Greenland into the Canada Basin (Fig. 3b, Table 2). In the fall, the APP-x product contains mainly thinner ice for all regions (Table 2), except in 2011 in the Canada Basin, where the APP-x product suggests thicker ice than in the CPOM product, and in 2017 around the ice edge in regions 3–6 (Fig. 3a). Also, in fall, the APP-x product estimates thicker ice for regions 3-7,

and 11, compared to the blended CS2SMOS product (Table 2). With correlations of 0.49 and 0.53 in the spring and fall, respectively, the APP-x thickness data do not correlate as well with the CPOM product as the other data sources (Fig. 4).

### 4.3 Differences in Winter-time Growth Rates

We now consider the winter-time growth rates across the central Arctic. The evolution of monthly mean ice thickness during winter is shown in Figure 5 for the entire study period spanning fall 2010 to spring 2018, and growth rates are provided in

Table 5. The results are dependent on the product availability (Table 1), and in the case of the CPOM product, only seasonal means are assessed. Monthly mean ice thickness (Fig. 5) in the central Arctic can differ by up to 1.2 m across products. As we might expect, the CPOM, AWI and JPL products are the most similar in terms of both the monthly mean trends in ice thickness and growth rates (Fig. 5, Table 5). While the maximum difference between the CPOM and AWI products is 0.1 m,

the JPL product differs most noticeably from the AWI product by ~ 0.14 m in October - December 2013 (Fig. 5). Ice in the GSFC product is consistently thicker than the other CS2-only products, with ice thickness estimates beginning each season by up to 0.4 m thicker in fall, before converging towards the other CS2-only products by the end of the winter (Fig. 5). In November 2010, January 2016, November 2016, and April 2018 the GSFC product indicates a small (maximum 0.1 m) decrease in thickness, whereas CPOM and AWI have a constant upward trend during the ice growth period (Fig. 5). The daily ice growth rates are consistently lowest for the GSFC product (Table 5). Interestingly, for some early seasons, e.g. fall 2011, CS2SMOS estimates slightly higher mean sea ice thickness than the AWI product (Fig. 5). CS2SMOS has daily growth rates higher than those of GSFC, but lower than other CS2-only products, differing from the AWI product by a maximum of 0.001 m/day (Table 5). APP-x has the highest growth rate (i.e. the smallest minimum and largest maximum thickness) exceeding the CS2-only products, and the ice cover can gain 2 m within an ice growth season and up to 0.0108 m/day. The strongest increase in the APP-x mean thickness takes place at the end of winter between February and April, when the ice grows by up to 1 m (Fig. 5). This differs significantly from the winter-time evolution of the ice as shown in the CS2 products, which suggest very little growth in ice thickness at the end of winter (March–April), likely due to the insulating properties of the overlying snow cover. There is agreement across all products containing CryoSat-2 data that the largest daily growth rates occurred in winter 2011-2012 (Table 5). Although the GSFC product has the lowest daily growth rate, it has the highest interannual variability in growth rate (0.0026 m/day), whereas the JPL product has the smallest (0.0005 m/day).

The year-to-year seasonal trends in central Arctic ice thickness (Fig. 6) are very coherent among the CS2-only products, with an increase in mean ice thickness in the fall between 2011 and 2014, followed by a slight decrease and leveling off (Fig. 6a). The GSFC product shows a similar year-to-year trend in fall to the other CS2-only products, but is 0.3-0.5 m thicker on average (Fig. 6a). CS2SMOS follows the CS2-only products with maximum difference of -0.06 m compared with the AWI product and almost exactly the same mean ice thickness as the JPL product in fall 2011 and 2012 (Fig. 6a). The fall averages in the APP-x product are lower than for the other products by 0.2-0.6 m.and the trend in seasonal mean thickness of the APP-x product does not follow those of the CS2 products, in particular for the 2012 and 2016 seasons, where the APP-x product hast an opposite trend (Fig. 6a).

In spring the central Arctic mean ice thickness differs little across the CS2-only products (Fig. 6b), with the strongest similarities again between the CPOM, AWI and JPL products. The CS2 products show a drop in thickness in 2012 continuing to 2013, followed by a slight recovery in 2014, preceding another drop in mean thickness in spring 2015, which has persisted since then. The exception to the similar direction of CS2 products is in 2017 when the GSFC product suggests an increase in thickness, and is 0.2 m higher than the other CS2-only products. By spring 2018 ice thickness in the GSFC product has decreased and is once again in line with the AWI estimate, though slightly lower than the CPOM estimate (Fig. 6b). CS2SMOS product is very similar to the other CS2 products, the trend line being consistently approximately 0.2 m

lower than the AWI one. APP-x estimates are higher than for the other products in spring with an almost constant mean thickness of ~2.5–2.6 m, and very little year-to-year variability (Fig. 6b).

We have calculated the annual deviations from the mean thickness across the central Arctic for the baseline period 2011-
2015, and present the results in Table 3. The departure from the baseline mean thickness in spring for the CS2-only data products was 0.17 m in spring 2013 (0.12 m in 2012), and 0.17 m thicker in 2014, in line with the thickness increase shown in Figure 6b. The CS2-only products are similar in the direction of annual departures from the baseline mean except for GSFC in spring 2017, which shows a positive departure (0.05 m), whereas the CPOM and AWI product thicknesses are lower than their baseline means (0.1 m and 0.08 m, respectively). A very similar pattern of departures can be seen for fall
CS2-only thickness data, where there is a mean departure of 0.21 m in fall 2011, 0.13 m in 2012 and thickening of 0.18 m in fall 2013. The CS2SMOS deviations differ very little from the CS2-only product annual deviations, and are actually more in line with them in fall 2015 and 2016 compared to the GSFC product. APP-x shows no noticeable deviation from its fall baseline mean, as previously observed in Figure 6b, although there is clearly more deviation in the fall compared to the spring. The changes in APP-x annual deviations follow those of the CS2 products in falls 2012-2015, as well as 2017, except
for 2016, when it deviates by -0.21 m (see also Figure 6a). The low ice thickness records in fall 2011 and 2012 (Fig. 2a) are also evident in the results shown in Table 3.

## 4.4 Comparison against Independent Observations

Next we consider the satellite-derived sea ice thickness products in the context of independent measurements from ULS and IceBridge to evaluate the utility of the satellite products for providing information on the full thickness distribution. As
mentioned previously, the draft to thickness ratio is approximately 0.9 (Rothrock et al., 2008). Therefore we do not expect the modal thickness and draft to be equivalent, but we do expect the distributions to have the same characteristic shapes.

Histograms of the draft/thickness distributions are shown in Figure 7, and suggest that none of the satellite products capture either the thickest or thinnest ice, although the CPOM, AWI and CS2SMOS products do have some observations of ice
thickness below 0.5 m, with CS2SMOS performing best in this regard. This general result may be due to the fact that the monthly/seasonal satellite-derived products have been provided at 25 km resolution (Table 1), thus potentially averaging out the thinnest and thickest satellite observations per grid cell. Modal ice draft is 1.3 m while modal ice thickness ranges 1.7 m to 2.3 m (Fig. 7). Comparing the characteristics of the draft/thickness distributions, including modal values and distribution width, the CS2SMOS, CPOM and AWI data sets most closely align with the ULS data. The JPL product distribution reveals
slightly thicker ice than the CPOM and AWI products. The GSFC product has a bimodal thickness distribution for the study period, with modes at 1.7 and 2.3 m, and the APP-x product also has a modal ice thickness of 2.3 m.

The correlations between the monthly-averaged, satellite-derived ice thickness and ULS ice draft are shown in Figure 8 for the months of November through April for each of the three BGEP moorings. The results are consistent with the approximation of a ~0.9 ice draft to thickness ratio (Rothrock et al., 2008). The correlations were calculated between satellite and ULS monthly averages, which were combined across years 2010-2017 to aid visualization (e.g. November 2010–2017,

December 2010–2017, etc.) with the following exceptions: JPL product data were assessed over the period 2011-2015, spring 2017 data were not available for the APP-x comparison, and seasonal averages were evaluated for the CPOM product. Correlation results for the CPOM, AWI and JPL products are strongest, followed by APP-x, GSFC and CS2SMOS (Fig. 8), but all satellite thickness products display very strong correlations with the ULS draft data, in line with previous results from Laxon et al., (2013), Kwok and Cunningham (2015), and Tilling et al., (2018). The exception to this is for ice drafts > 1.3 m,

when there is an observable divergence in the results for the CS2SMOS and APP-x products. The APP-x thickness estimates are ~ 0.4 m higher than the CS2-only products, while the CS2SMOS data are 0.1-0.3 m thinner, suggesting a thickness overestimation in APP-x and an underestimation in CS2SOS, with respect to the thickest ice. The yearly means for ULS ice draft and satellite-derived ice thickness are provided in Table 4. The results suggest the mean ice thickness was largest in 2014 and 2015, and lowest in 2011 and 2012 (Table 4). In years with lower ice thickness, there is some disagreement

between the ULS observations and the satellite products. 2013 appears to have the thinnest ULS ice draft results for the observation period, whereas the CPOM product suggests that this was one of the thickest years, while other CS2 products appear to have low ice thickness in 2013 only for buoy D.

Next, the satellite-derived thickness estimates for spring are compared with seven years of independent OIB thickness data

(Fig. 9). The JPL product has the highest correlation (r=0.76) with the OIB data and we note that this is higher than the correlation value of 0.53 noted in Kwok and Cunningham (2015), who only considered data for March and April 2011–2012. Likewise, our correlation value of 0.70 between CPOM and OIB thickness is slightly better than the results shown in Laxon et al. (2013) and Tilling et al. (2018), who found values of 0.61 and 0.67 when assessing CS2 against the 2011-2012 and 2011-2014 OIB campaign data, respectively. Similar to the results observed with the ULS comparisons, the satellite products

seem to be missing the thickest ice seen by the OIB measurements, but overall the agreement between the CS2 products and OIB is good. In terms of absolute differences, JPL thickness estimates are slightly higher on average than those of OIB by 0.10 m, whereas CPOM and AWI estimates are lower by 0.11 m, GSFC by 0.05 m and CS2SMOS by 0.22 m (Fig. 9). Although CS2SMOS differs twice as much from the OIB estimates compared to AWI and CPOM, for ice thickness between 0 m and 1.5 m the CS2SMOS estimates appear to agree best with OIB. Thus the greater difference would be explained by

the thicker ice, where CS2SMOS estimates are lower than OIB. These findings for CS2SMOS are in line with previous validation studies (Ricker et al., 2017b) who evaluated CS2SMOS using observations from an airborne electromagnetic (AEM) induction thickness sounding device. APP-x has the smallest correlation of 0.54 and a peculiar vertical concentration of data in the scatterplot (Fig. 9), where a majority of APP-x sea ice thickness estimates fall into a thickness category between 2.25 and 2.5 m.

## 4.5 Ice Freeboard

Recall (from Section 2) that only two products provide freeboard estimates: AWI and GSFC. Figure 10 compares sea ice freeboard across these two products for spring and fall. Our assessment reveals a prevalence of negative freeboard estimates in the GSFC product that do not appear in the AWI product (Fig. 10). For the period October 2010 to April 2018, an average of 29.5 % of the freeboard measurements provided in the GSFC data product are negative, in contrast to 0.9 % of the freeboard measurements in the AWI product. Negative freeboard estimates in the GSFC product often correspond with significantly higher freeboards in the AWI product for the same grid cell locations. An example of the GSFC freeboard product for April 2014 is shown in Figure 10c and highlights the spatial prevalence of anomalous, negative freeboard estimates, especially in the Kara and Barents Seas, where negative values persist throughout the winter. However, we note that negative freeboard estimates also occur in the Beaufort, Chukchi and Greenland Seas. This suggests that some negative estimates may be related to regional masking in the processing algorithm, or they could be due to the use of the DTU10 MSS in the ice thickness derivation (e.g. see Fig. 3 in Skourup et al., 2017).

Despite the high proportion of negative freeboard values in the GSFC product, it does not contain any negative thickness values. While GSFC freeboard observations are on average 0.08 m and 0.14 m thinner than corresponding observations from AWI in fall and spring, respectively (Fig. 10), the GSFC product has higher sea ice thickness for both seasons, and almost all regions, compared to the AWI product, as shown in Figures 2 and 3, and Table 2. Closer examination of individual data points in the GSFC product indicates that ice thickness in grid cells containing negative freeboard values is not significantly lower than adjacent data points, suggesting that a filter may be applied to remove negative freeboard values before calculating ice thickness, and/or that ice thickness values are derived from interpolations across many grid cells.

We furthermore note that the AWI and CPOM data products are the only two data products that include negative sea ice thickness estimates. Approximately 0.8 % and 0.2 % of the thickness estimates are negative in the CPOM and AWI data products, respectively. The locations of negative ice thickness estimates for the month of April 2014 are shown in Figure S2, and are representative of the general pattern observed in other years of the study period. Negative data points are found along the ice edge (as defined by the ice concentration threshold of 70 % in the OSI-SAF product, plotted from the variable included in the AWI product, Table 1), suggesting that these thickness values are anomalous and are a result of edge effects in the sea surface height interpolation scheme.

## 5 Discussion

We expect thickness estimates across the CS2-only products to be similar, since their primary differences are due to the algorithmic approach and some of the auxiliary data inputs. These expectations are borne out as can be seen in the basin-scale maps shown in Figs. 2 and 3. The CPOM and AWI products differ very little, while the JPL thickness estimates are

also generally in close agreement. The GSFC product is among the products with thickest sea ice overall, particularly at the beginning of the growth season (Figs. 4, 5, 8), despite containing a very high percentage of negative freeboard values (Fig. 10). On the other hand, due to the inclusion of SMOS data, the CS2SMOS product is weighted for thinner ice, such that we expect overall thickness in this data product to be lower than in the CS2-only datasets(Fig. 2, Fig. 3), with a more realistic

representation of areas with thin ice in the peripheral seas (regions 3-10). Since the APP-x product relies on a thermodynamical model to derive thickness, Wang et al. (2010, 2016) state that the product is expected to perform best over level ice. We find that in fall, APP-x indeed has similar ice thickness to the CS2SMOS product, except over the thickest MYI in the central Arctic (Fig. 2a, Fig. 3a). In spring, however, APP-x appears to overestimate ice thickness across the entire Arctic Ocean (Fig. 2b, Fig. 3b). Despite similarities, there are also major regional differences, as seen in Figs. 2 and 3, and

Table 2. Even among CS2-only products, ice thickness in the peripheral seas (regions 3-10) on average varies by 0.33 m in the fall and by 0.6 m in spring.

We have shown that the CS2-only satellite data products include reliable estimates for sea ice between ~0.5–4 m thick, depending on the product. In general, all satellite products capture a realistic winter-time ice thickness distribution, when

compared to independent ice draft measurements, as demonstrated in Figure 7, with the exception of APP-x, which overestimates the sea ice thickness and underestimates the thickness variation in the Beaufort Sea. Ice in the Beaufort Sea is characterised by mixed amounts of deformed sea ice, which may partially explain the poor results from the APP-x product in this area. The CS2SMOS product has the best representation of the thin ice thickness, which would be expected as SMOS-based estimates are used in the majority of the thin ice regions (Ricker et al., 2017). However our results also show that no

product adequately captures the thinnest sea ice in the thickness distribution at the end of winter (Fig. 7). The CS2-only products do not resolve the thickness of sea ice less than 0.5 m thick, equivalent to freeboards of less than approximately 0.05 m (Figs. 4, 7, 9, 10), and the CS2SMOS product is the most reliable product in this regard (Fig. 7, Fig. 9). Ricker et al. (2014) found that uncertainties in sea ice thickness estimates are large for CS2 in areas where ice is less than 1 m thick. In contrast, the sensitivity is lost for SMOS when the ice is thicker than 1 m. While we have demonstrated that CS2-only

products provide good results in the central Arctic ice pack, they lack robust estimates in some regions, particularly around the ice margins (regions 10-12), and in areas of new ice formation, where thin ice is expected. CS2SMOS, and APP-x to some extent, perform better in the peripheral seas.

The biggest difference in the temporal trend among the thickness products occurs in spring 2012 and 2013, when ice

thickness was at its lowest during the observation period (Table 3, Fig. 5). The winter growth rate among the CS2-only products is very consistent, with the exception of the GSFC product (Fig. 5, Table 5). The GSFC product also shows a decrease in mean thickness between March and April in both 2016 and 2018, while all other products show an increase in thickness during the same periods. The CS2SMOS growth rate for March to April is lower than for the other products, resulting in a mean ice thickness that is ~0.2 m lower than the CS2-only products. The decrease of the CS2SMOS growth

rates could be due to the areas of less than 100 % ice concentration contributing to the growth in others, falling within the SMOS weighted area of the product. As stated by Ricker et al. (2017) and Tian-Kunze et al. (2014), SMOS assumes 100 % ice concentration in the thickness retrieval algorithm, which could cause underestimation of ice thickness in areas with lower concentration. The APP-x product did not resolve notable year-to-year variability in mean thickness. During the ice growth

season the APP-x product shows the largest magnitude of ice growth, over 1.5 m between fall and spring (Fig. 6), with very little inter-annual variability (Fig. 2, Fig. 5). The largest growth in the APP-x product occurs between January and March, at the end of winter (Fig. 5), when in situ measurements typically show inhibited ice growth due to the insulating effects of the overlying snow cover. This suggests that thermodynamic assumptions in the OTIM algorithm for the end of winter may need further refinement.

Finally, we note that all of the satellite-derived products depend on additional auxiliary data sets (Table 1) in the derivation of ice thickness. As we have outlined in Table 1, there is great variation in the source of the auxiliary products and how they are used, particularly the mean sea surface model, ice type delineation and ice concentration. Detailed comparison of the auxiliary products is outside the scope of this study, but these could give rise to differences across products, in addition to

the algorithmic differences. There is also a large range in the ice concentration threshold used to indicate the ice edge across products, varying from 15 % (in the APP-x and CS2SMOS products) to 75 % (in the CPOM product). With regards to the APP-x product we believe that an erroneous ice concentration threshold could be one possible explanation for the peculiar extent of the ice thickness estimates in fall 2017 (Fig. 2a). Additional differences across products may arise due to the treatment of ice density and snow depth on ice. Even though all of the satellite-derived thickness products assessed here

make use of the MWC, they vary in their implementation method, as described in Section 2.1 and none of the products resolve year-to-year variations in snow depth. Although there are multiple approaches proposed to obtain seasonal snow depth estimates on sea ice, they have yet to be routinely incorporated into a publicly-available, satellite-derived thickness data product. Potential solutions include utilizing model simulations (e.g., Blanchard-Wrigglesworth et al., 2015), atmospheric reanalysis data (e.g., Blanchard-Wrigglesworth et al., 2018), extrapolating in situ observations (e.g. Shalina and

Sandven, 2018), or widely expanding the spatial and temporal coverage of current airborne measurement techniques (e.g. Kurtz and Farrell, 2011; King et al., 2015). Satellite passive microwave radiometer observations have also been used to derive snow on first-year sea ice (Brucker and Markus, 2013), as well as snow on thick ice (Maaß et al., 2013; Rostosky et al., 2018). One additional promising remote-sensing method is to combine two satellite altimeter observations retrieved at different wavelengths, enabling snow retrieval due to differences in penetration (Shepherd et al., 2018). For example this

could be achieved through a combination of dual-band radar freeboard observations (e.g. Armitage and Ridout, 2015; Guerreiro et al., 2016; Lawrence et al., 2018) or by comparing freeboard measurements from laser and radar altimeters, to obtain an estimate of year-to-year changes in snow depth (e.g., Kwok and Markus, 2017).

**6 Conclusions and Future Outlook**

Satellite techniques have revolutionized our ability to measure the thickness of ice in the Arctic Ocean, providing critical information for scientists conducting studies of environmental change in the region, as well as a new source of data for forecasters, modelers, operators, and decision makers. Here we assessed a suite of existing satellite-derived, publicly-available, Arctic sea ice thickness data products, conducting a comprehensive examination of regional and seasonal differences over an eight-year period. As expected, the CS2-only products were similar, particularly at the end of winter (Fig. 6b). In April 2011-2016, APP-x reached a mean Arctic-wide thickness of ~2.6-2.7 m, which was thicker than any other satellite product (Fig. 5), and showed little to no inter-annual variability (Fig. 6). On the other hand, likely due to its inclusion of thin sea ice thickness, derived from passive microwave radiometer data, the CS2SMOS data product is on average 0.2 m thinner than the CS2-only estimates at the end of the winter growth season. In fall, there was a larger spread in mean thickness across the products, and the GSFC thickness product diverged from the three other CS2-only products by approximately ~0.3-0.4 m (Fig. 6a). Evaluation of the satellite data products through comparisons with OIB and ULS measurements revealed that all products were well correlated with the independent ice draft/thickness estimates, with correlations of 0.54 and higher (Figs. 8 and 9). Five of the six products resolved an accurate winter-time sea ice thickness distribution for the Beaufort Sea when compared with ULS observations of ice draft, with the AWI and CS2SMOS data sets producing the most robust results (Fig. 7). The APP-x data product did not resolve ice thickness variability in this region, and was biased thick compared to both the ULS ice draft observations and the alternative satellite thickness products. However in fall, APP-x sea ice thickness estimates in the peripheral seas (regions 3-10) resembled those of CS2SMOS (Fig. 2a). Our study revealed some other remarkable differences across the products utilizing CS2 data: there were occasional reductions in mean ice thickness during the winter growth season in the GSFC product, and it diverged from the other CS2 products in winter 2016-2017 by approximately 0.3 m (Fig. 5), despite the prevalence of negative freeboard estimates in this product (Fig. 10). Such anomalies require further study to evaluate their actual causes, and this can be accomplished for example through more detailed, along-orbit comparisons between CS2-derived data products and coincident observations collected during aircraft underflights (e.g., Connor et al., 2011).

In terms of end user applications, the suitability of a particular product depends on the region of interest, as well as data latency and availability (Table 1). Moreover the purpose for which the data are used is critical in selecting the most suitable satellite product. For example climate assessments favour accuracy, while those engaged in operational or forecasting activities require low-latency, high-frequency observations. The frequency of the satellite data products evaluated here varies from twice a day (APP-x) to monthly (AWI), and latency varies from three / four days (APP-x and CPOM NRT) to products that are updated seasonally, or on an ad-hoc basis (GSFC and the CPOM seasonally-averaged thickness data product, Table 1). If access to NRT measurements, with year-round availability is required, APP-x would be the first choice, since it is the only product that provides daily coverage across the Arctic Ocean in both summer and winter, and provides a reasonable

measure of mean ice thickness, especially in some of the peripheral seas (regions 3-10) in the fall. However, it does not resolve the cross-basin ice thickness gradient nor the location of the thickest ice, and overestimates FYI thickness at the end of winter. Should basin-scale gradients in ice thickness (i.e., the thickness distribution) be important to the end user, then the CPOM NRT product is preferable, although it is only available for the winter-growth season (Tilling et al., 2018). In terms of climatological studies, or model initialization and hindcast studies, the CS2-only data products are appropriate options, but for navigation in the Arctic, none of these products are suitable as a single source of information, and the utility of the observations would only be realised when combined with additional ice charting analyses.

A remaining challenge for the satellite-derived thickness products is the treatment of snow depth on sea ice. All of the satellite-derived thickness products assessed here make use of the modified Warren et al. (1999) snow climatology, as outlined in Laxon et al., 2013, but there is variation in the implementation method, as described in Section 2.1. In addition, none of the implementations resolve year-to-year variations in snow depth. This has led a selection of end users, particularly those conducting data assimilation experiments, to use satellite-derived sea ice freeboard measurements, rather than ice thickness, since freeboard represents the remote sensing observation (rather than derived ice thickness). Currently there are only two products that provide the freeboard parameter, AWI and GSFC. Our analysis suggests that the AWI data set is preferable for ice freeboard due to a more realistic representation of measurements across the Arctic (Fig. 10). We found a high prevalence of erroneous, negative freeboard estimates throughout the Arctic in the GSFC product, that were especially concentrated in the peripheral seas, particularly in regions 7 and 8, as well as in the Beaufort Sea. The source of these anomalies is most likely associated with aspects of the GSFC algorithm and interpolation of the mean sea surface between lead tie-points.

In conclusion we suggest that low-latency, monthly composites, derived from CS2 data, or similar, but updated daily with the latest-available measurements, would benefit many sea ice thickness applications and provide an ideal solution to address many end-user needs. Further, it may be possible to obtain a more robust thickness distribution, through the inclusion of passive microwave observations of thin-ice thickness in the marginal ice zone. Although the CS2SMOS results are promising, most of our evaluations with independent data were focused on the MYI zone. Applying an appropriate ratio, that adequately combines microwave and altimeter observations, is challenging and requires further evaluation with independent observations collected over FYI and in the peripheral seas. Higher resolution (<= 5 km) along-orbit and gridded data products would advance the utility of the observations at the regional scale. We also recommend that future products include both ice thickness and freeboard parameters, as well as an estimate of thickness uncertainty and/or data quality flags, so that the satellite observations may be used in data assimilation experiments aimed at improving ice forecasting.

**Author contribution**

SLF designed the study, HS and JM carried out the data analysis under the guidance of SLF, and JM was responsible for producing the figures. HS led the manuscript preparation, and SLF, ER and JM contributed to the writing and internal review.

**Competing interests**

The authors declare that they have no conflicts of interest.

**Acknowledgements**

All data utilized in this study are publicly available and all data sources are listed in Table 1. This work has been funded through the ESA Sea Ice Climate Change Initiative (CCI) project and the NOAA Product Development, Readiness, and Application (PDRA) / Ocean Remote Sensing (ORS) Program, under NOAA grant NA14NES4320003 (NOAA Cooperative Institute for Climate and Satellites, University of Maryland). We thank the Editor, reviewers J. Landy, T. Armitage and R. Ricker, as well as N. Kurtz, S. Hendricks, R. Tilling and S. Fleury for their comments and valuable reviews, which helped to improve the manuscript.

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

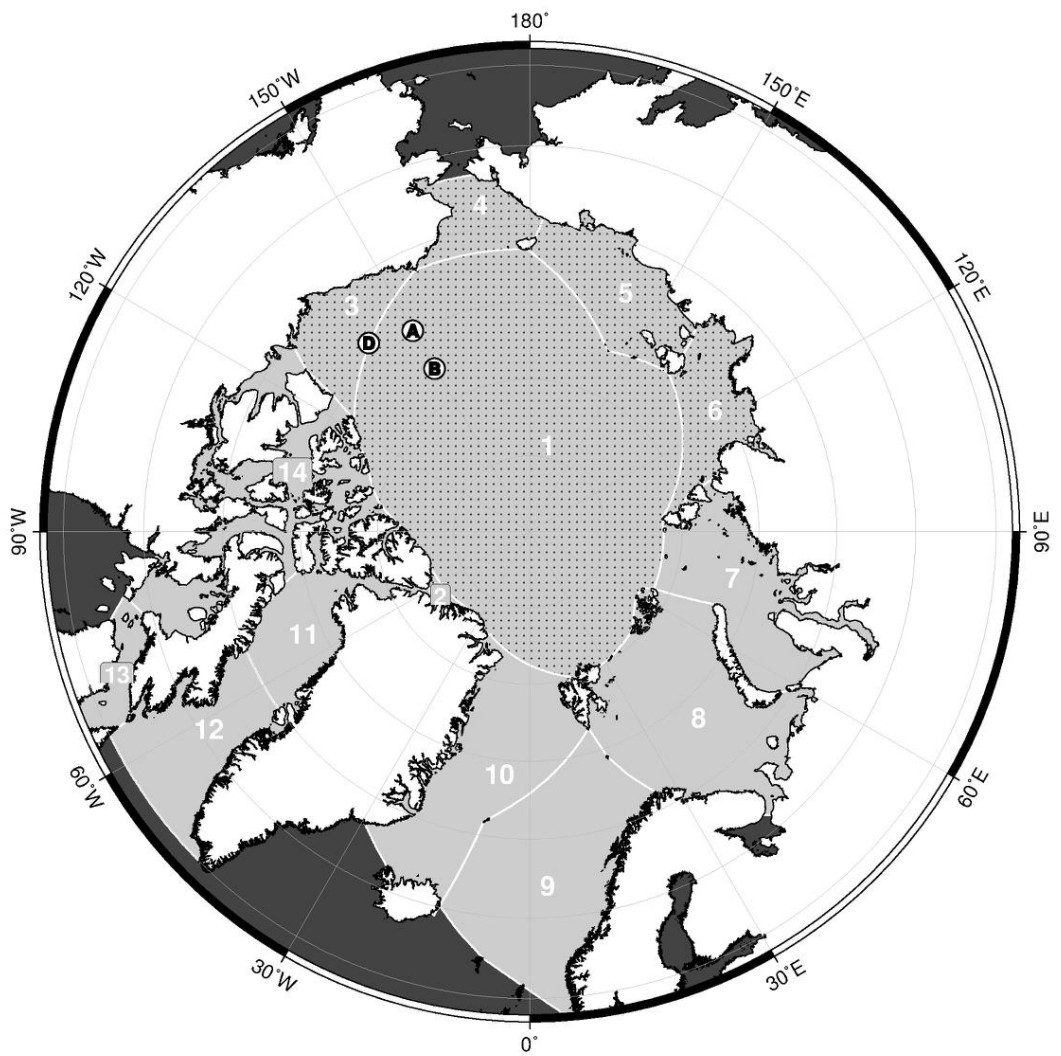

**Figure 1: Regions utilised in data analysis: (1) Central Arctic Ocean, (2) Lincoln Sea, (3) Beaufort Sea, (4) Chukchi Sea, (5) East Siberian Sea, (6) Laptev Sea, (7) Kara Sea, (8) Barents Sea, (9) Norwegian Sea, (10) Greenland Sea, (11) Baffin Bay, (12) Davis Strait, (13) Hudson Strait, and (14) Canadian Arctic Archipelago. The dotted area represents the central Arctic region (1–6) within which all data products are available. The locations of Beaufort Gyre Exploration Project (BGEP) moorings A, B, and D are also indicated (white circles with mooring designation).**

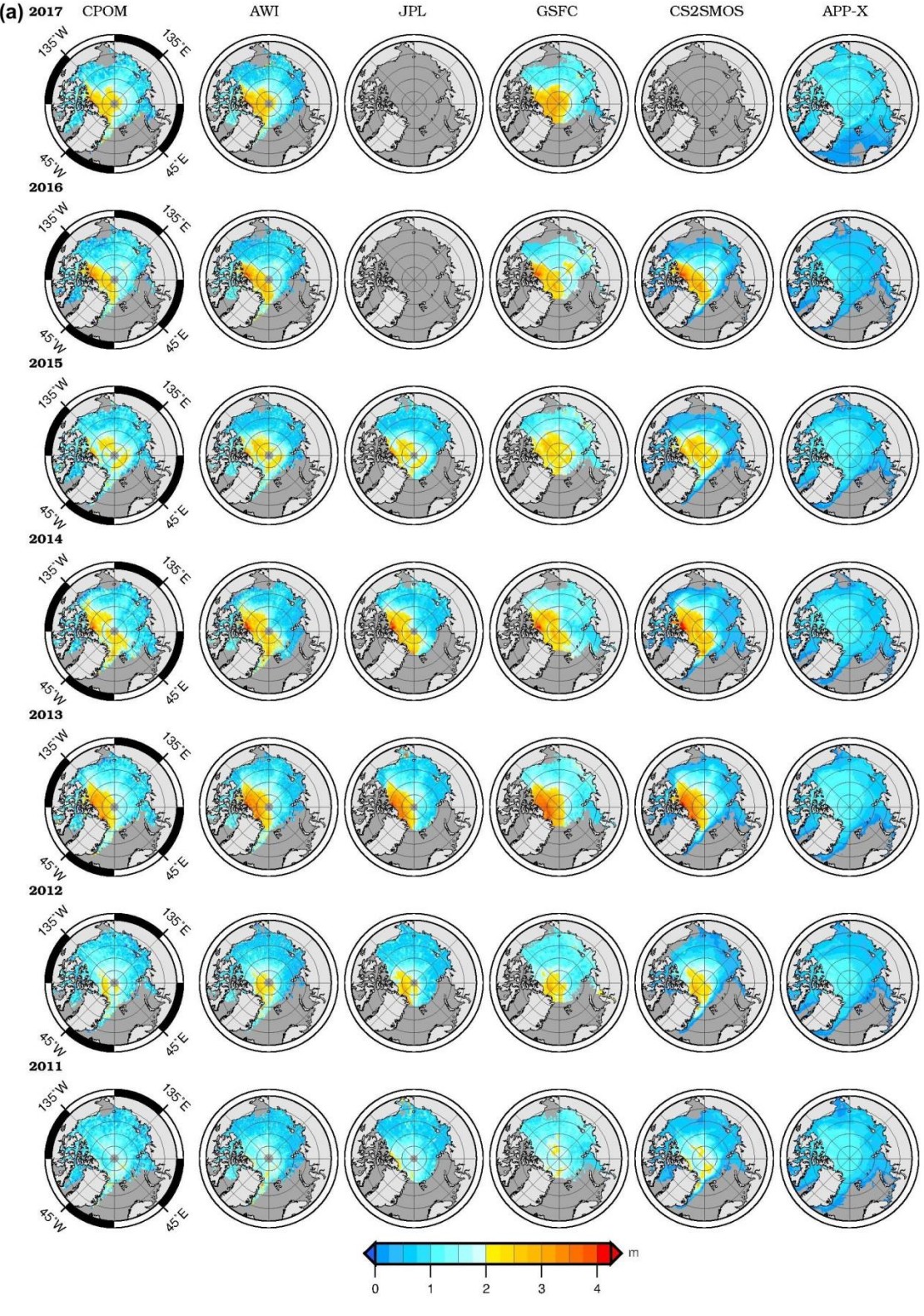

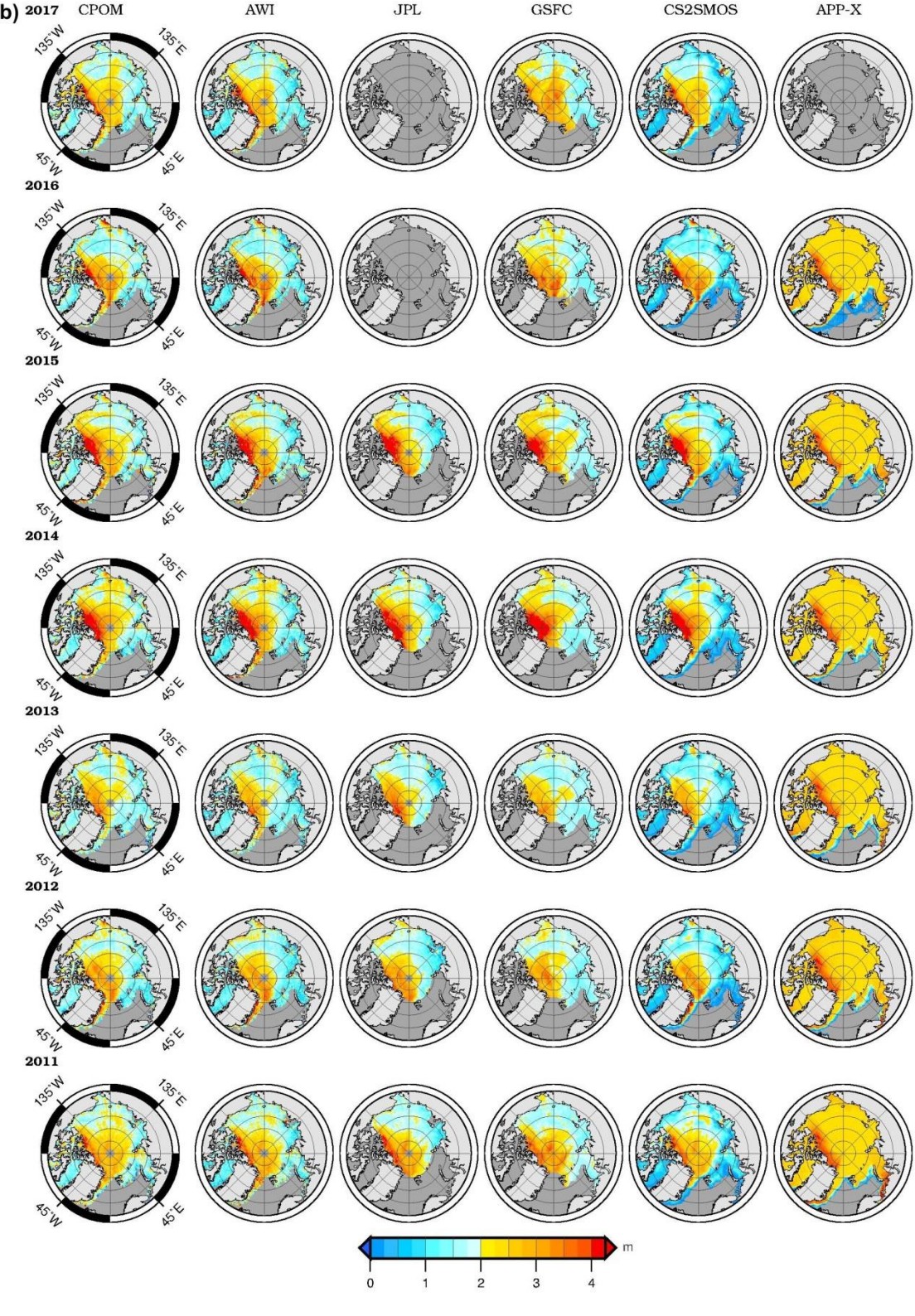

**Figure 2: Maps of seasonally-averaged sea ice thickness for each product over the period 2011–2017, for (a) October–November, and (b) March–April, for regions 1-14, where data are available.**

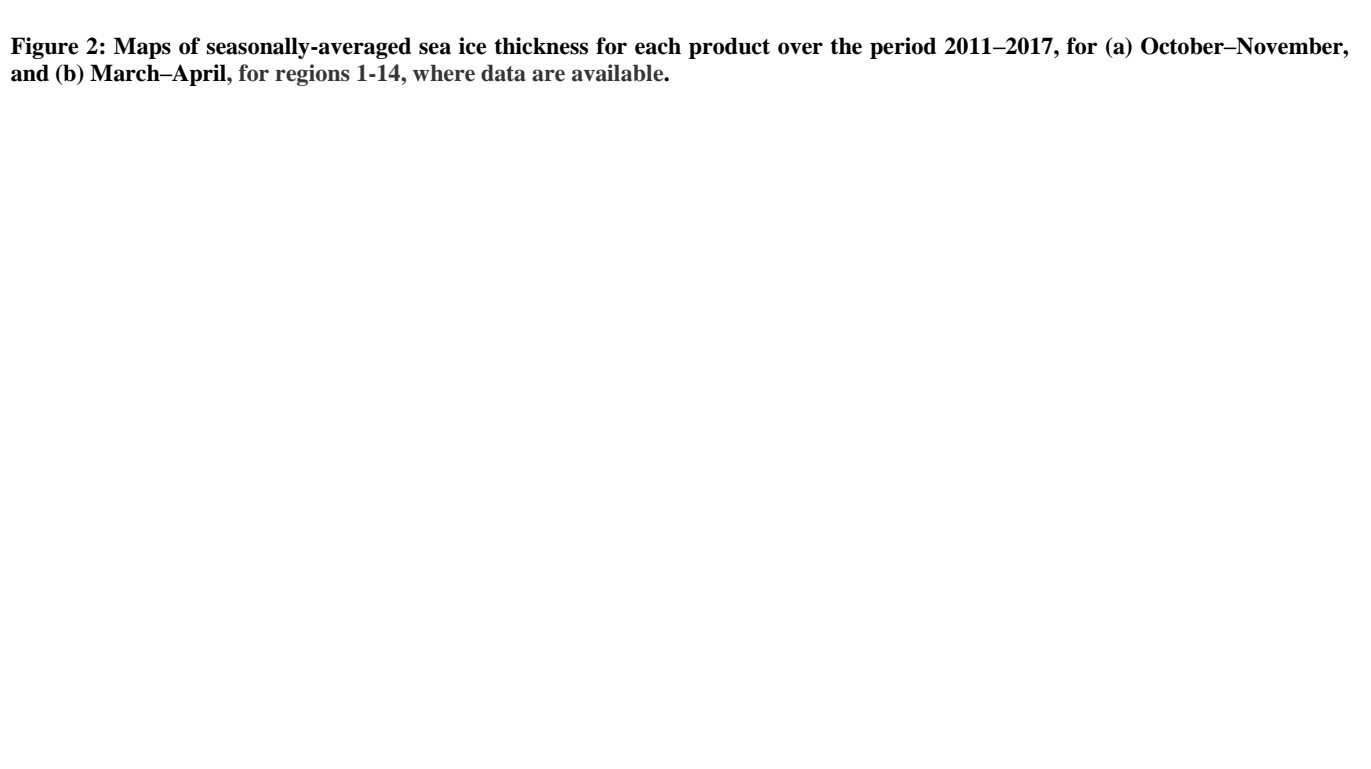

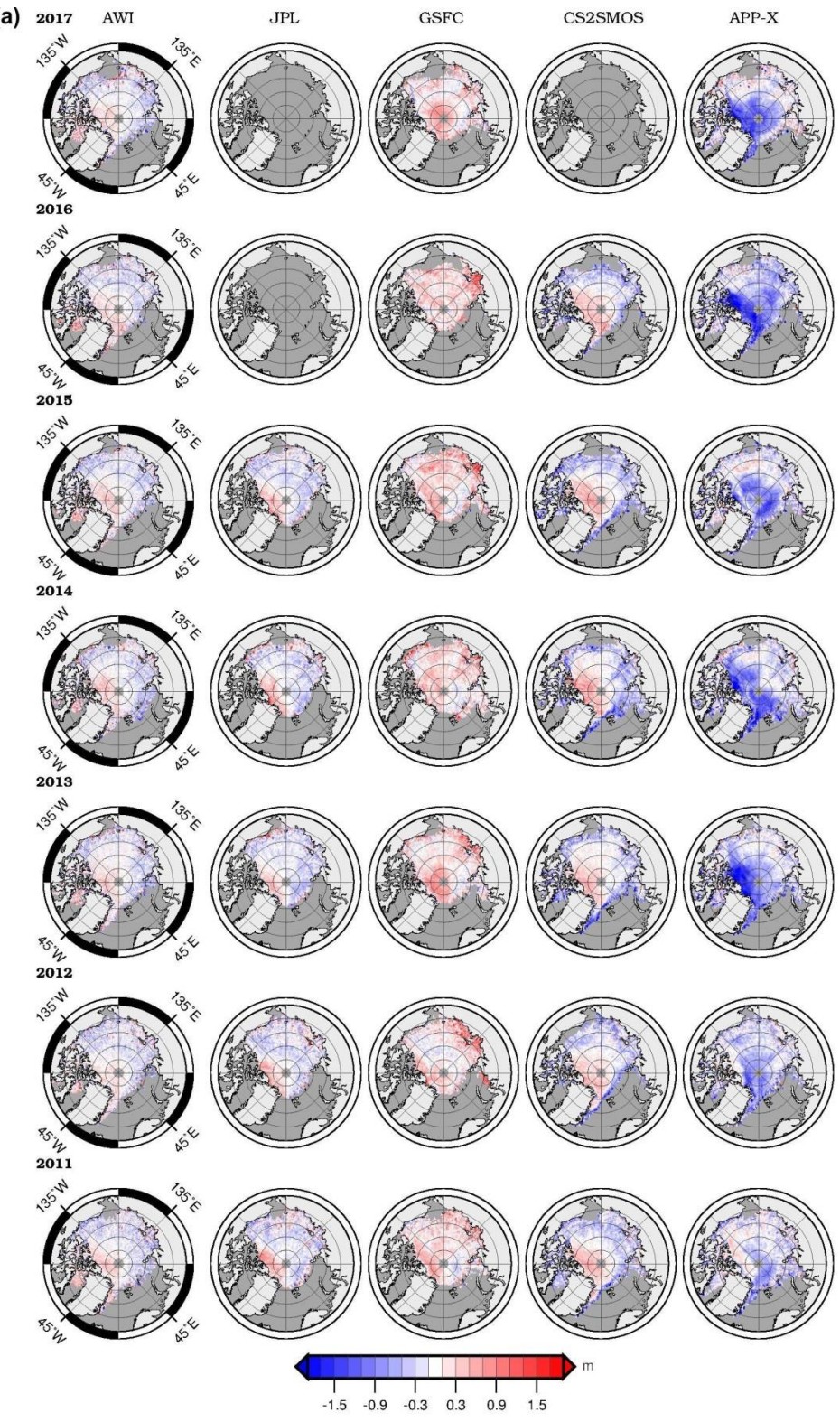

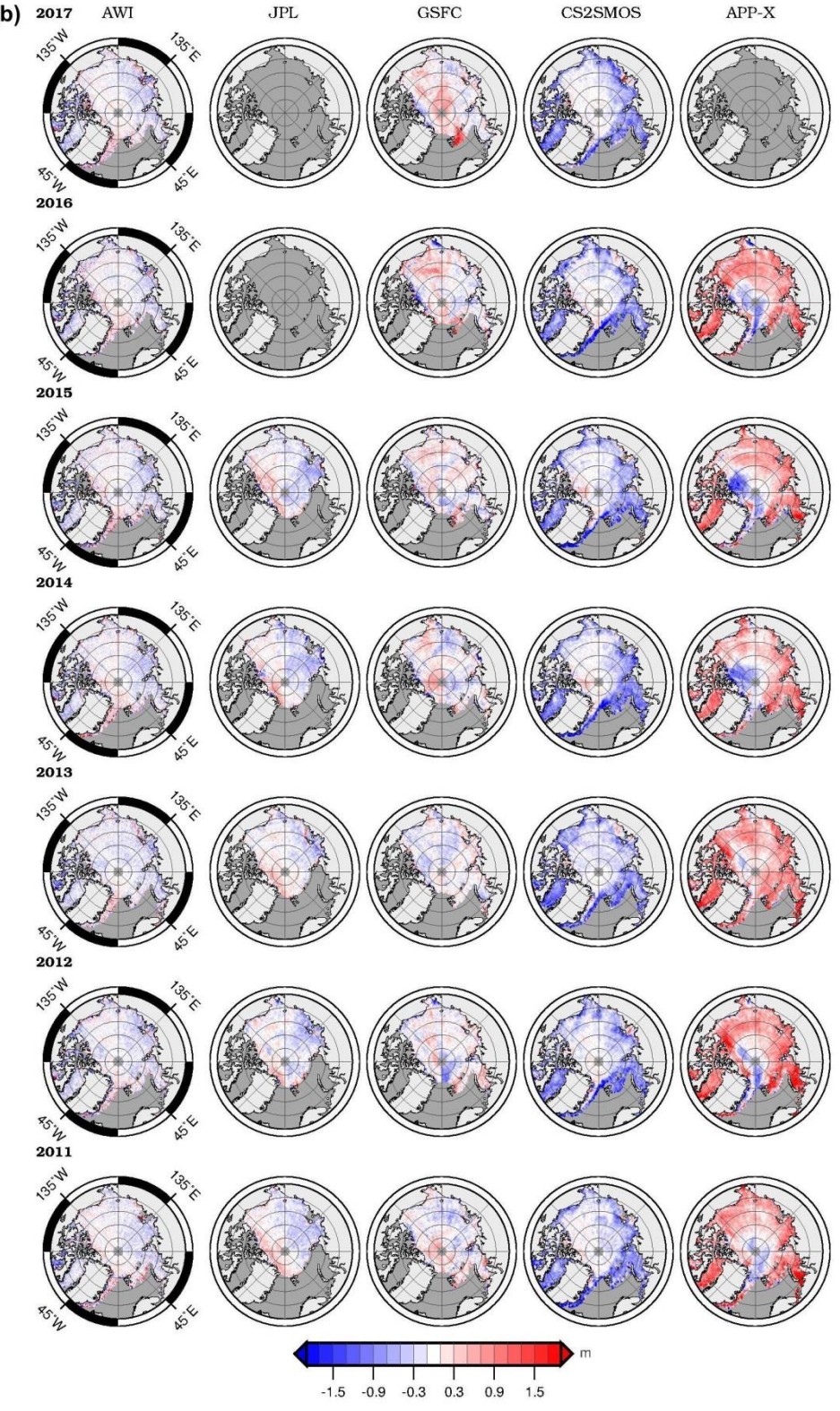

**Figure 3: Maps of seasonally-averaged sea ice thickness differences for the period 2011–2017, where the reference data set (CPOM) is subtracted from each data product, for (a) October–November and (b) March–April, for regions 1-14, where data are available. Red (blue) regions indicate areas where the seasonally-averaged thickness is greater (less) than the reference data product.**

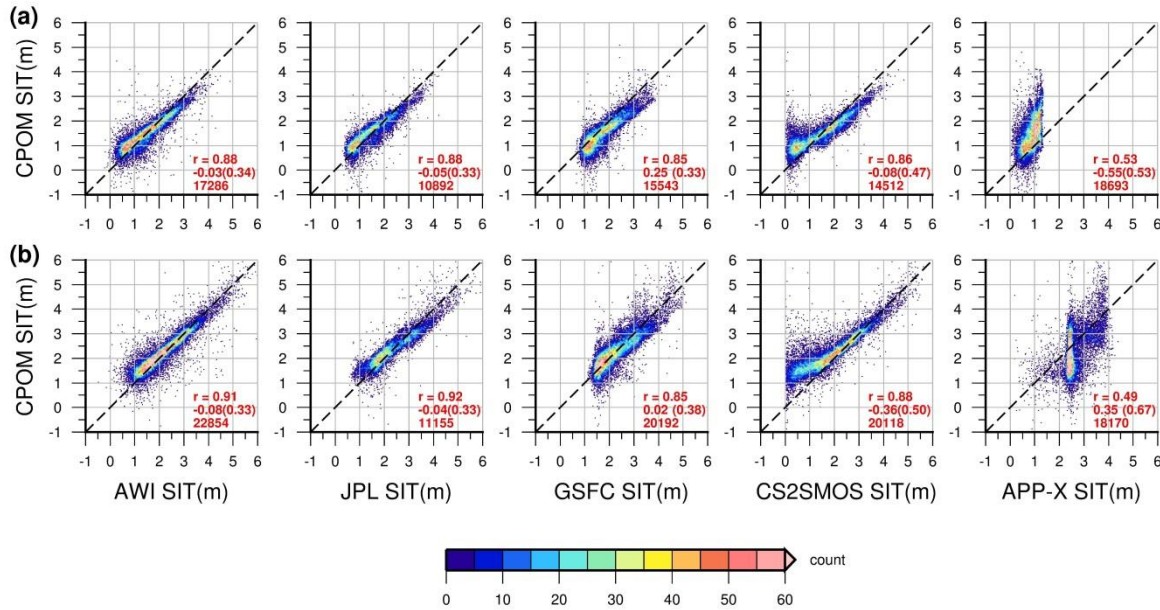

**Figure 4: Comparisons of sea ice thickness for each product and the reference data set for (a) October–November and (b) March–April, for the period 2010-2018, regions 1-14 north of 65°N, subject to data availability (Table 1). Colour indicates measurement density, derived from the number of data points within each 0.05 m cell. Statistics for correlation (r), mean difference (std. dev.), and number of data points are provided.**

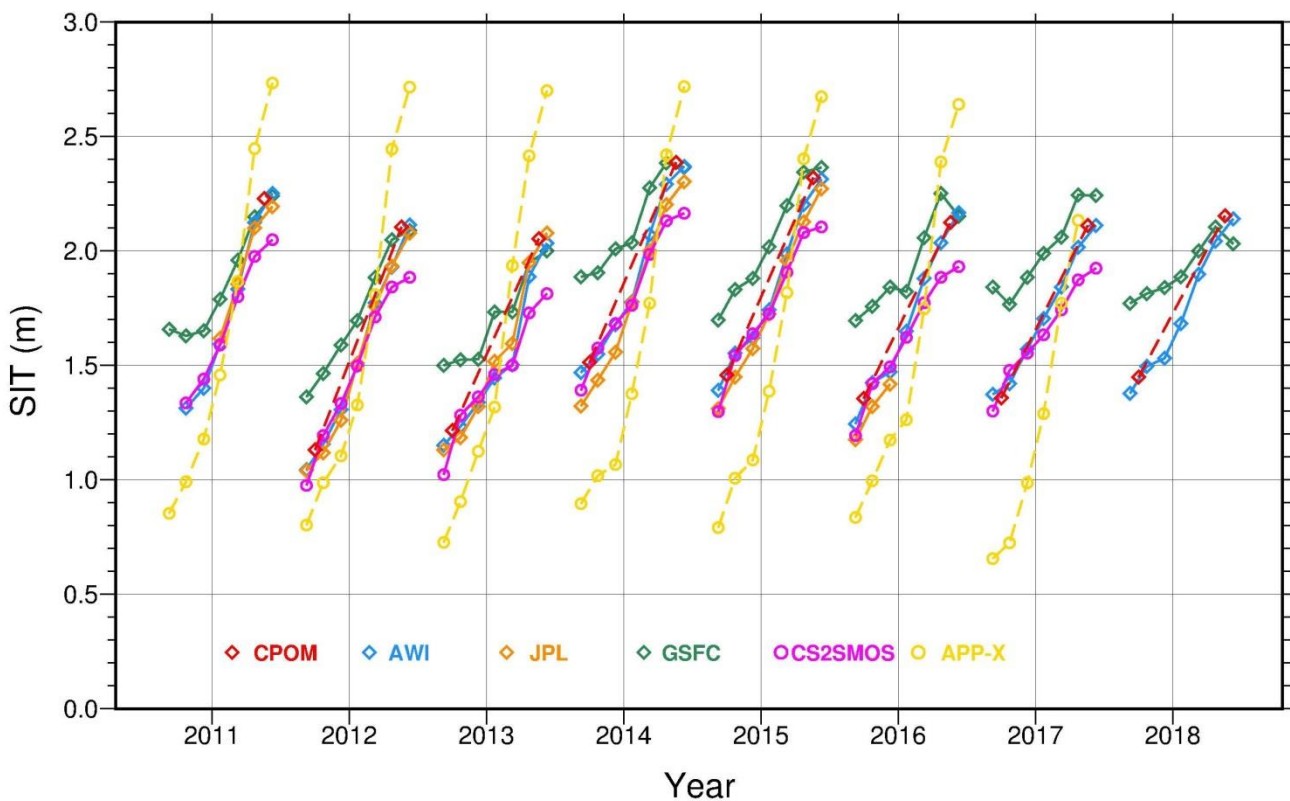

**Figure 5: Sea ice thickness growth curves for October–April (monthly averages) for the central Arctic (regions 1-6) for the period 2010–2018, subject to data availability (Table 1), indicating interannual variability in the winter-time thickness evolution. For the CPOM data product, October–November and March–April seasonal averages are shown (red diamonds connected by a dashed red line).**

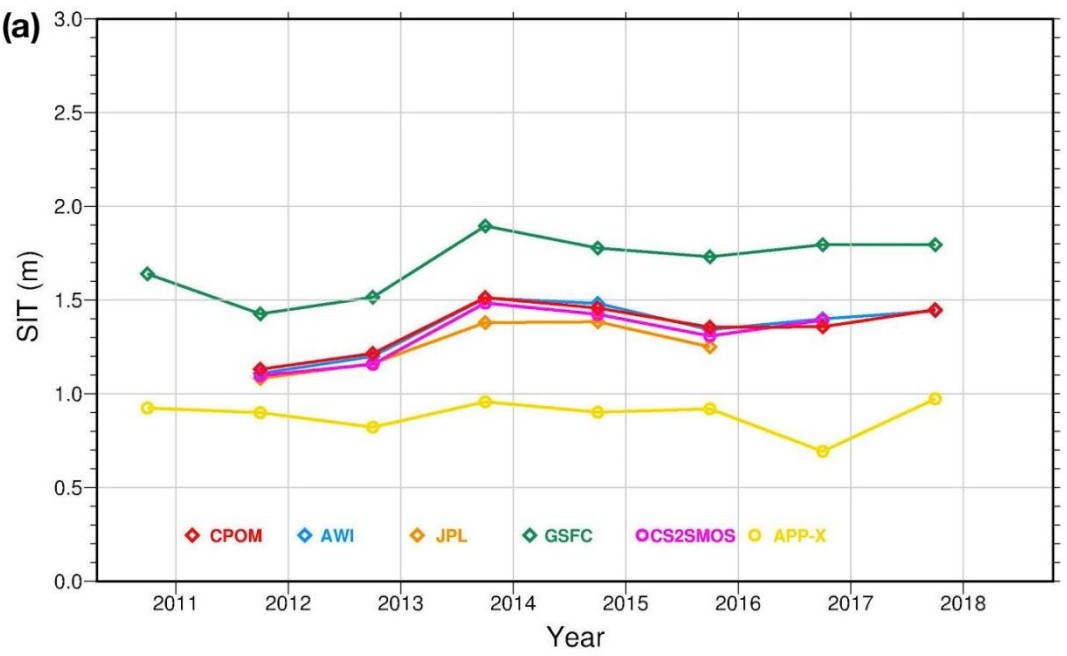

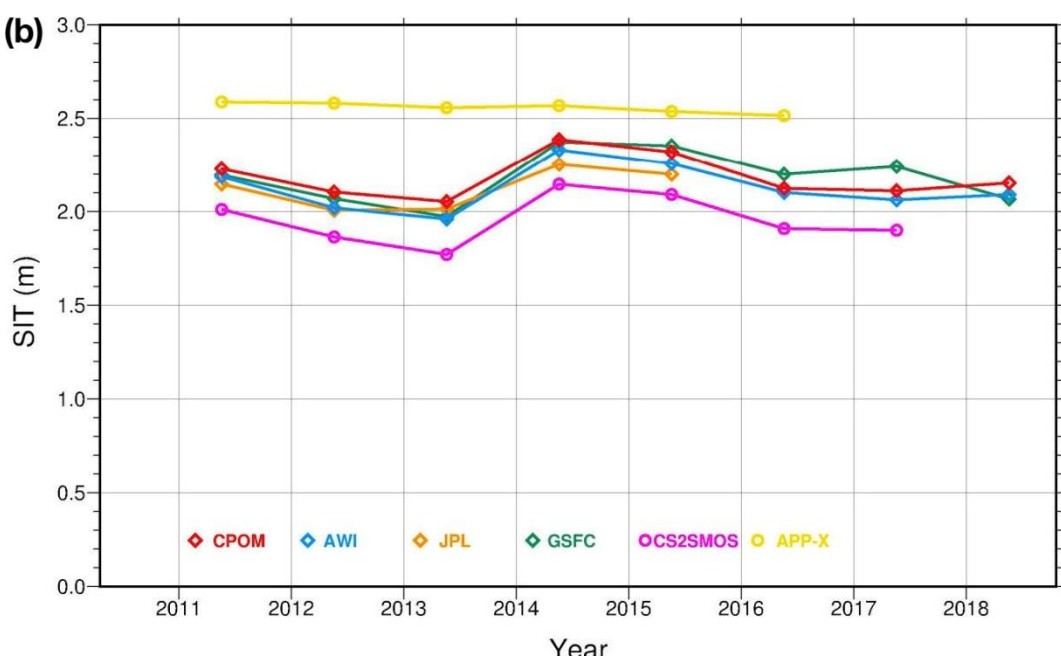

**Figure 6: Time series of seasonally-averaged sea ice thickness for (a) October–November, and (b) March–April, over the central Arctic (regions 1–6) during the period 2010–2018, subject to data availability (Table 1).**

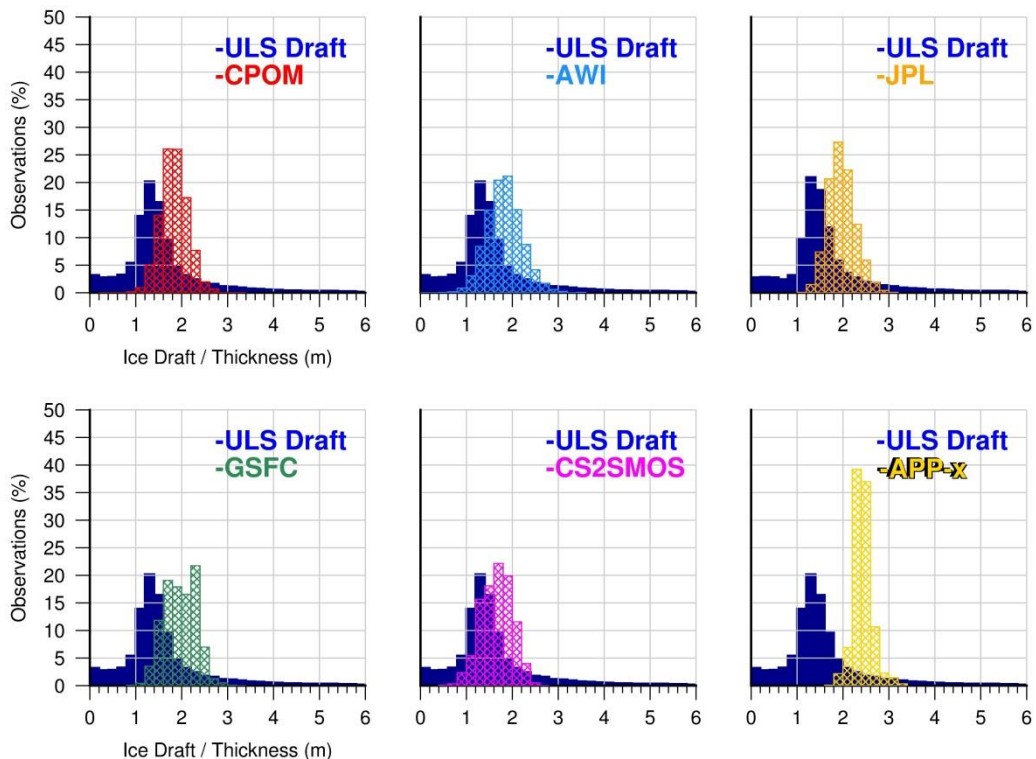

**Figure 7: Spring (March–April) sea ice thickness distributions (cross-hatched) for each data product within 200 km radius of BGEP mooring locations, averaged for the period 2011 to 2017\*, overlaid on the corresponding BGEP upward looking sonar (ULS) ice draft distribution (dark blue, solid). \*The JPL data product averaging period is 2011 to 2015, and the APP-x data product average does not include thickness data for April 2017. Histogram bin width is 0.2 m.**

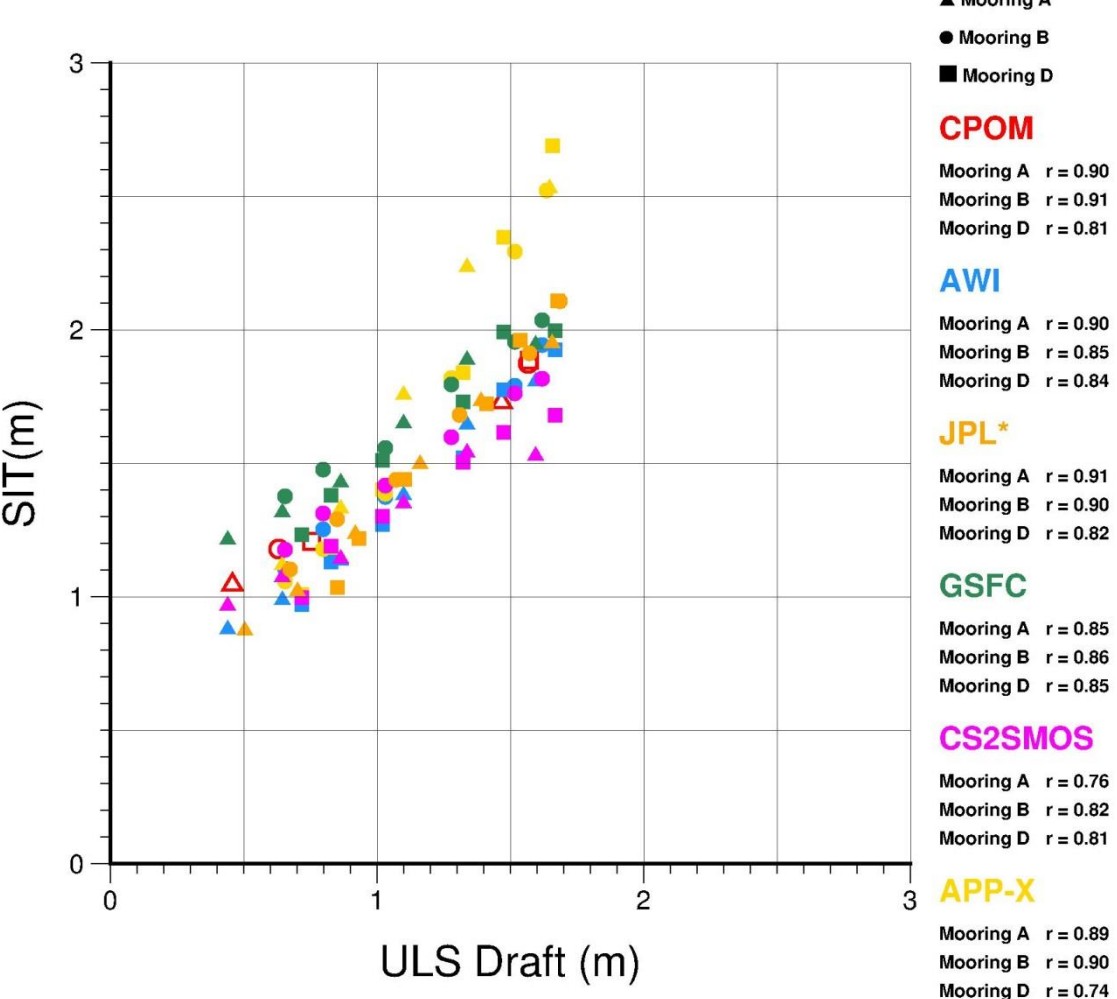

**Figure 8: Correlation between monthly-averaged, satellite-derived ice thickness and ULS ice draft, for six months spanning November to April. Correlation coefficients are provided per ULS mooring (mooring location indicated by symbols), for the period 2010–2017, with the exception of the JPL product, wherein monthly averages and correlation coefficients are calculated for the period 2011–2015, and the APP-x product, where spring 2017 was excluded. In the case of the CPOM product the correlation coefficients are calculated based on seasonal (October–November, March–April) rather than monthly averages. To aid visualisation, monthly averages were further combined across years (e.g. November 2010–2017, December 2010–2017, etc.) to provide six data points per mooring-product comparison.**

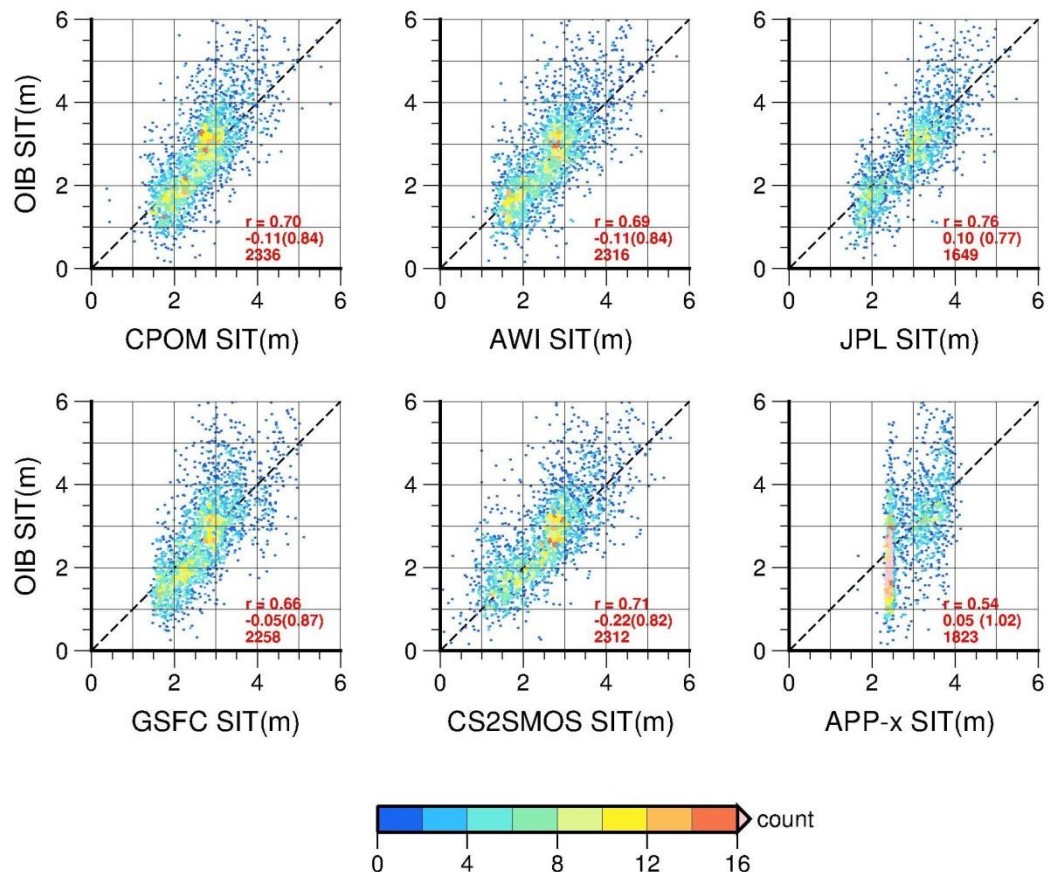

**Figure 9: Comparison of satellite-derived ice thickness with Operation IceBridge thickness estimates at the end of the winter growth season (March–April). Comparisons were conducted by gridding satellite and aircraft data onto a common 0.4° latitude by 4° longitude grid and using grid cells in which both data sets contained thickness estimates. Colour indicates measurement density (number of data points within each 0.01 m cell). Statistics for correlation (r), mean difference (std. dev.), and number of data points are calculated for the period 2011–2017, with the exception of the JPL product, wherein statistics are calculated for the period 2011–2015, and the APP-x product, where spring 2017 was excluded.**

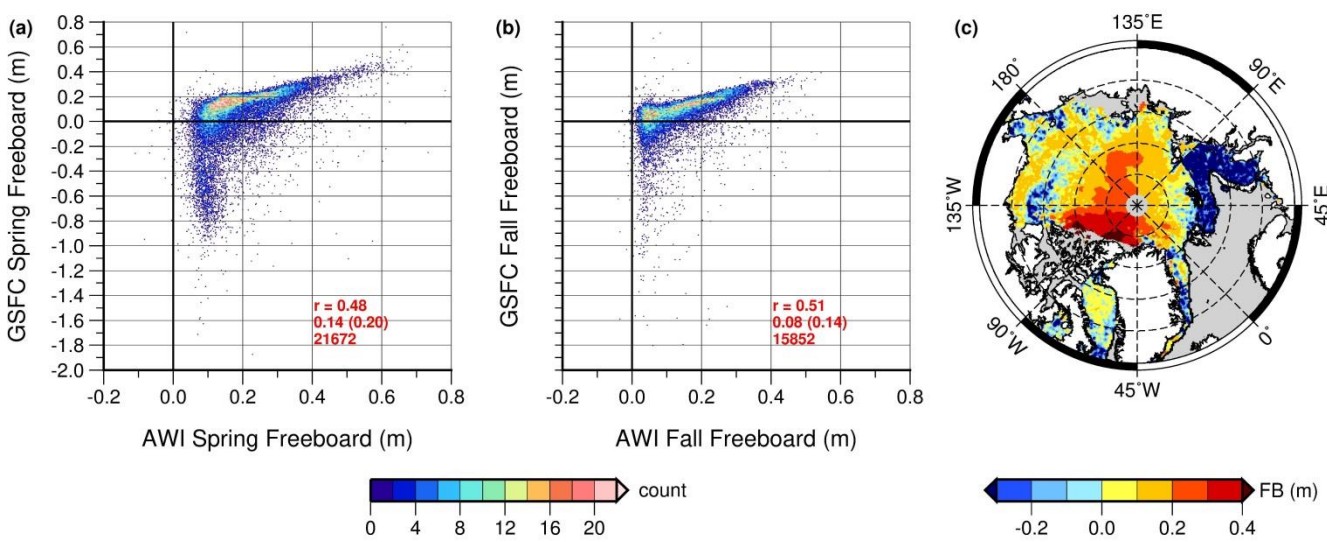

Figure 10: Comparison of sea ice freeboard in the GSFC and AWI data products for (a) March–April and (b) October–November, for the period 2011–2018, regions 1-14, north of 65°N. Colour indicates measurement density, derived from the number of data points within each 0.01 m cell. Statistics for correlation (r), mean difference (std. dev.), and number of data points are provided. (c) GSFC product freeboard (FB) for April 2014.

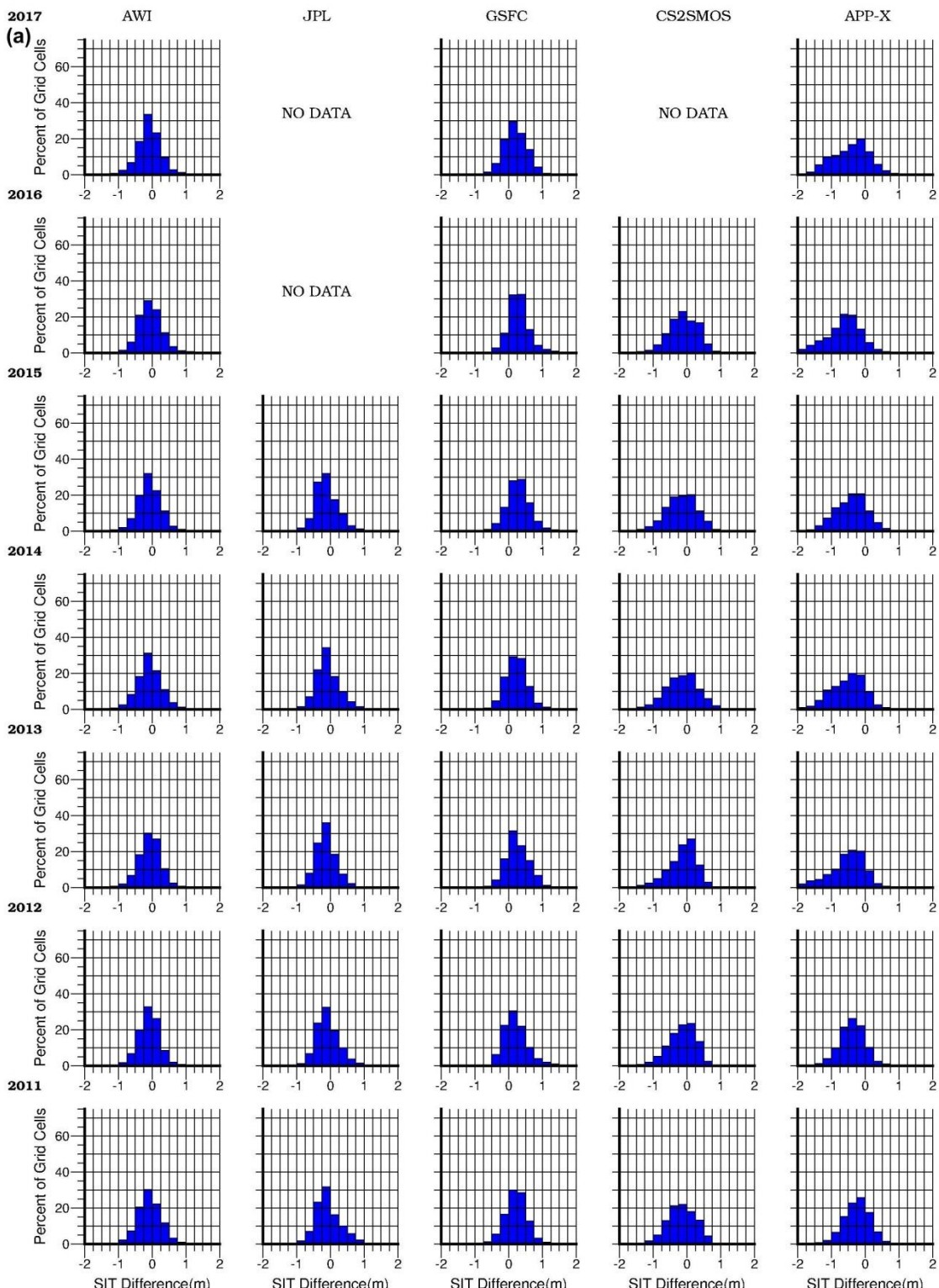

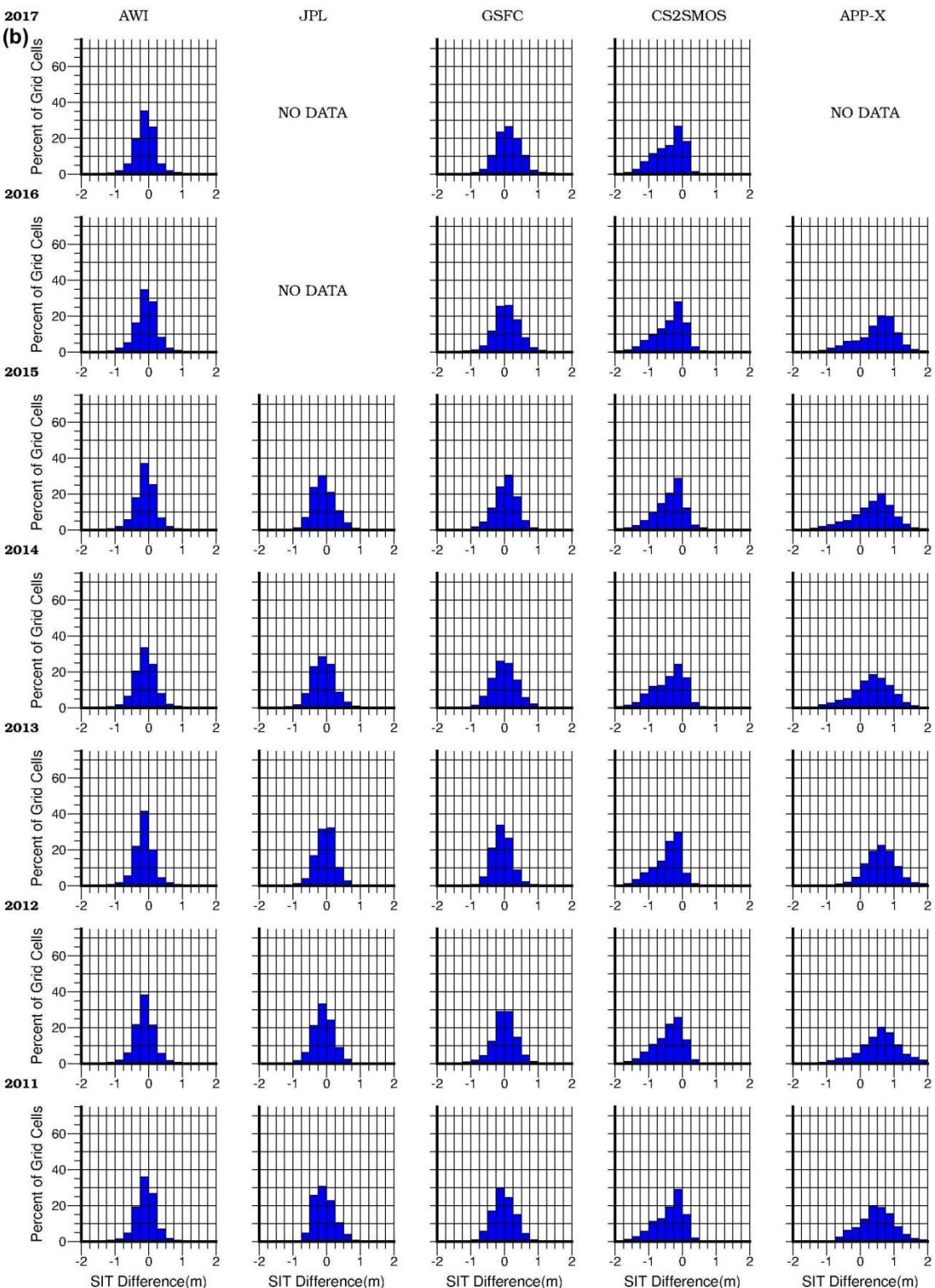

**Figure S1: Histograms of seasonally-averaged sea ice thickness, where the reference data set (CPOM) is subtracted from each product, for (a) October–November, and (b) March–April, in the central Arctic (regions 1–6), for the period 2011–2017, subject to data availability (Table 1).**

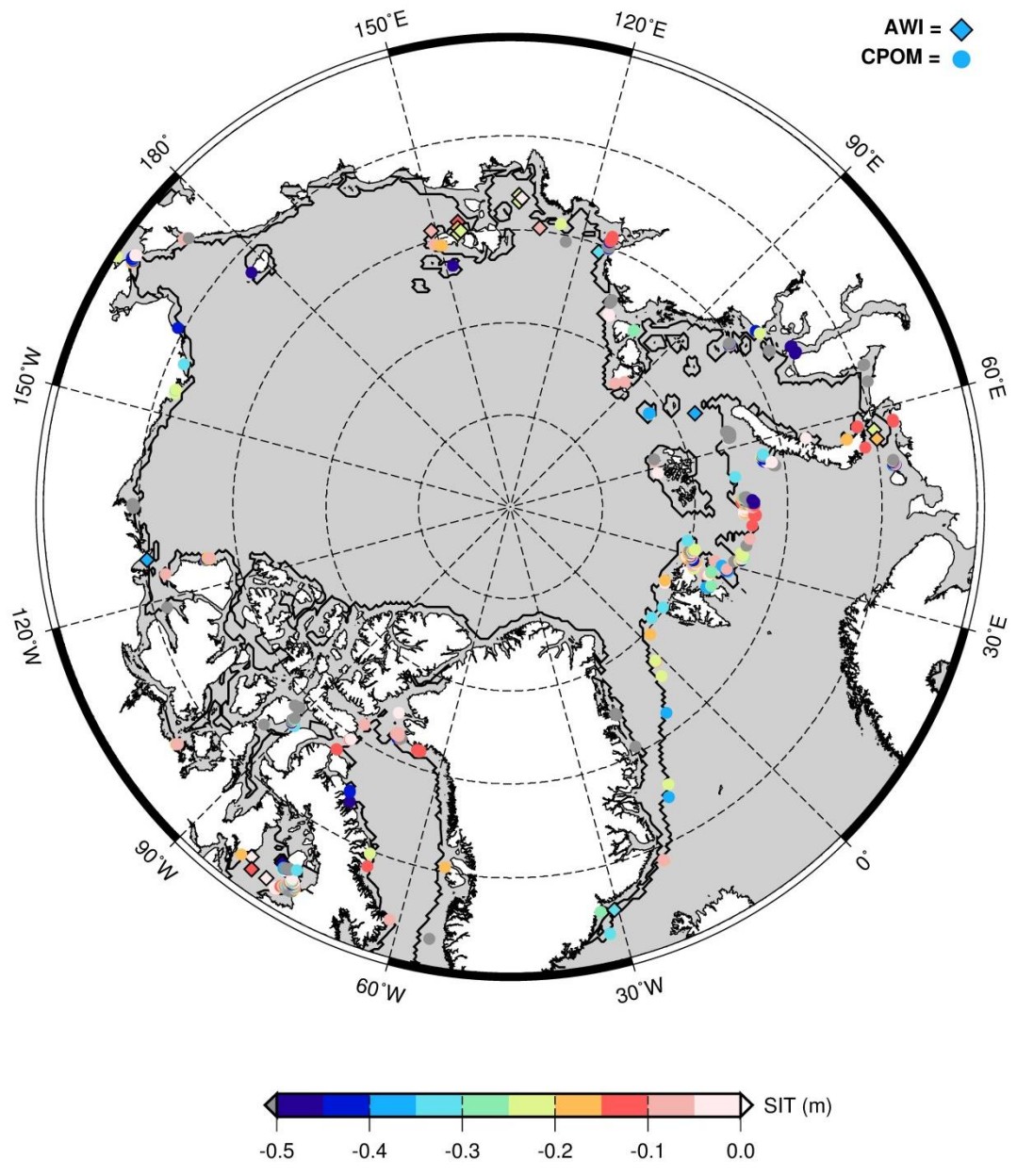

**Figure S2: Locations of negative sea ice thickness values in the CPOM (circles) and AWI (diamonds with black outline) data products, for March–April 2014 (CPOM) and April 2014 (AWI). A contour representing the 70 % ice concentration level is also shown (thick black line).**

**Table 1. Characteristics of satellite-derived ice thickness products ("quality information" refers to the availability of a thickness uncertainty estimate or quality flag). Latency is the difference between the date of data acquisition and data delivery, estimated from data portal time stamps at the time of writing. Frequency refers to the temporal offset between consecutive datasets (e.g. daily, weekly, monthly). \*The northern limit of coverage (81.5 °N) impacts the availability of the Envisat product, especially in the Central Arctic (region 1).**

| Product Name | CPOM | AWI | JPL | GSFC | CS2SMOS | APP-x | LEGOS TFMRA | ESA CCI |
|---|---|---|---|---|---|---|---|---|
| Temporal Range | November 2010-date | November 2010-date | January 2011-December 2015 | October 2010-date | November 2010-April 2017 | January 1982-date | November 2002-March 2012 * November 2010-April 2017 | October 2002-March 2012 * November 2010-April 2017 |
| Frequency | Daily (NRT)/monthly / seasonally | Monthly | Not updated post 2015 | Daily | Weekly | Twice daily | Monthly | Daily |
| Geographical Coverage (by region number) | (1)–(14), poleward of 40 ºN | (1)–(14), poleward of 60 ºN | (1)–(6) | (1)–(8), (10, partial), poleward of 55 ºN | (1)–(14), poleward of 50 ºN | (1)–(14), poleward of 50 ºN | 65 ºN-81.5 ºN*; poleward of 65 ºN | (1), (3)-(14) south of 81.5 ºN*; (1)-(14), poleward of 16.6 ºN |
| Averaging Period | NRT data product: 2, 14, 28 days. Final products: monthly, seasonally-averaged | 1 month | 1 month | 30 day | 7 day | 12 hours | 1 month | 1 month (moving average) |
| Latency | 3 days (NRT) / seasonally | Variable | Not updated post 2015 | ~6 weeks | Variable | ~4 days (as of April 2018) | ~1 year | Variable |
| Grid Resolution | 25 km, 5 km (full Arctic), 1 km (individual regions) | EASE2, 25 km | 25 km | Polar Stereographic SSM/I, 25 km | EASE2, 25 km | EASE, 25 km | 12.5 km | EASE2, 25 km |
| Freeboard | No | Yes | No | Yes | No | No | Yes | Yes |
| Quality Information | No | Yes | No | No | Yes | No | Yes | Yes |
| Retracking technique | Retracking based on ice type (lead, floe); Lead retracker follows Giles et al. 2007; Floe retracker based on 70 % of peak amplitude on the waveform leading edge | 50% TFMRA | Waveform centroid retracker | Waveform fitting using waveform model | Blended SMOS and CS2 data | One-dimensional Thermodynamic Ice Model (OTIM) | 60% TFMRA | 50% TFMRA |
| Mean Sea Surface | UCL2013 | DTU15 MSS | EGM2008 geoid | DTU10 MSS | DTU15 MSS (for CS-2 data) | N/A | DTU15 MSS | DTU15 MSS |
| Snow Depth | Monthly constants for FYI and MYI based on the modified climatology of Warren et al., 1999 (MWC) | MWC | MWC | MWC | MWC (CS2 data); linear relation with ice thickness (SMOS data) | Climatology of Warren et al. (1999) combined with other available estimates | MWC | MWC |
| Ice Density | FYI: 916.7 kg/m$^3$; MYI: 882.0 kg/m$^3$ | FYI: 916.7 kg/m$^3$; MYI: 882.0 kg/m$^3$ | FYI: 917 kg/m$^3$; MYI: 882.0 kg/m$^3$ | 915 kg/m$^3$ | FYI: 916.7 kg/m$^3$; MYI: 882.0 kg/m$^3$ | N/A | FYI: 916.7 kg/m$^3$; MYI: 882.0 kg/m$^3$ | FYI: 916.7 kg/m$^3$; MYI: 882.0 kg/m$^3$ |
| Ice Concentration, Threshold | NSIDC Near-Real-Time DMSP SSMIS Daily Polar Gridded Sea Ice Concentrations, 75% | OSI-SAF, 70% | Not specified | NSIDC Near-Real-Time DMSP SSMIS Daily Polar Gridded Sea Ice Concentrations, 70% | OSI-SAF, 15% | NSIDC Nimbus-7 SMMR and DMSP SSM/I passive microwave data with NASA Team Algorithm applied, 15% | Sea ice age (NSIDC), 50% | OSI-SAF, 70% |
| Ice Type | OSI-SAF | OSI-SAF | ASCAT | OSI-SAF | OSI-SAF | Converted from reflectances | Sea ice age (NSIDC) | Integrated Climate Data Center (ICDC) ice type fraction |
| References | Laxon et al., 2013; Tilling et al., 2016; Tilling et al., 2018 | Ricker et al., 2014; Hendricks et al., 2016 | Kwok and Cunningham, 2015 | Kurtz et al., 2014; Kurtz and Harbeck, 2017 | Tian-Kunze et al., 2014; Kaleschke et al., 2015, Ricker et al., 2017b | Wang et al., 2010; Key and Wang, 2015; Wang et al. 2016 | Guerreiro et al., 2017 | Hendricks, S., 2017; Paul et al., 2017; Hendricks et al., 2018 |
| Public Data Source | http://www.cpom.ucl.ac.uk/csopr/seaice.html | Meereisportal, http://data.meereisportal.de/ (Grosfeld et al. 2016) | https://rkwok.jpl.nasa.gov/cryosat/download.html | https://nsidc.org/data/RDEFT4 | Meereisportal, http://data.meereisportal.de/ (Grosfeld et al. 2016) | https://www.ncei.noaa.gov/data/avhrr-polar-pathfinder-extended/access/nhem/ | http://ctoh.legos.obs-mip.fr | http://cci.esa.int/data |

**Table 2. Seasonally-averaged sea ice thickness for each Arctic region (in meters) for all six data products. Regional thickness is reported for all regions that contain valid thickness estimates. The data used in this table spans the period 2010-2018, based on product availability, as outlined in Table 1. Average thicknesses are calculated only based on years in which ice is present.**

| | October–November | | | | | | March–April | | | | | |
|---|---|---|---|---|---|---|---|---|---|---|---|---|
| | CPOM | AWI | JPL | GSFC | CS2SMOS | APP-x | CPOM | AWI | JPL | GSFC | CS2SMOS | APP-x |
| Arctic Ocean [1] | 1.48 | 1.48 | 1.39 | 1.77 | 1.49 | 0.96 | 2.3 | 2.25 | 2.29 | 2.33 | 2.15 | 2.58 |
| Lincoln Sea [2] | 1.99 | 2.11 | 2.52 | 2.42 | 2.43 | 1.19 | 3.61 | 3.51 | 3.7 | 3.34 | 3.47 | 3.66 |
| Beaufort Sea [3] | 1.03 | 0.79 | 0.89 | 1.2 | 0.57 | 0.77 | 2.03 | 1.97 | 2.09 | 2.00 | 1.59 | 2.59 |
| Chukchi Sea [4] | 0.84 | 0.61 | 0.96 | 1.13 | 0.21 | 0.47 | 1.91 | 1.79 | 1.81 | 1.92 | 1.46 | 2.51 |
| East Siberian Sea [5] | 0.92 | 0.77 | 0.81 | 1.34 | 0.52 | 0.68 | 1.83 | 1.72 | 1.56 | 1.72 | 1.39 | 2.44 |
| Laptev Sea [6] | 0.82 | 0.55 | 0.67 | 1.26 | 0.43 | 0.61 | 1.53 | 1.27 | 1.12 | 1.56 | 0.97 | 2.4 |
| Kara Sea [7] | 0.85 | 0.48 | 0.87 | 1.01 | 0.37 | 0.49 | 1.59 | 1.38 | 1.63 | 1.55 | 0.79 | 2.52 |
| Barents Sea [8] | 1.31 | 0.78 | 1.26 | 1.51 | 0.37 | 0.36 | 1.41 | 1.44 | 1.99 | 1.73 | 0.41 | 1.99 |
| Greenland Sea [10] | 1.69 | 1.78 | 2.14 | 2.4 | 1.16 | 0.56 | 2.47 | 2.79 | 3.02 | 2.97 | 1.65 | 2.18 |
| Baffin Bay [11] | 0.91 | 1.28 | - | 2.47 | 0.26 | 0.63 | 1.5 | 1.27 | - | 3.39 | 0.64 | 2.47 |
| Davis Strait [12] | 1.01 | - | - | - | 0.29 | 0.24 | 1.55 | 1.15 | - | - | 0.47 | 2.54 |
| Hudson Strait [13] | 1.12 | - | - | - | 0.12 | 0.22 | 1.53 | 0.82 | - | - | 0.53 | 2.93 |
| Canadian Archipelago [14] | 1.25 | 1.26 | 1.19 | 2.03 | 0.81 | 0.84 | 1.95 | 1.58 | 2.09 | 2.27 | 1.4 | 2.64 |

**Table 3. Anomalies of seasonally-averaged annual sea ice thickness (in meters) relative to the seasonally-averaged baseline mean within the central Arctic (regions 1–6). The baseline mean is calculated for the period 2011–2015 . Anomalies relative to the 2011-2015 mean are included for later seasons according to the product availability.**

| October–November | 2011 | 2012 | 2013 | 2014 | 2015 | 2016 | 2017 | 2018 |
|---|---|---|---|---|---|---|---|---|
| CPOM | -0.2 | −0.12 | 0.18 | 0.12 | 0.02 | 0.02 | 0.11 | |
| AWI | -0.22 | -0.13 | 0.18 | 0.15 | 0.01 | 0.06 | 0.1 | |
| JPL | -0.17 | −0.09 | 0.13 | 0.13 | 0.0 | - | - | |
| GSFC | -0.25 | -0.16 | 0.22 | 0.1 | 0.06 | 0.13 | 0.13 | |
| CS2SMOS | -0.2 | -0.14 | 0.19 | 0.12 | 0.01 | 0.1 | - | |
| APP-x | 0.0 | -0.08 | 0.06 | 0.0 | 0.02 | −0.21 | 0.07 | |
| **March-April** | **2011** | **2012** | **2013** | **2014** | **2015** | **2016** | **2017** | **2018** |
| CPOM | 0.01 | -0.11 | -0.17 | 0.17 | 0.1 | -0.09 | -0.1 | -0.06 |
| AWI | 0.04 | -0.13 | -0.19 | 0.18 | 0.11 | -0.04 | -0.08 | -0.05 |
| JPL | 0.02 | -0.12 | -0.11 | 0.13 | 0.08 | - | - | |
| GSFC | 0.0 | -0.12 | -0.22 | 0.18 | 0.16 | 0.01 | 0.05 | -0.13 |
| CS2SMOS | 0.03 | -0.11 | -0.21 | 0.17 | 0.11 | -0.07 | -0.08 | |
| APP-x | 0.02 | 0.01 | -0.01 | 0.0 | -0.03 | −0.05 | - | |

**Table 4.** Annual winter (January–April*) sea ice draft/thickness (meters, top value) and deviation from the 2011-2017 mean (meters, bottom value) for each BGEP ULS mooring (A, B, D) and six satellite products (using measurements within 200 km of BGEP moorings). *March–April for CPOM statistics.

| | | ULS | CPOM | AWI | JPL | GSFC | CS2SMOS | APP-x |
|---|---|---|---|---|---|---|---|---|
| **Mooring A** | **2011–2016** | 1.23 | 1.74 | 1.5 | 1.61 | 1.74 | 1.4 | 1.99 |
| | **2011** | 1.15<br>-0.8 | 1.66<br>-0.07 | 1.4<br>-0.1 | 1.45<br>-0.16 | 1.5<br>-0.24 | 1.43<br>0.03 | 2.01<br>0.02 |
| | **2012** | 1.39<br>0.16 | 1.66<br>-0.08 | 1.43<br>-0.07 | 1.42<br>-0.19 | 1.68<br>-0.06 | 1.2<br>-0.2 | 2.04<br>0.05 |
| | **2013** | 1.05<br>-0.18 | 1.82<br>0.08 | 1.51<br>0.01 | 1.61<br>-0.01 | 1.48<br>-0.26 | 1.27<br>-0.13 | 2.02<br>0.03 |
| | **2014** | 1.51<br>0.28 | 1.98<br>0.24 | 1.71<br>0.21 | 1.85<br>0.24 | 2.09<br>0.35 | 1.68<br>0.28 | 1.96<br>-0.03 |
| | **2015** | 1.32<br>0.1 | 1.9<br>0.16 | 1.73<br>0.23 | 1.74<br>0.12 | 2.09<br>0.36 | 1.69<br>0.29 | 1.99<br>0.0 |
| | **2016** | 1.2<br>-0.03 | 1.44<br>-0.3 | 1.3<br>-0.21 | | 1.67<br>-0.07 | 1.21<br>-0.19 | 1.91<br>-0.08 |
| | **2017** | 0.96<br>-0.26 | 1.72<br>-0.02 | 1.43<br>-0.07 | | 1.66<br>-0.08 | 1.33<br>-0.07 | |
| | | ULS | CPOM | AWI | JPL | GSFC | CS2SMOS | APP-x |
| **Mooring B** | **2011–2016** | 1.36 | 1.88 | 1.68 | 1.79 | 1.84 | 1.65 | 2.01 |
| | **2011** | 1.42<br>0.06 | 1.9<br>0.02 | 1.61<br>-0.07 | 1.61-0.17 | 1.61<br>-0.22 | 1.64<br>-0.01 | 2.03<br>0.02 |
| | **2012** | 1.4<br>0.04 | 1.89<br>0.02 | 1.67<br>0.0 | 1.72<br>-0.07 | 1.7<br>-0.13 | 1.65<br>0.0 | 2.06<br>0.05 |
| | **2013** | 1.2<br>-0.16 | 2.04<br>0.17 | 1.77<br>0.09 | 1.83<br>0.04 | 1.75<br>-0.09 | 1.68<br>0.04 | 2.03<br>0.02 |
| | **2014** | 1.47<br>0.11 | 2.02<br>0.14 | 1.77<br>0.1 | 1.85<br>0.06 | 2.09<br>0.25 | 1.820.17 | 1.98<br>-0.04 |
| | **2015** | 1.55<br>0.19 | 2.02<br>0.15 | 1.83<br>0.16 | 1.92<br>0.14 | 2.06<br>0.22 | 1.74<br>0.1 | 2.02<br>0.0 |
| | **2016** | 1.26<br>-0.1 | 1.55<br>-0.33 | 1.47<br>-0.21 | | 1.95<br>0.12 | 1.42<br>-0.23 | 1.96<br>-0.05 |
| | **2017** | 1.21<br>-0.15 | 1.71<br>-0.17 | 1.61<br>-0.06 | | 1.69<br>-0.15 | 1.59<br>-0.06 | |
| | | ULS | CPOM | AWI | JPL | GSFC | CS2SMOS | APP-x |
| **Mooring D** | **2011–2016** | 1.37 | 1.88 | 1.62 | 1.81 | 1.81 | 1.52 | 2.08 |
| | **2011** | 1.48<br>0.11 | 1.7<br>-0.18 | 1.48<br>-0.15 | 1.54<br>-0.27 | 1.66<br>-0.15 | 1.44<br>-0.09 | 2.15<br>0.07 |
| | **2012** | 1.39<br>-0.02 | 1.75<br>-0.13 | 1.51<br>-0.11 | 1.6<br>-0.21 | 1.66<br>-0.14 | 1.41<br>-0.11 | 2.15<br>0.06 |
| | **2013** | 1.06<br>-0.31 | 2.06<br>0.17 | 1.51<br>-0.11 | 1.59<br>-0.21 | 1.55<br>-0.26 | 1.28<br>-0.25 | 2.16<br>0.08 |
| | **2014** | 1.64<br>0.27 | 2.12<br>0.23 | 1.89<br>0.26 | 2.13<br>0.33 | 2.15<br>0.34 | 1.86<br>0.34 | 2.09<br>0.01 |
| | **2015** | 1.59<br>0.22 | 2.13<br>0.25 | 1.97<br>0.35 | 2.16<br>0.36 | 2.26<br>0.45 | 1.93<br>0.4 | 2.0<br>-0.08 |
| | **2016** | 1.31<br>-0.06 | 1.61<br>-0.27 | 1.42<br>-0.2 | | 1.67<br>-0.14 | 1.26<br>-0.27 | 1.95<br>-0.14 |
| | **2017** | 1.13<br>-0.24 | 1.82<br>-0.07 | 1.57<br>-0.05 | | 1.69<br>-0.11 | 1.49<br>-0.03 | |

**Table 5. Winter (October-April) sea ice thickness growth rate (md$^{-1}$) for the period 2010-2018 in the central Arctic (regions 1-6), where rates are calculated based on product availability during each growth season.**

| Growth Rate (m/d) | CPOM | AWI | JPL | GSFC | CS2SMOS | APP-x |
|---|---|---|---|---|---|---|
| 2010-2011 | | | | 0.0032 | | 0.0103 |
| 2011-2012 | 0.0064 | 0.0058 | 0.0057 | 0.0040 | 0.0050 | 0.0105 |
| 2012-2013 | 0.0055 | 0.0048 | 0.0052 | 0.0027 | 0.0044 | 0.0108 |
| 2013-2014 | 0.0058 | 0.0049 | 0.0054 | 0.0026 | 0.0042 | 0.0100 |
| 2014-2015 | 0.0057 | 0.0051 | 0.0053 | 0.0037 | 0.0044 | 0.0103 |
| 2015-2016 | 0.0051 | 0.0050 | | 0.0025 | 0.0040 | 0.0099 |
| 2016-2017 | 0.0050 | 0.0041 | | 0.0022 | 0.0034 | |
| 2017-2018 | 0.0047 | 0.0042 | | 0.0014 | | |