# Peer review of "Figure S1: Sea ice thickness deviation (in meters) for each product subtracted from reference data set (CPOM), for (a) October–November, and (b) March–April, in the central Arctic (regions 1–6), for the period 2011–2017."

_The Cryosphere, 2018_

## Short Comment (SC1) · 26 Sep 2018

S. Hendricks

stefan.hendricks@awi.de

The paper "Assessment of Contemporary Satellite Sea Ice Thickness Products for Arctic Sea Ice" is an interesting study on the consistency of different Arctic sea-ice thickness products from CryoSat-2 and other Earth Observation data sets. I however want to shortly point out a potential irregularity that I have noticed in Figure 5. This figure shows the increase of mean thickness in the Arctic Basin for the different products and I cannot reproduce the growth curves of the merged CryoSat-2/SMOS (cs2smos) product as shown in Figure 5. While the general magnitude seems correct, there is significant noise on the cs2smos curve with irregular decreases of monthly mean thicknesses in the Winter growth season.

[Figure]

I do not observe this behaviour when I compute mean Arctic Basin sea ice thickness with the weekly merged CryoSat-2/SMOS thickness product (version 1.4) and it also has not been observed in previous publications (e.g. Ricker et al, 2017). I have attached two figures illustrating my version of the mean thickness curve and the respective region definition for the Central Arctic Basin. My suggestion to the authors is to verify their methodology and amend the figure and conclusions derived from it if needed.

Best Regards, Stefan Hendricks

Ricker, R., S. Hendricks, F. Girard-Ardhuin, L. Kaleschke, C. Lique, X. Tian-Kunze, M. Nicolaus, and T. Krumpen (2017), Satellite-observed drop of Arctic sea ice growth in winter 2015–2016, Geophys. Res. Lett., 44, 3236–3245, doi:10.1002/2016GL072244.
* * *
**Arctic Basin**

Fig. 1. Mask for the Arctic Basin (green area indicates area for computing mean sea-ice thickness)

[Figure]

**Fig. 2.** Mean sea-ice thickness time series in the Arctic Basin from merged CryoSat-2/SMOS data (version 1.4)

---

## Short Comment (SC2) · 26 Sep 2018

I also note a similar discrepancy in Figure 5 for the GSFC product. In my plotting there is a difference in both the overall magnitude and month to month change. I'm not sure what the difference could be due to at a quick glance.

---

## Short Comment (SC3) · 27 Sep 2018

The website in table 1 for CPOM does not work (http://www.cpom.ucl.ac.uk/csopr/seaice.html). It is likely it is the CPOM server that is down. But I wanted to raise the issue here in case the url has a typo, or there is an alternative source for this dataset that can be provided.

---

## Referee Comment (RC1) · J.C. Landy (Referee) · 9 Oct 2018

Assessment of Contemporary Satellite Sea Ice Thickness Products for Arctic Sea Ice
Sallila et al., 2018

This study provides the first genuine inter-comparison of pan-Arctic sea ice thickness data products obtained from three different satellite sensors. It is systematic in approach, equitably comparing the datasets to identify seasonal and interannual patterns of ice thickness variability, as well as assessing biases versus independent reference data. Results from the assessment will prove to be a valuable contribution to the field of Arctic climate science and potentially to operational stakeholders.

However, I have some concerns about the robustness of methodology used to process

and compare datasets. These concerns focus on the procedure used to resample data, as described in the comments below. I have also made a few suggestions to improve the analysis and impact of the author's findings. I'd recommend this manuscript is published in The Cryosphere, following these revisions.

Please do get in contact if you have questions regarding these comments. Kind regards, Jack Landy

General comments:

1. The method used to re-grid the satellite products may be introducing errors into the data inter-comparison and artificially improving the correlations. By gridding to 5 km with a search radius of 50 km you are introducing significant spatial autocorrelation between adjacent grid cells (through your method cells can theoretically only be treated as independent samples at a length scale 25 km). It is not statistically robust to be calculating the pixel-wise correlation between spatially-dependent samples, and this may be artificially improving the coefficient and significance. Perhaps you could try recalculating the correlation coefficient for a down-sampled 25-km version of your current grid? Take a look at this paper https://www.cambridge.org/core/journals/annals-of-glaciology/article/impact-of-spatial-aliasing-on-seaice-thickness-measurements/AB5DB924171B219D25E64A829E418840 to consider your gridding approach. (This is also worth considering for your OIB comparisons in Fig 9).

2. Linked to the above comment, you suggest at Page 14, Line 19 that the satellite products do not capture the thinnest or thickest ice because of the grid-averaging process. It surely makes sense then, for a fair comparison between satellite obs and the ULS, to average the ULS measurements over some window, rather than binning individual obs in your histograms. I understand this is difficult because it means averaging over time and comparing it to grid averages over space. However, could you identify a time-averaging window based on recorded ice drift speeds (either a seasonally-representative drift speed or obtained from satellite ice motion data)?

3. Your correlations between all the CS2 products are understandably very high (Page 12, Line 12) and there's not too much new analysis here on the differences between the CS2-only datasets. I would be very interested to know how closely each product picks up small-scale spatial variability in ice thickness. For instance, you could calculate the anomaly maps of each year compared to the climatological average, then assess the pixel-wise inter-product correlations between positive and negative anomalies. Do some of the CS2 products identify more or less of the smaller-scale thickness features than others? Are they realistic? This analysis would improve e.g. the first paragraph of the discussion.

4. There is not a lot of analysis or discussion of regional variations in the products: whether any obvious differences are regionally-dependent etc., except from Table 2. Can you add some deeper discussion on regional differences in the products, and offer some interpretation as to why these differences may exist? This would be a really valuable addition to the work.

Minor comments/edits:

Page 2. Lines 15-16. By what mechanism does Laxon 13 speculate that lower ice volume could be a factor in recent minima in ice extent?

P2. L29-31. Can you briefly introduce here some of the main differences in approach?

P3. L17-18. This is confusing: you're not including the IECSat data in your evaluation right, so why do you discuss evaluating ICESat here? Do you mean they have been used as a tool for evaluating satellite products in the past?

P4. L6. What is the time period under investigation?

P4. L8. What do you mean by mean scattering horizon?

P4. L30+. I understand that estimates for snow depth are pretty similar between the CS-only datasets, but you do not refer to them here and they are critical. Can you add some info on how snow depth and density are estimated for each product?

P6. L16-18. The description of the waveform-fitting method used for the GSFC product needs to be improved. The physical model of Kurtz 14 does not calculate expected returns empirically. Their model is semi-analytical, producing a simulation of the expected echo shape based on statistical parameterizations of the sea ice backscattering properties and height distribution of surface roughness.

P7. L26-27. What is the physical relationship between the surface EB and ice thickness?

P7. L30-31. Why does sea ice roughness affect the surface EB?

P8. L 5. How is snow depth estimated in the model?

P9. L2. What measurement of uncertainty is that? Bias? RMSE?

P9. L20. It is unconventional to subtract the focus product from the reference – typically it's the other way around. This makes the plots e.g. Fig 3 harder to interpret, because you have products with higher thickness than the CPOM data given as negative numbers.

P10. L3. Why do you pick a 200 km radius? Is this an arbitrary threshold or based on estimated ice drift displacements over the season? Do you use the same threshold in winter and summer? Have you tried varying the threshold +/- 100 km to see if you get the same results?

P10. L9. So, one correlation coeff for the whole time period, using each monthly average as a single sample?

P10. L17. Above you refer to the issue of oversampling at the start or end of a seasonal period and then treating products as the same when comparing. The IceBridge spring data are collected in discrete campaigns, so the period of observations will often not be spread evenly over the March-April period. This seems like the same issue – so how might this be introducing uncertainty into your satellite-OIB product comparisons?

P11. L4. Can you give the mean and +/- (between the techniques) of the fall 2011/2012 ice thickness, to provide the reader with a bit more context? Here and in other similar places within the paper?

P11. L18-20. Is there a way you can quantify this? e.g. by calculating the spatial covariance of the gridded data between years, or by simply calculating the average pixel standard deviation in ice thickness across the period? i.e. how sensitive are each of the products to spatial features of ice thickness like the thick ice tail in the Beaufort Gyre, at what scales?

P12. L3-4. For which season?

P13. L 14. And indeed of the thicker sea ice too... It's very strange that the technique based on a thermodynamic model and surface EB suggests highest ice growth rates at the end of winter, when you'd expect such a method to be better at capturing the expected reduction in thermodynamic growth.

P13. L15-34. It is very interesting that the CS2 only products are more similar in spring than in fall, when intuitively you'd expect them to diverge more when the ice is thicker. Can you provide any insight on why this might be? In what ways may the processing differences induce variations in fall that are not apparent in spring?

P14. L 34. What do you mean by 'varying dependency to draft'?

P15. L2-3. Since the thickness is relatively low in this region (<1.5 m) this would imply a higher contribution of SMOS data to the optimal interpolation, right? So does this give you any information about the reliability of SMOS data?

P16. L1-15. It is also worth noting that GSFC freeboard for both fall and spring is generally lower than the AWI freeboard, whereas the GSFC ice thickness was determined as the highest in Figs 2-3 etc. So the positive bias (whether an error or not) in GSFC ice thickness compared to CPOM and AWI is not coming from the freeboard measurements, and must be from the conversion to ice thickness (snow depth, snow

or ice density differences, or perhaps filtering/processing chain differences).

P17. L7. 'freeboards of less than approximately 0.05 m'.

P17. L 23. You suggest the assumption is that SMOS data is too heavily weighted in the combined product, but where do you get this assumption from? Are you making this assumption, or have you got it from previous papers? (if so which ones?). Potentially the uneven thickness increase detected by SMOS is more realistic and the CS2 products cannot detect new thin ice..?

P18. L 5-8. Re. variations in ice concentration thresholds, you could have defined your own conservative threshold (e.g. 75

P19. Snow section. This is quite out of place here, as you do not pay much attention to variations in snow treatment between products, and the focus of the paper is on intercomparison rather than the systematic issue of estimating snow (common between all products). I suggest removing this passage and if desired, reference the challenge in a single sentence or so.

Table 1. This is great! Really useful compilation.

Fig 5. Can you add the linear per winter ice growth rate for each product and year to the plot? i.e. X cm/month.

Fig 7. Can you emphasize more clearly on the plot that the bold distribution is ice draft, as a quick glance at the figure gives one the instant conclusion that CS2SMOS data must be best.

Fig 8. Can you add the x = 0.9y line to the plot as a reference?
* * *

---

## Short Comment (SC4) · 10 Oct 2018

This was indeed due to the website being down. It seems to be fixed now.

---

## Referee Comment (RC2) · T. W. K. Armitage (Referee) · 17 Oct 2018

**Review of "Assessment of Contemporary Satellite Sea Ice Thickness Products of Arctic Sea ice" by Sallila et al.**

The authors present an intercomparison of several different sea ice thickness data products, principally derived from CryoSat-2, and compare these datasets against BGEP upward-looking sonar draft data and Operation IceBridge airborne thickness data. Unsurprisingly, they find good agreement between the three CS2 products which use empirical retrackers (CPOM/AWI/JPL), while the GSFC data appears to overestimate the mean thickness early in the season and shows quite different spatial variability, and the CS2SMOS data potentially underestimates the thickness throughout the season. The analysis is straightforward, presented clearly enough and, while there are no new science results, I believe this paper could be useful to the community. I'm happy to recommend publication once my points below have been addressed.

I have a few suggestions that I think will improve the paper:

1.  Introduction: I found the introduction cumbersome. I would combine the first two paragraphs into one, and reduce the length by ~half as some of the material you discuss here is not really relevant for the current manuscript. Most of the fifth paragraph (Page 3, lines 12-27) is unnecessary in the intro, particularly as you essentially repeat it all in section 2, so remove – the first sentence can go at the end of the previous paragraph, and combine this with the final intro paragraph.

2.  Why don't you use the EM bird data? I believe it is a fairly extensive dataset, is commonly used by groups to assess CS2 thickness data, and is the only technique that directly measures thickness rather than a proxy (draft/freeboard).

3.  You do not discuss the impact of the different treatments of snow on your product intercomparison. There is a paragraph on the general difficulty of snow in sea ice altimetry in the conclusions (Page 19, lines 14-34) – I don't think this belongs here, I would move it to the discussion – but I would also like to see some discussion on how the different snow treatments impact your results. Importantly, you say that all the CS2 data use the 'modified' Warren climatology (Table 1), however I believe that the CPOM processing uses the basin-mean value of the Warren snow depth over the central Arctic region for MYI and half this value for FYI (at least this is my understanding from Tilling et al. (2018)), whereas I believe AWI/JPL use the spatially varying 'modified' Warren snow depth. I don't know if you realized this but it is not clear in the manuscript? [It might be worth checking with Tilling et al.] This could explain some of the differences in Figure 2 (AWI/JPL overestimate thick ice, underestimate thin ice relative to CPOM) as well as why the AWI and JPL maps look quite similar (they treat snow the same, but different to CPOM), and some of the differences in the in-situ comparison.

4.  The APP-x data is very poor, and I don't really see what value it brings to the paper, other than to say that it is very poor (you could be more unequivocal about this). It

shows almost no interannual or spatial variability, and you find that the mean thickness grows by 1m between Feb and Apr (page 13 line 10) which is physically unbelievable. After showing the APP-x data in Figures 2 and 3, maybe Figure 4, I would consider dropping it from the rest of the manuscript and say in your conclusions that it is unrealistic and shouldn't be used over a CS2 product. I was very surprised that you suggest (page 19, line 8) users should use the APP-x data simply because it is available daily in NRT and Arctic-wide! Judging from your analysis, given that APP-x does not capture interannual or spatial variability, users would be better off simply using the monthly CS2 climatology because at least it captures the spatial variability and does not grow unrealistically thick ice at the end of winter.

Minor comments:

It's not clear what you mean by "contemporary" in the title – occurring at the same time, or occurring in the present? Either way I don't think it's necessary; consider revising: "An assessment of satellite-derived Arctic sea ice thickness data"

Page 1, line 19-20: Should read "Among the data compared, the blended…"

Page 2, line 26: Re-write: "The most widely-used thickness datasets are derived from the radar altimeter…"

Page 2, line 29: "..retrack*ing*…"

Page 2, line 32: "…*basin-scale* gradients…"

Page 3, line 4: "Given the variety of *sea ice thickness* data…"

Page 4, line 8: You should reference Tilling et al. (2018) as well as Laxon, as the data you are using was produced by Tilling et al and is somewhat different from the original Laxon data.

Page 5, line 9, Table 1: The CPOM processing uses separate retrackers for leads (Gaussian+Exponential model fit) and floes (threshold), and they apply a correction to account for this.

Page 5, lines 24-29: I don't think it is true to say that the AWI processing "does not differ significantly" from the CPOM processing. They are 'similar', but a number of important differences I can think of from the top of my head: 1) different retracking approaches (I believe AWI apply a 40% or 50% threshold retracker to all waveforms), 2) different sea level interpolation, 3) different waveform discrimination criteria (e.g., right and left sided peakiness), the MSS which you mention, different treatment of snow (see my comments above).

Page 6, line 9: I think Kwok and Cunningham (2015) is missing from the bibliography? Please check all references in the text actually appear in the bibliography.

Section 3.1: Calculating a pixel-by-pixel correlation after oversampling the data to 5km doesn't make sense, as it may artificially increase the correlation stats. I would suggest sampling all the data onto an identical 25km grid for the analysis. Also, why use a 50km search radius for the interpolation when the data are posted on 25km grids? Surely this will act to smooth the data, also improving the correlation?

Page 9, line 13: You should provide some justification for using the CPOM data as your baseline, it seems rather arbitrary.

Page 11, lines 5-9: The phrasing here jarred a bit, as it suggests that the "cooler summer = thicker ice in 2013/14" result is a "finding" of yours which was "also noted" by Tilling et al. (2015). Obviously, you haven't shown any linkages between temperature and thickness, and I know you didn't mean to imply this, so please rephrase this section to say that "Tilling et al found that... The thickness changes in Figure 2 are consistent with their result."

Page 11, line 25: The AWI CS2 data appears to show extensive thin ice (<0.5m) in fall but this is the noise floor of CS2, so not necessarily believable – is there something in the AWI algorithm that causes this? I would like some more discussion/explanation of this.

Page 14, lines 1-10, Table 3: You should calculate the anomalies relative to the same baseline period to make the numbers comparable i.e., 2011-2015 considering this is the common baseline.

Page 14, line 16: "CS2-only products" – judging from figure 7, *all* of the satellite products miss the thickest/thinnest ice including CS2SMOS?

Page 14, lines 20-21: by what measure does the AWI product most closely align with the ULS draft? Likewise, by what measure does the GSFC data least agree? I would say the CPOM/AWI/JPL show the same agreement from figure 7?

Section 4.5, paragraph 1, Figure 10: I would think the negative freeboards in the GSFC data must have something to do with the different retracking, as this is the major difference between the data products? I would speculate that the GSFC floe retracking might be sensitive to off-nadir scattering, as they use a functional fit to the entire waveform and power in the trailing edge could skew the fit to later delay times.

---

## Referee Comment (RC3) · R. Ricker (Referee) · 4 Nov 2018

Review of Assessment of Contemporary Satellite Sea Ice Thickness Products for Arctic Sea Ice by Sallila et al.

**Summary**

The paper "Assessment of Contemporary Satellite Sea Ice Thickness Products for Arctic Sea Ice" evaluates and compares 6 different sea ice thickness data products that are publicly available, including data from CryoSat-2 (CS2), SMOS and the Advanced Very-High-Resolution Radiometer (AVHRR). For the evaluation, they use independent ice draft and thickness measurements obtained from the Beaufort Gyre Exploration Project (BGEP) upward looking sonars (ULS) and Operation IceBridge. The authors find that products utilizing CS2-only measurements are reliable for sea ice thickness between ~0.5 m and 4 m, while the merged CS2-SMOS product was the most reliable for thin ice. In contrast to the other products, the AVHRR dataset does not seem to represent a reliable ice thickness distribution at the end of the winter season.

I think this study is potentially very useful for the user community of those sea ice thickness data sets, since so far, such a review of current sea-ice thickness data sets is not existing to my knowledge. It is well written, has a clear structure and generally is easy to follow. It presents useful information bundled in one paper. Table 1, for example, is great as an overview. However, I also find that the paper lacks some crucial information regarding the products and the way the data are analyzed. I also have the feeling that there are some inconsistencies in the data analysis. My 3 major concerns are:

1. I find that the description of the sea-ice thickness data sets in section 2.1 is incomplete. The AWI and the CPOM processing is quite different in some aspects, e.g. retracking of ice and lead waveforms. I think this should be described in more detail. See therefore also my comments below.

2. I am not really sure about the way the original data are re-gridded for the comparison. The paper lacks motivation regarding the chosen values for the re-gridding resolution of 5 km and the search radius of 50 km. I am also not sure if this is really the best way to do it, see detailed comments below. In any case, I suggest to revise section 3.1 in order to motivate your chosen method.

3. It also seems strange that CS2SMOS underestimates ice thickness in the MYI zone compared to the AWI product. CS2SMOS uses the AWI product for merging with SMOS, but in MYI areas, SMOS should not have a significant impact. See also my detailed comments below.

**Detailed Comments**

P1 L18-19: "… are reliable for sea ice [thickness] between ~0.5 m and 4 m" -  it should be mentioned that you are talking about thickness.

P2 L26: „One of the most widely used thickness data sets derives from the radar altimeter flown on CryoSat-2 …" - this sentence sounds a bit odd to me.

P4 L11-12: Please mention the version of the CS2SMOS weekly ice thickness product.

P5 L8-9: I don't think that the CPOM processor uses the TFMRA algorithm (AWI). They also use a different retracking method for lead returns: a gaussian function is fitted to the waveform, as far as I know. See therefore, Tilling et al. (2017) - „Estimating Arctic sea ice thickness and volume using CryoSat-2 radar altimeter data".

[Figure]

*Figure R1: CS2SMOS Analysis Thickness from 20150323-20150329, version 1.3.*

P5 L18: Please provide the version number of the AWI product. Is it 1.2?

P5 L18-28: Here it should be mentioned that the TFMRA is used. Moreover, I do not agree with the statement „The algorithm employed by AWI does not differ significantly from that used in the derivation of the CPOM product". While I agree that there are some similarities in the processing and also the derived ice thickness distributions look similar, some signifiant differences exist. For example, AWI uses the same retracking for lead and sea ice echoes, while CPOM does not (see above). Moreover, the surface type classification is different. CPOM also applies a retracker bias correction (Tilling et al. (2017).

P7 L16: What is the reason to include the APP-x data here? They do not seem to represent the entire Arctic sea ice thickness distribution and ice growth during the season (see Figure 2 a,b). Then, one could argue to also include the SMOS sea ice thickness product.

P9 L9-11: Why do you choose a 5 km grid and a 50 km search radius? What happens at the ice edges or at the coasts. For example, areas, where the original grid contains NaN's in case of open water. Does this mean that you obtain ice thickness estimates on your new grid, where there were NaN's before, if valid ice thickness grid cells are found within the 50 km search radius? Wouldn't that erroneously enlarge the ice area?
If there are more than one grid cell found within the search radius, are you only considering the closest one? Or is there some weighting applied?

P10 L3: Why do you choose such a large radius (200 km)? Especially in the Beaufort Sea, where those moorings are deployed, you may have very mixed ice regimes with some chunks of MYI surrounded by FYI. Wouldn't it be better to only choose the nearest one or at least a smaller radius?

P10 L16: Why are the IceBridge data interpolated on a 50 km grid? Why not 5 km as the re-gridded satellite products?

P11L15-16: I am a bit confused about the CS2SMOS ice thickness maps. Here, it seems that they consequently underestimate ice thickness. Especially in MYI areas, the thickness should be very similar to the AWI CS-2 Product, since this is used for the data merging, and SMOS data should have almost no impact in the MYI zones. When I plot the CS2SMOS thickness for the week 2015-03-23 - 2015-03-29 (see **Figure R1,** data from Meereisportal, version 1.3), using a similar color scale with the switch from light blue to yellow at 2m, the MYI tail in the Beaufort Sea appears yellow, indicating thickness above 2 m. In your map it seems to be below 2m. Of course, your map shows the average from March - April, but I would assume that this is not much different.

Figure 2: I really would recommend to use a different color scale with a linear color gradient, starting at 0.0 m. Moreover I would also suggest to use a finer resolution, e.g. 0.25 m. 0.5 m is too coarse from my point of view.

P12 L12-14: As mentioned above, it seems a bit strange that CS2SMOS is significantly thinner than AWI CS-2 in the MYI regions, since it uses the AWI CS-2 data, while the SMOS ice thickness estimates are not valid in MYI areas.

P17 L2-3: See above.

Figures 5: The CS2SMOS data seem very noisy and show some strange behavior, e.g. strong decreases in mean ice thickness in some months. This should be checked. It does not seem to occur in Ricker et al (2017): "Satellite–observed drop of Arctic sea ice growth in winter 2015–2016". Although they show sea-ice volume, I would expect similar behavior.

Figure 5 and 6: Over which area are these averages calculated? Regions 1-6 as indicated in Figure 1?

Figure 6: "a)" and "b)" are missing in the figure.

---

## Author Comment (AC1) · 30 Jan 2019

Thank you for the comment. This was indeed due to the CPOM website being down at the time it was accessed by the reviewer. A response was shortly after provided in SC4, by Dr. Rinne.

———————————————

---

## Author Comment (AC2) · 30 Jan 2019

Thank you for the comment. This was indeed due to the website being down. It seems to be fixed now.
* * *

---

## Author Comment (AC3) · 30 Jan 2019

**Response to Short Comment 1 (Stefan Hendricks)**

The paper "Assessment of Contemporary Satellite Sea Ice Thickness Products for Arctic Sea Ice" is an interesting study on the consistency of different Arctic sea-ice thickness products from CryoSat-2 and other Earth Observation data sets. I however want to shortly point out a potential irregularity that I have noticed in Figure 5. This figure shows the increase of mean thickness in the Arctic Basin for the different products and I cannot reproduce the growth curves of the merged CryoSat-2/SMOS (cs2smos) product as shown in Figure 5. While the general magnitude seems correct, there is significant noise on the cs2smos curve with irregular decreases of monthly mean thicknesses in the Winter growth season.

I do not observe this behaviour when I compute mean Arctic Basin sea ice thickness with the weekly merged CryoSat-2/SMOS thickness product (version 1.4) and it also has not been observed in previous publications (e.g. Ricker et al, 2017). I have attached two figures illustrating my version of the mean thickness curve and the respective region definition for the Central Arctic Basin. My suggestion to the authors is to verify their methodology and amend the figure and conclusions derived from it if needed.

Best Regards, Stefan Hendricks

Ricker, R., S. Hendricks, F. Girard-Ardhuin, L. Kaleschke, C. Lique, X. Tian-Kunze, M. Nicolaus, and T. Krumpen (2017), Satellite-observed drop of Arctic sea ice growth in winter 2015–2016, Geophys. Res. Lett., 44, 3236–3245, doi:10.1002/2016GL072244.

[Figure]

**Fig. SC1.1** Mask for the Arctic Basin (green area indicates area for computing mean sea-ice thickness)

[Figure]

**Fig. SC1.2** Mean sea-ice thickness time series in the Arctic Basin from merged CryoSat-2/SMOS data (version 1.4)

We thank Dr. Hendricks for this comment and for catching this issue. The peculiarities identified within the CS2SMOS data product were due to an error that occurred during the data transfer process at our end. Upon review, we found that many of the CS2SMOS product data files were only partially complete. We have downloaded the CS2SMOS data product again and we can now confirm we get the same results as you provide (see Figure SC1.3). We have revised all of the tables and figures throughout the manuscript so that they now contain the updated and complete CS2SMOS data product.

[Figure]

Fig. SC1.3 Comparison of weekly CS2SMOS thickness data (as provided in SC1. 2) overlayed on the monthly averaged CS2SMOS ice thickness data, based on the revised version of the CS2SMOS data product now used throughout the manuscript. The monthly averages are now fully consistent with the weekly CS2SMOS data as shown in SC1.2. Monthly data are averaged over the central Arctic (regions 1-6), consistent with the region shown in Figure SC1.1.

---

## Author Comment (AC4) · 1 Feb 2019

**Response to Short Comment 2 (Nathan Kurtz)**

I also note a similar discrepancy in Figure 5 for the GSFC product. In my plotting there is a difference in both the overall magnitude and month to month change. I'm not sure what the difference could be due to at a quick glance.

Thank you for this comment. Based on the response to SC1, we have revised Figure 5. We also conducted an extensive review of the GSFC product, to examine both the magnitude of the monthly means, and the month to month changes.

We examined 5 instances within the GSFC ice thickness time series where average monthly ice thickness decreased during the winter growth period: October 2010 - November 2010; December 2015 - January 2016; March 2016 - April 2016; October 2016 - November 2016; and March 2018 - April 2018 (see Figure 5). We examined 1 instance within the GSFC ice thickness time series where average monthly ice thickness increased rapidly during the winter growth period: December 2012 - February 2013 (see Figure 5). Below we provide our analysis of two of these six events: March 2018 - April 2018 (Figure SC2.1) and October 2016 - November 2016 (Figure SC2.2). These are illustrative of all 6 events noted above. We then provide our overall conclusion of the analysis at the end of this comment.

Analysis of Fig. SC2.1: For regions 1-6 there is no substantial difference between the GSFC sea ice thickness for March and April 2018 as directly downloaded from product provider's data source (a) and (b), versus the GSFC sea ice thickness data used in our analysis (c) and (d). This rules out any corrupt data in our GSFC data product files as was the case with the CS2SMOS data product, as described in our response to SC1 above.

Maps of average ice thickness for March 2018 (c) and April 2018 (d) in the GSFC product indicate a decrease in the areal extent of the thickest ice between March and April 2018, which was replaced with a larger area of thin ice in April 2018, particularly evident in region 1, north of regions 4 and 5. Conversely in the AWI data product, the maps show an overall thickening of the ice between March 2018 (e) and April 2018 (f) across all regions 1 - 6. The is a decline in average ice thickness between March 2018 (g) and April 2018 (h) in the GSFC product of 0.072 meters, while there is a corresponding increase in average ice thickness in the AWI data product between March 2018 (i) and April 2018 (j) of 0.098m. Modal ice thickness in the GSFC product is 1.875 m in March 2018 (g) and decreases to 1.625 m in April 2018 (h).

The decline in ice thickness in the GSFC product is associated with a reduction in the number of observations of ice > 2 m thick, and a corresponding increase in the number of observations of ice < 2 m thick, between March and April 2018. The total number of observations with ice > 2 m thick is 4721 in March 2018 (g) and 3752 in April 2018 (h) for the GSFC data product. The total number of

valid ice thickness observations in the GSFC data product remains stable between March and April 2018, illustrating that the decline in ice thickness is not related to a decrease in the total ice extent within the data product.

Analysis of Fig. SC2.2: For regions 1-6, there is no substantial difference between the GSFC sea ice thickness for October and November 2016 as directly downloaded from product provider's data source (a) and (b) versus the GSFC sea ice thickness data used in our analysis (c) and (d). This again rules out any corrupt data in our GSFC data product files.

Maps of average ice thickness for October 2016 (c) and November 2016 (d) in the GSFC product indicate a large increase in the areal extent of thin ice (

---

## Author Comment (AC5) · 19 Mar 2019

**Response to Review #3 (Robert Ricker)**

**Date: 20 March 2019**

We thank the Editor, reviewers and those who provided short comments on the manuscript for their inputs. The feedback has helped to improve both the clarity and content of the manuscript. We have provided responses to both the short comments and full reviews. We indicated by section, specified by paragraph, where revisions were made within the manuscript text. Since we include figures in response to both the short comments and the full reviews, there is a letter code to indicate if the figure is related to a short comment (e.g. SC1.1) or to a reviewer comment (e.g. RC3.1). The figures in the manuscript itself maintain normal numbering convention (e.g. Figure 1, Figure 2, etc.).

The following is a list of the major changes to the manuscript:

- We have revised the manuscript text for clarity and brevity. In particular we have shortened the Introduction and rearranged the text regarding the treatment of snow depth in each satellite data product.
- Based on input received during the review, we have revised the text of Section 2.1 and the information provided in Table 1 to clarify specific aspects of the processing chain and waveform retrackers used in each satellite thickness product.
- We have extended Table 1 so as to include the details of two additional satellite-derived sea ice thickness data sets, although these data are not included in the further analysis. This decision is a compromise between providing the pertinent details of publicly-available data products, while not overwhelming the reader with too much information in the figures and tables.
- We have replaced the CS2SMOS data set used in the original submission with an updated version of the data set, and revised all figures and tables containing CS2SMOS data.
- We have updated figures and tables wherever possible with new data that has become available since the original submission. In particular, we now include the BGEP ULS ice draft observations for the 2016-2017 season.
- We expanded our results to include winter growth rates, adding a new table (Table 5).
- The reviewers highlighted concerns regarding the original methods used to calculate the correlation between data products, and that using a near-neighbour interpolation with a search radius of 50 km could potentially artificially improve the correlation results. To address these concerns, we have revised the approach to calculate the correlation statistics between the satellite data products, as well as between the satellite and airborne observations. In the revised manuscript the thickness observations are placed onto a common grid (0.4° latitude by 4° longitude) before common grid cells are compared and correlation statistics calculated. This follows the approach originally taken in Laxon et al. (2013) as well as in subsequent studies and allows the reader to place our results in the context of the published literature. We note that this did not change the results of the correlation analyses in a substantial way.

Within the manuscript text, all edits (additions/deletions) are indicated in red font. The manuscript version indicating track changes is posted as a separate author's comment in order to keep the response document concise.

**Review of Assessment of Contemporary Satellite Sea Ice Thickness Products for Arctic Sea Ice by Sallila et al.**

**Summary**

The paper "Assessment of Contemporary Satellite Sea Ice Thickness Products for Arctic Sea Ice" evaluates and compares 6 different sea ice thickness data products that are publicly available, including data from CryoSat-2 (CS2), SMOS and the Advanced Very-High-Resolution Radiometer (AVHRR). For the evaluation, they use independent ice draft and thickness measurements obtained from the Beaufort Gyre Exploration Project (BGEP) upward looking sonars (ULS) and Operation IceBridge. The authors find that products utilizing CS2-only measurements are reliable for sea ice thickness between ~0.5 m and 4 m, while the merged CS2-SMOS product was the most reliable for thin ice. In contrast to the other products, the AVHRR dataset does not seem to represent a reliable ice thickness distribution at the end of the winter season.

I think this study is potentially very useful for the user community of those sea ice thickness data sets, since so far, such a review of current sea-ice thickness data sets is not existing to my knowledge. It is well written, has a clear structure and generally is easy to follow. It presents useful information bundled in one paper. Table 1, for example, is great as an overview. However, I also find that the paper lacks some crucial information regarding the products and the way the data are analyzed. I also have the feeling that there are some inconsistencies in the data analysis. My 3 major concerns are:

**RC 3.1**. I find that the description of the sea-ice thickness data sets in section 2.1 is incomplete. The AWI and the CPOM processing is quite different in some aspects, e.g. retracking of ice and lead waveforms. I think this should be described in more detail. See therefore also my comments below.

Thank you for pointing out that the AWI and the CPOM processing is quite different. Text in Sections 2.1.1 and 2.1.2 has been substantially revised to present additional details of the CPOM and AWI processing chain. We have also revised Table 1 to clarify the details of the CPOM retrackers.

**RC 3.2**. I am not really sure about the way the original data are re-gridded for the comparison. The paper lacks motivation regarding the chosen values for the re-gridding resolution of 5 km and the search radius of 50 km. I am also not sure if this is really the best way to do it, see detailed comments below. In any case, I suggest to revise section 3.1 in order to motivate your chosen method.

Thank you for your comment on gridding which is consistent with a comment from Reviewer 1 (see RC1.1). We have changed the gridding approach used to calculate correlation between data products and we have clarified this in the methods, Section 3.1. We have also revised all text where the results of the gridded data are discussed. Please see further details in our response to RC1.1.

**RC 3.3**. It also seems strange that CS2SMOS underestimates ice thickness in the MYI zone compared to the AWI product. CS2SMOS uses the AWI product for merging with SMOS, but

in MYI areas, SMOS should not have a significant impact. See also my detailed comments below.

See our response to SC1. We discovered that there was an issue related to data used in the first version of our manuscript, where some corrupt data files were used accidentally. The results for CS2SMOS have changed now that we are using the correct data product files, as shown in Figure RC3.2 below. In agreement with your comment, the thickness of the MYI region in the CS2SMOS data product now resembles that of the AWI CryoSat-2 product.

**Detailed Comments**

P1 L18-19: "... are reliable for sea ice [thickness] between ~0.5 m and 4 m" - it should be mentioned that you are talking about thickness.

Added "thickness" on page 1, Abstract.

P2 L26: "One of the most widely used thickness data sets derives from the radar altimeter flown on CryoSat-2 ..." - this sentence sounds a bit odd to me.

Altered this text.

P4 L11-12: Please mention the version of the CS2SMOS weekly ice thickness product.

To avoid being repetitive and keeping the introductory paragraph of 2.1 compact, we decided to include the details of the product versions in the product descriptions that appear in the subsections. Specifically, please see Section 2.1.5, where we added the version number (v1.3) for the CS2SMOS data product.

P5 L8-9: I don't think that the CPOM processor uses the TFMRA algorithm (AWI). They also use a different retracking method for lead returns: a gaussian function is fitted to the waveform, as far as I know. See therefore, Tilling et al. (2017) - "Estimating Arctic sea ice thickness and volume using CryoSat-2 radar altimeter data".

Thank you for pointing this out. This has been corrected in Section 2.1.1. and also in Table 1.

[Figure]

**Figure RC3.1**: CS2SMOS Analysis Thickness from 20150323-20150329, version 1.3.

[Figure]

Figure RC3.2: Response to comment RC 3.3 and Figure RC3.1: Revised mapping of sea ice thickness from the CS2SMOS data product, version 1.3, for the week 20150323-20150329. The revised map is consistent with Figure RC3.1 and illustrates that data in the files we are now using in the revised manuscript are consistent with those from the data provider.

P5 L18: Please provide the version number of the AWI product. Is it 1.2?

The version of the AWI data product used in our manuscript is 2.0. This detail was added to Section 2.1.2.

P5 L18-28: Here it should be mentioned that the TFMRA is used. Moreover, I do not agree with the statement "The algorithm employed by AWI does not differ significantly from that used in the derivation of the CPOM product". While I agree that there are some similarities in the processing and also the derived ice thickness distributions look similar, some significant differences exist. For example, AWI uses the same retracking for lead and sea ice echoes, while CPOM does not (see above). Moreover, the surface type classification is different. CPOM also applies a retracker bias correction (Tilling et al. (2017).

Thank you for this clarification and please also see our response to RC3.1. We have substantially revised the data product descriptions in Section 2 and we now include details of the TFMRA retracking algorithm in the second paragraph of Section 2.1.2.

P7 L16: What is the reason to include the APP-x data here? They do not seem to represent the entire Arctic sea ice thickness distribution and ice growth during the season (see Figure 2 a,b). Then, one could argue to also include the SMOS sea ice thickness product.

Regarding the APP-x data product, please see our response to RC2.4.

Regarding the SMOS-only sea ice thickness data product: While we agree that the SMOS data product differs to the ones presented in the manuscript, it is only reliable for studies of thin ice <1 m thick (Ricker et al., The Cryosphere, 11, 1607–1623, 2017). Thus it is not representative of overall ice thickness in regions 1-6 of the Arctic, which is the focus area of our study. Also, for the sake of readability we limit the comparison to six data products. We believe that by including the CS2SMOS data set in our analysis, which differs from the CS2-only products and highlights the most relevant aspects of the SMOS observations, it provides the most useful comparison and insight for the reader.

P9 L9-11: Why do you choose a 5 km grid and a 50 km search radius? What happens at the ice edges or at the coasts. For example, areas, where the original grid contains NaN's in case of open water. Does this mean that you obtain ice thickness estimates on your new grid, where there were NaN's before, if valid ice thickness grid cells are found within the 50 km search radius? Wouldn't that erroneously enlarge the ice area? If there are more than one grid cell found within the search radius, are you only considering the closest one? Or is there some weighting applied?

We agree with the Reviewer that the gridding approach used in the original version of the manuscript (a 5 km grid with a 50 km search radius) was not appropriate for comparing the data sets against each other. We examined the results of the original gridding approach and found issues similar to those you describe, in particular in the Central Arctic region (at the northern limits of region 1), and also around the ice edge. We have therefore changed our approach and now follow the method originally introduced in Laxon et al. (2013), and used again in Tilling et al. (2018), to conduct the product vs. product comparisons and to calculate the correlations (Figures 4 and 9). In the revised manuscript we use a 0.4° latitude by 4°

longitude grid. We have revised the relevant areas of the manuscript text (in particular, Section 3) to clarify this change. Please also see our response to RC1.1.

P10 L3: Why do you choose such a large radius (200 km)? Especially in the Beaufort Sea, where those moorings are deployed, you may have very mixed ice regimes with some chunks of MYI surrounded by FYI. Wouldn't it be better to only choose the nearest one or at least a smaller radius?

We used a radius of 200 km, since it is consistent with the original evaluation presented in Laxon et al. (2013) and subsequently in Kwok and Cunningham (2015). This allows the reader to place our results in the context of the conclusions of prior studies.

P10 L16: Why are the IceBridge data interpolated on a 50 km grid? Why not 5 km as the re-gridded satellite products?

We have revised our methodology to calculate the correlations between the satellite data products and the IceBridge data. In order to place our results in the context of the existing literature, we have adopted the same approach as that of Laxon et al. (2013), who used a 0.4° latitude by 4° longitude grid to compare IceBridge and CryoSat-2 thickness estimates. This approach accounts for the uneven spatial and temporal sampling of the sea ice along the IceBridge flight-lines compared to the monthly means obtained from the satellite data products.

P11 L15-16: I am a bit confused about the CS2SMOS ice thickness maps. Here, it seems that they consequently underestimate ice thickness. Especially in MYI areas, the thickness should be very similar to the AWI CS-2 Product, since this is used for the data merging, and SMOS data should have almost no impact in the MYI zones. When I plot the CS2SMOS thickness for the week 2015-03-23 - 2015-03-29 (see Figure R1 (now Figure RC3.1), data from Meereisportal, version 1.3), using a similar color scale with the switch from light blue to yellow at 2 m, the MYI tail in the Beaufort Sea appears yellow, indicating thickness above 2 m. In your map it seems to be below 2 m. Of course, your map shows the average from March - April, but I would assume that this is not much different.

Your confusion is well justified! Based on your remarks, and comment SC1, we re-examined the CS2SMOS data and found that the CS2SMOS dataset we had used in the original manuscript was incomplete, and some corrupted data files were accidentally used. For this revision, we have corrected the issue. We downloaded all of the CS2SMOS version 1.3 data from Meereisportal again, and we can confirm that the thickness data are now consistent with that shown in Figure RC3.1 (see Figure RC3.2 above, and Figure 2 in the manuscript).

Figure 2: I really would recommend to use a different color scale with a linear color gradient, starting at 0.0 m. Moreover I would also suggest to use a finer resolution, e.g. 0.25 m. 0.5 m is too coarse from my point of view.

We revised the colour bars used in Figures 2 and 3, to start at 0 m, and we changed the increment from 0.5 m to 0.25 m according to this suggestion.

P12 L12-14: As mentioned above, it seems a bit strange that CS2SMOS is significantly thinner than AWI CS-2 in the MYI regions, since it uses the AWI CS-2 data, while the SMOS ice thickness estimates are not valid in MYI areas.

We have corrected this. See response to SC1 and RC3.3.

P17 L2-3: See above.

We have corrected this. See response to SC1.

Figures 5: The CS2SMOS data seem very noisy and show some strange behavior, e.g. strong decreases in mean ice thickness in some months. This should be checked. It does not seem to occur in Ricker et al (2017): "Satellite-observed drop of Arctic sea ice growth in winter 2015–2016". Although they show sea-ice volume, I would expect similar behavior.

We have corrected this. See response to SC1.

Figure 5 and 6: Over which area are these averages calculated? Regions 1-6 as indicated in Figure 1?

Yes, the averages are calculated for the regions 1-6 as defined in Figure 1. While we had indicted this in the figure caption for Figure 6, we had omitted the information in the caption for Figure 5. We have thus modified the text of the figure caption for Figure 5.

Figure 6: "a)" and "b)" are missing in the figure.

We have added the pertinent letters to Figure 6.

---

## Author Comment (AC6) · 19 Mar 2019

**Response to Review #2 (Thomas Armitage)**

**Date: 20 March 2019**

We thank the Editor, reviewers and those who provided short comments on the manuscript for their inputs. The feedback has helped to improve both the clarity and content of the manuscript. We have provided responses to both the short comments and full reviews. We indicated by section, specified by paragraph, where revisions were made within the manuscript text. Since we include figures in response to both the short comments and the full reviews, there is a letter code to indicate if the figure is related to a short comment (e.g. SC1.1) or to a reviewer comment (e.g. RC3.1). The figures in the manuscript itself maintain normal numbering convention (e.g. Figure 1, Figure 2, etc.).

The following is a list of the major changes to the manuscript:
- We have revised the manuscript text for clarity and brevity. In particular we have shortened the Introduction and rearranged the text regarding the treatment of snow depth in each satellite data product.
- Based on input received during the review, we have revised the text of Section 2.1 and the information provided in Table 1 to clarify specific aspects of the processing chain and waveform retrackers used in each satellite thickness product.
- We have extended Table 1 so as to include the details of two additional satellite-derived sea ice thickness data sets, although these data are not included in the further analysis. This decision is a compromise between providing the pertinent details of publicly-available data products, while not overwhelming the reader with too much information in the figures and tables.
- We have replaced the CS2SMOS data set used in the original submission with an updated version of the data set, and revised all figures and tables containing CS2SMOS data.
- We have updated figures and tables wherever possible with new data that has become available since the original submission. In particular, we now include the BGEP ULS ice draft observations for the 2016-2017 season.
- We expanded our results to include winter growth rates, adding a new table (Table 5).
- The reviewers highlighted concerns regarding the original methods used to calculate the correlation between data products, and that using a near-neighbour interpolation with a search radius of 50 km could potentially artificially improve the correlation results. To address these concerns, we have revised the approach to calculate the correlation statistics between the satellite data products, as well as between the satellite and airborne observations. In the revised manuscript the thickness observations are placed onto a common grid (0.4° latitude by 4° longitude longitude) before common grid cells are compared and correlation statistics calculated. This follows the approach originally taken in Laxon et al. (2013) as well as in subsequent studies and allows the reader to place our results in the context of the published literature. We note that this did not change the results of the correlation analyses in a substantial way.

Within the manuscript text, all edits (additions/deletions) are indicated in red font. The manuscript version indicating track changes is posted as a separate author's comment in order to keep the response document concise.

**Review of "Assessment of Contemporary Satellite Sea Ice Thickness Products of Arctic Sea ice" by Sallila et al.**

The authors present an intercomparison of several different sea ice thickness data products, principally derived from CryoSat-2, and compare these datasets against BGEP upward-looking sonar draft data and Operation IceBridge airborne thickness data. Unsurprisingly, they find good agreement between the three CS2 products which use empirical retrackers (CPOM/AWI/JPL), while the GSFC data appears to overestimate the mean thickness early in the season and shows quite different spatial variability, and the CS2SMOS data potentially underestimates the thickness throughout the season. The analysis is straightforward, presented clearly enough and, while there are no new science results, I believe this paper could be useful to the community. I'm happy to recommend publication once my points below have been addressed.

We thank the Reviewer for this recommendation for publication.
Here we review a range of publicly available, satellite-derived sea ice thickness data products, and show, for the first time, a side-by-side comparison of the thickness estimates. We disagree that there are "no new science results": we discuss the spatial and temporal variations in sea ice thickness and growth rates across the Arctic Ocean between 2010 and 2018, thereby providing scientific analyses of 8 years of CryoSat-2 observations, which is a longer time-series of satellite-derived thickness observations than has been previously published. In addition, the evaluation of the satellite products using in situ data (airborne and ULS) spans a time period that exceeds any previously-published result (e.g. Laxon et al., 2013; Kwok and Cunningham, 2015, Wang et al., 2016, or Tilling et al., 2018).

I have a few suggestions that I think will improve the paper:

**RC 2.1**. Introduction: I found the introduction cumbersome. I would combine the first two paragraphs into one, and reduce the length by ~half as some of the material you discuss here is not really relevant for the current manuscript. Most of the fifth paragraph (Page 3, lines 12-27) is unnecessary in the intro, particularly as you essentially repeat it all in section 2, so remove – the first sentence can go at the end of the previous paragraph, and combine this with the final intro paragraph.

We have modified the introduction (Section 1) according to this suggestion.

**RC 2.2**. Why don't you use the EM bird data? I believe it is a fairly extensive dataset, is commonly used by groups to assess CS2 thickness data, and is the only technique that directly measures thickness rather than a proxy (draft/freeboard).

We agree that the AEM data are very useful for assessment. However, one of our requirements for both the satellite sea ice thickness data products, as well as the validation data sets, is that they are publicly available to the user community. Unfortunately, we have not found a publicly available version, and the source information for the AEM data used in the previous studies you mention was not provided. It would have been possible to acquire the relevant data from one of the data providers examined here (AWI), as they kindly offered it, but such data do not currently fit the requirements of open access and may have raised concerns about the independence of the study, since it comes from one of the data

providers. While additional evaluation against EM bird data might potentially provide further insights, we conclude that its absence does not impact the results shown in our study, nor is the presented evaluation of the satellite data compromised in any way.

**RC 2.3**. You do not discuss the impact of the different treatments of snow on your product intercomparison. There is a paragraph on the general difficulty of snow in sea ice altimetry in the conclusions (Page 19, lines 14-34) – I don't think this belongs here, I would move it to the discussion – but I would also like to see some discussion on how the different snow treatments impact your results. Importantly, you say that all the CS2 data use the 'modified' Warren climatology (Table 1), however I believe that the CPOM processing uses the basin-mean value of the Warren snow depth over the central Arctic region for MYI and half this value for FYI (at least this is my understanding from Tilling et al. (2018)), whereas I believe AWI/JPL use the spatially varying 'modified' Warren snow depth. I don't know if you realized this but it is not clear in the manuscript? [It might be worth checking with Tilling et al.] This could explain some of the differences in Figure 2 (AWI/JPL overestimate thick ice, underestimate thin ice relative to CPOM) as well as why the AWI and JPL maps look quite similar (they treat snow the same, but different to CPOM), and some of the differences in the in-situ comparison.

Thank you for bringing the Tilling et al. (2018) publication to our attention. While it was not available at the time of writing, it is now published, and we are happy to add this reference to the manuscript. Based on personal communication with the data provider we confirmed that the snow in the CPOM product is developed according to your description above.
We added details about snow treatment in the product descriptions in Section 2.1 and corrected Table 1 accordingly. We now describe how snow is treated per product in Section 2. We have also shortened and modified the paragraph about snow (page 19 lines 14-34 of the original paper) and moved it to the discussion section.

**RC 2.4**. The APP-x data is very poor, and I don't really see what value it brings to the paper, other than to say that it is very poor (you could be more unequivocal about this). It shows almost no interannual or spatial variability, and you find that the mean thickness grows by 1m between Feb and Apr (page 13 line 10) which is physically unbelievable. After showing the APP-x data in Figures 2 and 3, maybe Figure 4, I would consider dropping it from the rest of the manuscript and say in your conclusions that it is unrealistic and shouldn't be used over a CS2 product. I was very surprised that you suggest (page 19, line 8) users should use the APP-x data simply because it is available daily in NRT and Arctic-wide! Judging from your analysis, given that APP-x does not capture interannual or spatial variability, users would be better off simply using the monthly CS2 climatology because at least it captures the spatial variability and does not grow unrealistically thick ice at the end of winter.

Our goal is not to make decisions or assumptions about how potential end-users will use these data products. Rather our goal is to provide an independent assessment of the currently available satellite sea ice thickness data products so as to allow end users to make reliable and informed decisions, based on their "use case". The APP-x product is an operational data product, meaning that it is derived twice daily, provided in a routine data format, and is available year-round in a continuous data stream with consistent latency from a national data centre. Moreover, the APP-x ice thickness data is part of a climate data

record maintained by the National Oceanic and Atmospheric Administration (NOAA). Because this is the only operational ice thickness data product currently available it is of wide interest to the community who rely on such observational inputs delivered in an operational setting. We have provided a balanced summary of both the advantages and disadvantages of the APP-x product in the second paragraph of the conclusions. However, we also acknowledge that none of the data products currently available meet the general needs of the end user, and, in the final paragraph of the manuscript text, we provide a set of recommendations for improvements.

**Minor comments:**

It's not clear what you mean by "contemporary" in the title – occurring at the same time, or occurring in the present? Either way I don't think it's necessary; consider revising: "An assessment of satellite-derived Arctic sea ice thickness data"

By contemporary we mean data that belong to the present. The word "contemporary" is a widely used adjective in scientific publishing and we use it here to indicate "current" data sets. This is to alert the reader to the fact that we do not consider historical satellite thickness datasets that are otherwise good or interesting, such as from ICESat or Envisat.

Page 1, line 19-20: Should read "Among the data compared, the blended..."

Corrected.

Page 2, line 26: Re-write: "The most widely-used thickness datasets are derived from the radar altimeter..."

Sentence revised.

Page 2, line 29: "..retrack*ing*..."

Corrected.

Page 2, line 32: "*...basin-scale* gradients..."

Added "basin-scale".

Page 3, line 4: "Given the variety of sea ice thickness data..."

Corrected.

Page 4, line 8: You should reference Tilling et al. (2018) as well as Laxon, as the data you are using was produced by Tilling et al and is somewhat different from the original Laxon data.

The publication by Tilling et al. (2018) was not available at the time of writing. However, as it is now published, we are happy to add this reference to the manuscript in Section 2.1.1 and, where applicable, we have updated the Tilling et al., 2016 references to Tilling et al., 2018.

Page 5, line 9, Table 1: The CPOM processing uses separate retrackers for leads (Gaussian+Exponential model fit) and floes (threshold), and they apply a correction to account for this.

We added details about the separate approaches taken for ice floe and lead retracking in the CPOM product description in Section 2.1.1 and we modified the description in Table 1.

Page 5, lines 24-29: I don't think it is true to say that the AWI processing "does not differ significantly" from the CPOM processing. They are 'similar', but a number of important differences I can think of from the top of my head: 1) different retracking approaches (I believe AWI apply a 40% or 50% threshold retracker to all waveforms), 2) different sea level interpolation, 3) different waveform discrimination criteria (e.g., right and left sided peakiness), the MSS which you mention, different treatment of snow (see my comments above).

We have modified the lines in question, in paragraph 2 in Section 2.1.2. We added more detail about the AWI processing chain in Section 2.1.2, and we modified Table 1 for the CPOM product following the description in Tilling et al. (2018), page 1211, Section 4.2.2.

Page 6, line 9: I think Kwok and Cunningham (2015) is missing from the bibliography? Please check all references in the text actually appear in the bibliography.

Thank you for pointing this out. We added Kwok and Cunningham (2015) to the references and we checked that all of the other references were complete.

Section 3.1: Calculating a pixel-by-pixel correlation after oversampling the data to 5km doesn't make sense, as it may artificially increase the correlation stats. I would suggest sampling all the data onto an identical 25 km grid for the analysis. Also, why use a 50km search radius for the interpolation when the data are posted on 25 km grids? Surely this will act to smooth the data, also improving the correlation?

We thank the Reviewer for their comment which is consistent with RC1.1 and RC3.2. The gridding approach has been modified. We have clarified this in the methods, Section 3.1. We have also revised all text where the results of the gridded data are discussed. Please see further details in our response to RC1.1.

Page 9, line 13: You should provide some justification for using the CPOM data as your baseline, it seems rather arbitrary.

We choose the CPOM data product as the baseline against which to compare the other products partly because it was the first CryoSat-2 sea ice thickness data made publicly available and is thus the most widely used product. Also, according to the number of citations, the method is the best known with roughly four times the citations for the associated product publications compared to the next most cited product, which is the AWI product. We added an explanation of our choice in Section 3.1.

Page 11, lines 5-9: The phrasing here jarred a bit, as it suggests that the "cooler summer =

thicker ice in 2013/14" result is a "finding" of yours which was "also noted" by Tilling et al. (2015). Obviously, you haven't shown any linkages between temperature and thickness, and I know you didn't mean to imply this, so please rephrase this section to say that "Tilling et al found that... The thickness changes in Figure 2 are consistent with their result."

We modified the text in the first paragraph of Section 4.1 as suggested.

Page 11, line 25: The AWI CS2 data appears to show extensive thin ice (<0.5m) in fall but this is the noise floor of CS2, so not necessarily believable – is there something in the AWI algorithm that causes this? I would like some more discussion/explanation of this.

In response to Reviewer 3, we modified the range and increments of the colour scale used in Figure 2, to more clearly show the thickness gradients. We have also modified the text in the first paragraph of Section 4.2 to more appropriately describe the data shown in Figure 2.

Page 14, lines 1-10, Table 3: You should calculate the anomalies relative to the same baseline period to make the numbers comparable i.e., 2011-2015 considering this is the common baseline.

Thank you for this suggestion. We have modified the analysis so that now the baseline period is 2011-2015, and all anomalies are calculated relative to the mean over that period. We updated the results in Table 3 and the associated interpretations in the text, which occur mainly in Section 4.3, paragraph 4. Also, the term "climatological mean" has been changed to "baseline mean".

Page 14, line 16: "CS2-only products" – judging from figure 7, all of the satellite products miss the thickest/thinnest ice including CS2SMOS?

We agree, and we have modified the text in Section 4.4, paragraph 2.

Page 14, lines 20-21: by what measure does the AWI product most closely align with the ULS draft? Likewise, by what measure does the GSFC data least agree? I would say the CPOM/AWI/JPL show the same agreement from figure 7?

Our statements are based on the agreement between modal draft and modal thickness (assuming a ratio of 0.9), the width of the distributions (full width half maximum/standard deviation), the shape of the thickness distributions, and how well the distributions capture both the distribution of thin ice <1.5 m, and the decay in the distribution for thicker ice >1.5 m. The results shown in Figure 7 have been updated through the addition of ULS draft and satellite thickness data for the 2016-2017 season and the use of the revised CS2SMOS data set. Consequently, we have revised our interpretation of the draft to thickness comparisons and results. We have modified the text in Section 4.4, paragraph 2 to more adequately describe the comparisons.

Section 4.5, paragraph 1, Figure 10: I would think the negative freeboards in the GSFC data must have something to do with the different retracking, as this is the major difference between the data products? I would speculate that the GSFC floe retracking might be

sensitive to off-nadir scattering, as they use a functional fit to the entire waveform and power in the trailing edge could skew the fit to later delay times.

Thank you for this very interesting insight. We agree that the abundance of negative freeboards in the GSFC product is anomalous compared with the other data products and may be a result of the empirical retracker. We also agree that it would be insightful for the community to be able to understand the sensitivity of the CryoSat-2 freeboard estimates to the specific retrackers. This analysis is unfortunately outside the scope of our current study which is targeted towards end users of the existing satellite thickness data products, rather than an assessment of retracker methodology and sensitivities

---

## Author Comment (AC7) · 20 Mar 2019

**Response to Review #1 (Jack Landy)**

**Date: 20 March 2019**

We thank the Editor, reviewers and those who provided short comments on the manuscript for their inputs. The feedback has helped to improve both the clarity and content of the manuscript. We have provided responses to both the short comments and full reviews. We indicated by section, specified by paragraph, where revisions were made within the manuscript text. Since we include figures in response to both the short comments and the full reviews, there is a letter code to indicate if the figure is related to a short comment (e.g. SC1.1) or to a reviewer comment (e.g. RC3.1). The figures in the manuscript itself maintain normal numbering convention (e.g. Figure 1, Figure 2, etc.).

The following is a list of the major changes to the manuscript:

- We have revised the manuscript text for clarity and brevity. In particular we have shortened the Introduction and rearranged the text regarding the treatment of snow depth in each satellite data product.
- Based on input received during the review, we have revised the text of Section 2.1 and the information provided in Table 1 to clarify specific aspects of the processing chain and waveform retrackers used in each satellite thickness product.
- We have extended Table 1 so as to include the details of two additional satellite-derived sea ice thickness data sets, although these data are not included in the further analysis. This decision is a compromise between providing the pertinent details of publicly-available data products, while not overwhelming the reader with too much information in the figures and tables.
- We have replaced the CS2SMOS data set used in the original submission with an updated version of the data set, and revised all figures and tables containing CS2SMOS data.
- We have updated figures and tables wherever possible with new data that has become available since the original submission. In particular, we now include the BGEP ULS ice draft observations for the 2016-2017 season.
- We expanded our results to include winter growth rates, adding a new table (Table 5).
- The reviewers highlighted concerns regarding the original methods used to calculate the correlation between data products, and that using a near-neighbour interpolation with a search radius of 50 km could potentially artificially improve the correlation results. To address these concerns, we have revised the approach to calculate the correlation statistics between the satellite data products, as well as between the satellite and airborne observations. In the revised manuscript the thickness observations are placed onto a common grid (0.4° latitude by 4° longitude) before common grid cells are compared and correlation statistics calculated. This follows the approach originally taken in Laxon et al. (2013) as well as in subsequent studies and allows the reader to place our results in the context of the published literature. We note that this did not change the results of the correlation analyses in a substantial way.

Within the manuscript text, all edits (additions/deletions) are indicated in red font. The manuscript version indicating track changes is posted as a separate author's comment in order to keep the response document concise.

**Assessment of Contemporary Satellite Sea Ice Thickness Products for Arctic Sea Ice**
**Sallila et al., 2018**

This study provides the first genuine inter-comparison of pan-Arctic sea ice thickness data products obtained from three different satellite sensors. It is systematic in approach, equitably comparing the datasets to identify seasonal and interannual patterns of ice thickness variability, as well as assessing biases versus independent reference data. Results from the assessment will prove to be a valuable contribution to the field of Arctic climate science and potentially to operational stakeholders.

We thank the Reviewer for their encouraging comments on the usefulness of our study for the science community and stakeholders.

However, I have some concerns about the robustness of methodology used to process and compare datasets. These concerns focus on the procedure used to resample data, as described in the comments below. I have also made a few suggestions to improve the analysis and impact of the author's findings. I'd recommend this manuscript is published in The Cryosphere, following these revisions.

Please do get in contact if you have questions regarding these comments. Kind regards, Jack Landy

**General comments:**

**RC 1.1.** The method used to re-grid the satellite products may be introducing errors into the data inter-comparison and artificially improving the correlations. By gridding to 5 km with a search radius of 50 km you are introducing significant spatial autocorrelation between adjacent grid cells (through your method cells can theoretically only be treated as independent samples at a length scale 25 km). It is not statistically robust to be calculating the pixel-wise correlation between spatially-dependent samples, and this may be artificially improving the coefficient and significance. Perhaps you could try recalculating the correlation coefficient for a down-sampled 25-km version of your current grid? Take a look at this paper https://www.cambridge.org/core/journals/annals-of-glaciology/article/impact-of-spatial-aliasing-on-seaice-thickness-measurements/AB5DB924171B219D25E64A829E418840 to consider your gridding approach. (This is also worth considering for your OIB comparisons in Fig 9).

Agreed. Originally data were placed on a grid with 5 km resolution using a nearest neighbour interpolation function with a search radius of 50 km. As you and other reviewers have commented, this could lead to an autocorrelation between adjacent grid cells, and an overestimate of the correlation between independent data products. As suggested, we have revised our approach and now place the data sets onto a common 0.4° latitude by 4° longitude grid before calculating the correlations between products. We have revised the description of the gridding procedure in Section 3.1, updated figures 4, 9 and 10, and Tables 2 and 3. We have updated Sections 3, 4, and 5 to reflect the changes in the Figures and Tables. Our revised method follows the approach originally taken in Laxon et al. (2013), and then used in subsequent studies (e.g. Tilling et al., 2018), thereby allowing the reader to place our new results in the context of the previously published statistics.

**RC 1.2.** Linked to the above comment, you suggest at Page 14, Line 19 that the satellite products do not capture the thinnest or thickest ice because of the grid-averaging process. It surely makes sense then, for a fair comparison between satellite obs and the ULS, to average the ULS measurements over some window, rather than binning individual obs in your histograms. I understand this is difficult because it means averaging over time and comparing it to grid averages over space. However, could you identify a time-averaging window based on recorded ice drift speeds (either a seasonally- representative drift speed or obtained from satellite ice motion data)?

We believe the Reviewer may have misunderstood a detail in the original text, regarding the satellite data. The monthly data sets as provided are actually gridded at 25 km resolution (without any additional gridding by us). Regarding the assessment of the satellite data products through comparison with the ULS analysis, we follow the methodology of Laxon et al. (2013) and Kwok and Cunningham (2015). This is to enable the reader to place our results in the context of the existing literature. We have revised the text in Section 4.1 for clarity.

**RC 1.3.** Your correlations between all the CS2 products are understandably very high (Page 12, Line 12) and there's not too much new analysis here on the differences between the CS2-only datasets. I would be very interested to know how closely each product picks up small-scale spatial variability in ice thickness. For instance, you could calculate the anomaly maps of each year compared to the climatological average, then assess the pixel-wise inter-product correlations between positive and negative anomalies. Do some of the CS2 products identify more or less of the smaller-scale thickness features than others? Are they realistic? This analysis would improve e.g. the first paragraph of the discussion.

Our results show for the first time the correlation between a variety of CS2-only products, which differ due to waveform retracking algorithms. We have also significantly extended the evaluation of the CS2 products, through comparisons with independent data over much longer time period (2010-2017) than has previously been published. We are unsure exactly what the Reviewer means by "small-scale spatial variability in ice thickness", since the majority of the satellite data products are provided on grids of resolution 25 km. Sea ice thickness varies on much smaller scales than this, but is not possible to assess it in this analysis of gridded data. However the results in Table 2 provide details of the spatial variability between products at the regional scale. APP-x shows the least variability from region to region, with mean ice thickness in spring across the product record falling within a range of 0.19 m for the southern regions of the central Arctic basin (regions 3-6). JPL shows the largest regional-scale variations, with a range of 0.97 m in average spring thickness across regions 3-6. Table 3 also provides some interesting results on differences between CS2-only products, in terms of seasonal variations relative to the 2011-2015 baseline. There are also differences in the growth rates (Table 5) and wintertime trends (Fig. 5) across the CS2-only products, where the GSFC product has the lowest daily growth rate. Based on these results, we have included additional discussion in Sections 4.2 and 5 about the regional differences.

**RC 1.4.** There is not a lot of analysis or discussion of regional variations in the products: whether any obvious differences are regionally-dependent etc., except from Table 2.

Can you add some deeper discussion on regional differences in the products, and offer some interpretation as to why these differences may exist? This would be a really valuable addition to the work.

Based on the results shown in Table 2 we have added new discussion to the second paragraph of Section 4.2, per this suggestion.

**Minor comments/edits:**

Page 2. Lines 15-16. By what mechanism does Laxon 13 speculate that lower ice volume could be a factor in recent minima in ice extent?

The direct statement is found in paragraph 27 of Laxon et al. (2013), which is as follows: "Finally, we can speculate that the lower ice thickness and volume in February/March 2012, as compared with February/March 2011, may have been one factor behind the record minimum ice extent reached in September 2012". The statement points out that the anomalously low ice volume at the end of winter in April 2012 may have contributed to the sea ice minimum observed in September 2012. We have revised the text (Section1, second paragraph) to clarify that we are referring to the September 2012 minimum.

P2. L29-31. Can you briefly introduce here some of the main differences in approach?

Section 2.1 is dedicated to the description of the satellite data products. We therefore elected to introduce information concerning the retrackers there, and we do not feel that this information is critical for inclusion in the introduction section.

P3. L17-18. This is confusing: you're not including the ICESat data in your evaluation right, so why do you discuss evaluating ICESat here? Do you mean they have been used as a tool for evaluating satellite products in the past?

We have revised the introduction section based on comment RC2.1 and subsequently removed this sentence about ICESat.

P4. L6. What is the time period under investigation?

The period under investigation is October 2010 - April 2018. For some analysis, e.g. the time-series of ice thickness shown in Fig. 5, data are shown based on the availability for each data product. The baseline period for comparisons is the period with full product overlap (spring 2011-fall 2015). We added a brief clarification in Section 2.1, paragraph 1.

P4. L8. What do you mean by mean scattering horizon?

We have revised Section 2.1.1, paragraph 2, based on updated information provided in Tilling et al., 2018 and the comments of Reviewer 2.

P4. L30+. I understand that estimates for snow depth are pretty similar between the CS-only datasets, but you do not refer to them here and they are critical. Can you add some info on how snow depth and density are estimated for each product?

For each product description in Section 2.1 we have added text about snow depths and densities, wherever that information was available.

P6. L16-18. The description of the waveform-fitting method used for the GSFC product needs to be improved. The physical model of Kurtz 14 does not calculate expected returns empirically. Their model is semi-analytical, producing a simulation of the expected echo shape based on statistical parameterizations of the sea ice backscattering properties and height distribution of surface roughness.

We have changed the explanation to better describe the method used to derive the GSFC thickness product, in Section 2.1.4, paragraph 2.

P7. L26-27. What is the physical relationship between the surface EB and ice thickness?

The OTIM model, used in the APP-x product, derives sea ice thickness as a function of heat fluxes, surface albedo and radiation, which all contribute to the surface energy budget. Furthermore, most of the flux and radiation parameters in the equations are functions of surface skin and air temperatures, surface pressure, surface air relative humidity, ice temperature, wind speed, cloud amount and snow depth, which are input parameters in the model. We refer the Reviewer to Wang et al. (2010) where the equations and a thorough explanation are provided. We revised the description of the APP-x data product, in Section 2.1.6, paragraph 2.

P7. L30-31. Why does sea ice roughness affect the surface EB?

We reference Wang et al. (2010), where this was stated in the conclusions. We added a reference to Wang et al. (2010) to the line in question, Section 2.1.6, paragraph 3.

P8. L 5. How is snow depth estimated in the model?

Based on personal communication with the data provider, the initial snow depth estimates are based on those from Warren et al. (1999), but these estimates are then adjusted using field observations and the final snow depth values have been chosen experimentally. We have added information about the treatment of snow in the APP-x sea ice thickness data product to Section 2.1.6.

P9. L2. What measurement of uncertainty is that? Bias? RMSE?

This measurement comes from the error propagation typically used in sea ice thickness remote sensing and follows Giles et al. (2007). The thickness uncertainty is calculated using an assumption of the probable errors associated with the variables used in the thickness equation. We added an explanatory subordinate clause in Section 2.2.2 and added the reference to Giles et al. (2007).

Giles, K. A., Laxon, S. W., Wingham, D. J., Wallis, D. W., Krabill, W. B., Leuschen, C. J., McAdoo, D., Manizade, S. S., Raney, R. K.: Combined airborne laser and radar altimeter

measurements over the Fram Strait in May 2002. Remote Sens. Environ., 111(2–3), 182–194, doi:10.1016/j.rse.2007.02.037, 2007.

P9. L20. It is unconventional to subtract the focus product from the reference – typically it's the other way around. This makes the plots e.g. Fig 3 harder to interpret, because you have products with higher thickness than the CPOM data given as negative numbers.

We have changed the order of product differences in Equations 1 and 2, leading to changes in Figure 3 and the corresponding interpretations.

P10. L3. Why do you pick a 200 km radius? Is this an arbitrary threshold or based on estimated ice drift displacements over the season? Do you use the same threshold in winter and summer? Have you tried varying the threshold +/- 100 km to see if you get the same results?

We used a radius of 200 km, since it is consistent with the original evaluation presented in both Laxon et al. (2013) and Kwok and Cunningham (2015). This allows the reader to place our results in the context of the conclusions of prior studies.

P10. L9. So, one correlation coeff for the whole time period, using each monthly average as a single sample?

We have added clarification on paragraph 2, Section 3.2.

P10. L17. Above you refer to the issue of oversampling at the start or end of a seasonal period and then treating products as the same when comparing. The IceBridge spring data are collected in discrete campaigns, so the period of observations will often not be spread evenly over the March-April period. This seems like the same issue – so how might this be introducing uncertainty into your satellite-OIB product comparisons?

We have revised our methodology to calculate the correlations between the satellite data products and the IceBridge data. We now place the IceBridge and satellite-derived thickness data on a 0.4° latitude by 4° longitude grid to conduct the comparisons. The approach helps to mitigate the uneven spatial and temporal sampling of the sea ice along the IceBridge flight-lines compared to the monthly means obtained from the satellite data products. Also, as we have adopted the same approach originally used in Laxon et al. (2013) and subsequent studies (e.g. Tilling et al., 2018), this allows the reader to place our extended results in the context of the existing literature.

P11. L4. Can you give the mean and +/- (between the techniques) of the fall 2011/2012 ice thickness, to provide the reader with a bit more context? Here and in other similar places within the paper?

Added values for the mean thickness and its variation on the suggested line as well as in other places within Section 4.1.

P11. L18-20. Is there a way you can quantify this? e.g. by calculating the spatial covariance of the gridded data between years, or by simply calculating the average pixel standard

deviation in ice thickness across the period? i.e. how sensitive are each of the products to spatial features of ice thickness like the thick ice tail in the Beaufort Gyre, at what scales?

This was meant as a brief discussion of results in Figure 2, based on visual inspection. Here we did not emphasize exact quantitative differences, as there are plentiful analyses included in the later figures and tables. We agree though that studying the sensitivity of the products, or even one product, to spatial features, would be a great study on its own!

P12. L3-4. For which season?

Good catch, these are for the spring season. We added the season in the second paragraph of Section 4.2.

P13. L 14. And indeed of the thicker sea ice too. . . It's very strange that the technique based on a thermodynamic model and surface EB suggests highest ice growth rates at the end of winter, when you'd expect such a method to be better at capturing the expected reduction in thermodynamic growth.

Unfortunately we have not discovered a reason behind this behaviour in the available references describing this data product. It seems like the air temperature, and other factors that likely support ice growth, outweigh some of the other physical phenomena in the EB model.

P13. L15-34. It is very interesting that the CS2 only products are more similar in spring than in fall, when intuitively you'd expect them to diverge more when the ice is thicker. Can you provide any insight on why this might be? In what ways may the processing differences induce variations in fall that are not apparent in spring?

After applying revisions to the method used to calculate seasonal averages, and obtaining a new version of the CS2SMOS data, we find that the CS2-only products tend to agree quite well in both the spring and the fall. The exception is the GSFC data product which diverges from the other CS2 products in the fall. Figure 3a shows that while ice thickness in the fall is on average thicker in the GSFC product, it is most apparent in regions 5 and 6 (East Siberian and Laptev Seas) and over the thickest multiyear ice in the central Arctic. Recall that the major difference between the GSFC product and the other CS2-only products is that the GSFC retracker uses a waveform fitting method (Table 1).

P14. L 34. What do you mean by 'varying dependency to draft'?

We were attempting to describe the varying correlation values for the three different buoys (Fig. 8). The line in question in Section 4.4 has been deleted from the revised manuscript.

P15. L2-3. Since the thickness is relatively low in this region (<1.5 m) this would imply a higher contribution of SMOS data to the optimal interpolation, right? So does this give you any information about the reliability of SMOS data?

The major difference between the CS2SMOS and AWI products is the inclusion of SMOS data, and this suggests that there are occasions when SMOS estimates higher thickness,

which could be due to the relatively thin ice in this region, as you state. However, we do not make statements about the reliability of SMOS data and refer the Reviewer to Ricker et al., 2017 regarding the CS2SMOS interpolation scheme.

P16. L1-15. It is also worth noting that GSFC freeboard for both fall and spring is generally lower than the AWI freeboard, whereas the GSFC ice thickness was determined as the highest in Figs 2-3 etc. So the positive bias (whether an error or not) in GSFC ice thickness compared to CPOM and AWI is not coming from the freeboard measurements, and must be from the conversion to ice thickness (snow depth, snow or ice density differences, or perhaps filtering/processing chain differences).

We agree that, on average, GSFC freeboard observations are 0.14 meters thinner than corresponding observations from AWI in spring, and 0.08 meters thinner in fall (Fig. 10), despite GSFC indicating higher sea ice thickness for both seasons compared to the AWI product, as shown in Figures 2 and 3. We have included this result in section 4.5, but can only speculate about the cause.

P17. L7. 'freeboards of less than approximately 0.05 m'.

Added the word 'approximately' to the line in question, in Section 5, paragraph 3.

P17. L 23. You suggest the assumption is that SMOS data is too heavily weighted in the combined product, but where do you get this assumption from? Are you making this assumption, or have you got it from previous papers? (if so which ones?). Potentially the uneven thickness increase detected by SMOS is more realistic and the CS2 products cannot detect new thin ice..?

We based this statement on existing studies where it is stated that SMOS assumes 100% ice concentration in its thickness retrieval algorithm, which may cause underestimation of ice thickness in conditions with less than 100% ice concentration, as mentioned in Tian-Kunze et al. 2014 and Ricker et al. 2017. It is certainly possible that SMOS estimates are more realistic for some conditions, most notably thin ice in areas of high ice concentration. We have revised the text of Section 5 (discussion) to reflect the reprocessed CS2SMOS data set used in the revised manuscript.

P18. L 5-8. Re. variations in ice concentration thresholds, you could have defined your own conservative threshold (e.g. 75

The focus of this study is in the assessment of publicly available data products as they are provided to the community. Furthermore, a variety of ice concentration products are used in the product processing chains. Choosing one of those, and applying it routinely across all products, would likely favour some products and hinder others. Table 1 provides further details of the ice concentration data and thresholds applied to each product.

P19. Snow section. This is quite out of place here, as you do not pay much attention to variations in snow treatment between products, and the focus of the paper is on inter-comparison rather than the systematic issue of estimating snow (common between all

products). I suggest removing this passage and if desired, reference the challenge in a single sentence or so.

We have added text in Section 2 about the treatment of snow in each data product, specifying the differences. Although all products use snow climatology estimates derived from Warren et. al. (1999), they employ different ways to implement the climatology. Based on this, the paragraph about snow in Section 6 has been modified and as suggested by Reviewer 2, it is now shorter and most of the text is moved to Section 5, the discussion.

Table 1. This is great! Really useful compilation.

Thank you.

Fig 5. Can you add the linear per winter ice growth rate for each product and year to the plot? i.e. X cm/month.

We expanded our results to include growth rates in a new table, Table 5.

Fig 7. Can you emphasize more clearly on the plot that the bold distribution is ice draft, as a quick glance at the figure gives one the instant conclusion that CS2SMOS data must be best.

Added bold text to the legends in Figure 7 so as to highlight "ULS Draft".

Fig 8. Can you add the x = 0.9y line to the plot as a reference?

The 0.9 factor is an estimate of the ratio between thickness and draft based on Rothrock et al. (2008), and therefore we do not want to emphasise it as the true ratio. Whereas for Figure 7, where the distribution shapes are compared, plotting also ULS draft is crucial.

We have modified the text in the third paragraph of Section 4.4 to emphasize the fact that 0.9 is an estimate of the draft to thickness ratio.

---

## Author Comment (AC8) · 21 Mar 2019

[revised manuscript text omitted]

| | | ULS | CPOM | AWI | JPL | GSFC | CS2SMOS | APP-x |
|---|---|---|---|---|---|---|---|---|
| | **2017** | 1.21
-0.15 | 1.71
-0.17 | 1.61
-0.06 | | 1.69
-0.15 | 1.59
-0.06 | |
| | | **ULS** | **CPOM** | **AWI** | **JPL** | **GSFC** | **CS2SMOS** | **APP-x** |
| | **2011–2016** | 1.37 | 1.88 | 1.62 | 1.81 | 1.81 | 1.52 | 2.08 |
| | **2011** | 1.48

0.11 | 1.7

-0.18 | 1.48
-0.15 | 1.54
-0.27 | 1.66

-0.15 | 1.44

-0.09 | 2.15
0.07 |
| | **2012** | 1.39
-0.02 | 1.75

-0.13 | 1.51

-0.11 | 1.6
-0.21 | 1.66

-0.14 | 1.41

-0.11 | 2.15
0.06 |
| | **2013** | 1.06

-0.31 | 2.06

0.17 | 1.51

-0.11 | 1.59
-0.21 | 1.55

-0.26 | 1.28

-0.25 | 2.16
0.08 |
| **Mooring D** | **2014** | 1.64

0.27 | 2.12

0.23 | 1.89
0.26 | 2.13
0.33 | 2.15

0.34 | 1.86

0.34 | 2.09
0.01 |
| | **2015** | 1.59

0.22 | 2.13

0.25 | 1.97

0.35 | 2.16
0.36 | 2.26

0.45 | 1.93

0.4 | 2.0
-0.08 |
| | **2016** | 1.31

-0.06 | 1.61

-0.27 | 1.42

-0.2 | - | 1.67

-0.14 | 1.26
-0.27 | 1.95
-0.14 |
| | **2017** | 1.13
-0.24 | 1.82
-0.07 | 1.57
-0.05 | | 1.69
-0.11 | 1.49
-0.03 | |

**Table 5. Winter (October-April) sea ice thickness growth rate (md$^{-1}$) for the period 2010-2018 in the central Arctic (regions 1-6), where rates are calculated based on product availability during each growth season.**

| Growth Rate (mm/d) | CPOM | AWI | JPL | GSFC | CS2SMOS | APP-x |
|---|---|---|---|---|---|---|
| 2010-2011 | | | | 0.0032 | | 0.0103 |
| 2011-2012 | 0.0064 | 0.0058 | 0.0057 | 0.0040 | 0.0050 | 0.0105 |
| 2012-2013 | 0.0055 | 0.0048 | 0.0052 | 0.0027 | 0.0044 | 0.0108 |
| 2013-2014 | 0.0058 | 0.0049 | 0.0054 | 0.0026 | 0.0042 | 0.0100 |
| 2014-2015 | 0.0057 | 0.0051 | 0.0053 | 0.0037 | 0.0044 | 0.0103 |
| 2015-2016 | 0.0051 | 0.0050 | | 0.0025 | 0.0040 | 0.0099 |
| 2016-2017 | 0.0050 | 0.0041 | | 0.0022 | 0.0034 | |
| 2017-2018 | 0.0047 | 0.0042 | | 0.0014 | | |